# Variations of polyphenols and carbohydrates of *Emiliania huxleyi* grown under simulated ocean acidification conditions

Milagros Rico[1,2], Paula Santiago-Díaz[1,2], Guillermo Samperio-Ramos[3], Melchor González-Dávila[1,2], Juana Magdalena Santana-Casiano[1,2]

[1]Departmento de Química, Universidad de Las Palmas de Gran Canaria, Campus de Tafira, 35017 Las Palmas de Gran Canaria, Spain.
[2]Instituto de Oceanografía y Cambio Global (IOCAG), Universidad de Las Palmas de Gran Canaria, Unidad Asociada ULPGC-CSIC, Las Palmas de Gran Canaria, Spain.
[3]Instituto de Investigaciones Oceanológicas. Universidad Autónoma de Baja California. Carretera Ensenada-Tijuana nº 3917, Fracc. Playitas, Ensenada Baja California. C.P. 22860.

*Correspondence to: Milagros Rico (milagros.ricosantos@ulpgc.es)*

**Abstract.** Cultures of the coccolithophore *Emiliania huxleyi* were grown under four different $CO_2$-controlled pH conditions (7.75, 7.90, 8.10, and 8.25) to improve understanding of its responses to ocean acidification scenarios. Acidification did not significantly affect final cell densities and carbohydrate contents. Intra- and extracellular phenolic compounds were identified and quantified by Reverse Phase-High Performance Liquid Chromatography (RP-HPLC), with the highest concentrations of total exuded phenolics at pH 8.25 (43±3 nM) and 7.75 (18.0±0.9 nM). Accumulation of intracellular phenolic compounds was observed in cells with decreasing pH, reaching the maximum level (9.24±0.19 attomole cell$^{-1}$) at the lowest pH (7.75). The phenolic profiles presented significant changes in exuded epicatechin and protocatechuic acid ($p<0.05$ and 0.01, respectively), and intracellular vanillic acid ($p<0.001$), which play an essential role as antioxidants, and in the availability of trace metals. A significant increase in chlorophyll *a* content was observed in cells grown at the most acidic pH ($p<0.01$), which also showed significantly higher radical inhibition activity ($p<0.01$). However, no significant differences were found between the iron reducing activities and the radical scavenging activities of the compounds present in the exudates. The nature and concentration of the organic compounds present in the culture medium may favour or inhibit the local growth of specific algal species, and influence trace metal bioavailability affecting the biogeochemical cycling of carbon and microbial functional diversity.

# VARIATIONS OF POLYPHENOLS AND CARBOHYDRATES OF *Emiliania huxleyi* GROWN UNDER SIMULATED OCEAN ACIDIFICATION CONDITIONS

### CO₂-CONTROLLED pH CONDITIONS

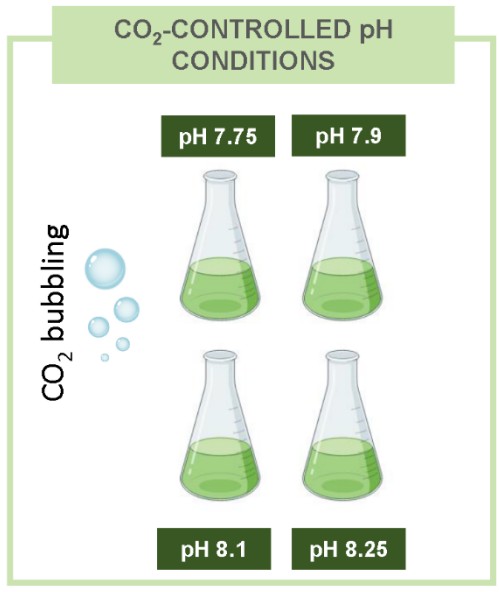

### *Emiliania huxleyi* RESPONSE

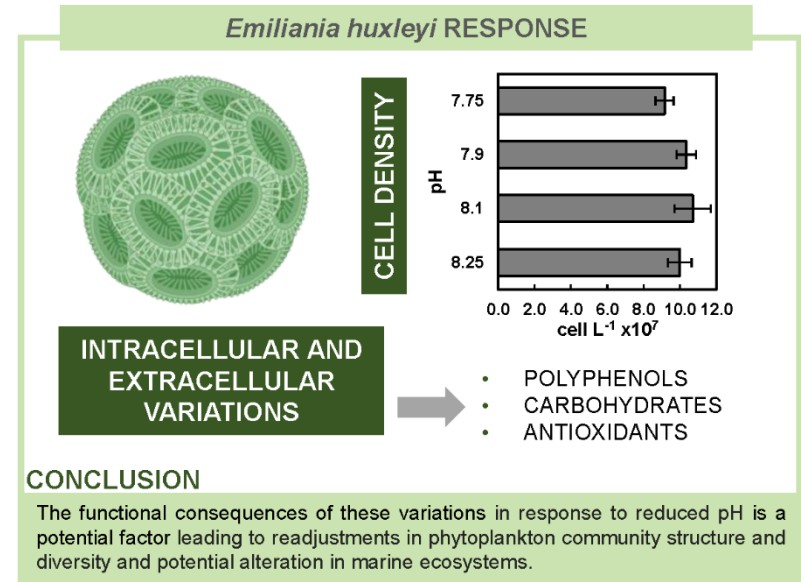

INTRACELLULAR AND EXTRACELLULAR VARIATIONS

- POLYPHENOLS
- CARBOHYDRATES
- ANTIOXIDANTS

**CONCLUSION**

The functional consequences of these variations in response to reduced pH is a potential factor leading to readjustments in phytoplankton community structure and diversity and potential alteration in marine ecosystems.

**Graphical abstract**

# 1 Introduction

Microalgae play a key role in marine ecosystems, forming the basis of the marine food chain as they are responsible for almost half of the total primary production (Usher et al., 2014; Dedman et al., 2023). They constitute a potential feedstock because of their valuable metabolites such as pigments, vitamins, proteins, carbohydrates, and lipids with valued fatty acids. Marine phytoplankton, primarily diatoms and coccolithophorids groups, drive the oceanic carbon cycle by sequestering inorganic carbon from the atmosphere during photosynthesis (Marinov et al., 2010). Coccolithophores are the most abundant calcifying phytoplankton and form gigantic blooms throughout the oceans, especially in mid-latitudes. The coccolithophore *Emiliania huxleyi* (*E. huxleyi*) is the main contributor to calcareous sea sediments, making it particularly sensitive to ocean acidification and playing a crucial ecological role (Arundhathy et al., 2021; Westbroek, 1992). When exposed to elevated $CO_2$ and low pH, the majority of coccolithophores reduce their growth rate and level of calcification, resulting in thinner coccosphaeres (Kholssi et al., 2023; Mackey et al., 2015; Meier et al., 2014). However, calcification is promoted in certain species and/or strains within the same species under enhanced $CO_2$ (Shi et al., 2009).

Global environmental changes, in particular those related to increasing temperature and decreasing pH, profoundly affect ocean ecosystems at many levels, as these are the two main variables controlling all chemical and biological cycles, with a major impact on the growth and metabolic functions of microalgae (Berge et al., 2010; Dedman et al., 2023; Kholssi et al., 2023; Lu et al., 2013). The absorption of anthropogenic $CO_2$ into seawater lowers its pH with adverse consequences for marine ecosystems and human societies (Gruber et al., 2023; Jiang et al., 2023; Lida et al., 2021). In fact, pre-industrial seawater pH (8.25) has already dropped to 8.10, and is expected to reach a pH of 7.85 in this century (Jacobson, 2005). For instance, pH homeostasis, which regulates the pH inside and outside the cell, is critical for the growth and metabolism of most microorganisms, including microalgae (Barakat et al., 2021; Guan and Liu, 2020; Lund et al., 2020). Different algal species show different optimal pH ranges for maximum growth (Hoppe et al., 2011; Kholssi et al., 2023). Changes in environmental pH could have consequences on the competitiveness of both sensitive and tolerant microalgae in mixed phytoplankton communities, modifying their structure, composition, and distribution, which are crucial in mitigating global environmental change by fixing and transporting carbon from the upper to the deep ocean in the major global carbon sink (Eltanahy and Torky, 2021; Kholssi et al., 2023, Marinov et al., 2010). Vasconcelos et al. (2002) found that exudates from *Phaeodactylumn tricornutum* (*P. tricornutum*) diatoms caused a toxic effect on *E. huxleyi*, while those from *Enteromorpha* spp. caused the enhancement of final cell yield, concluding that specific exudates produced by the bloom of one algal species may favour or inhibit the local growth of other species. Such changes could also affect the species at a higher trophic level, resulting in a potential shift in biodiversity (Jin and Kirk, 2018). Spisla et al. (2021) reported that extreme $CO_2$ events modify the composition of particulate organic matter, which leads to a substantial reorganization of the planktonic community, affecting multiple trophic levels from phytoplankton to primary and secondary consumers (Trombetta et al., 2019). Nelson et al. (2020) found modifications of planktonic and benthic communities in response to reduced seawater pH (from pH 8.1 to 7.8 and 7.4), concluding that a re-arrangement of the biofilm microbial communities occurred through a potential shift from autotrophic to

heterotrophic dominated biofilms. In addition, microbial biofilms obtained under reduced pH altered settlement rates in invertebrate larvae of *Galeolaria hystrix*. Barcelos e Ramos et al. (2022) showed that coexistence with other microorganisms modifies the response of *E. huxleyi* to increased $CO_2$, markedly decreasing its growth rate at elevated $CO_2$ concentrations with bacteria *Idiomarina abyssalis* (*I. abyssalis*) and *Brachybacterium* sp. Moreover, elevated $CO_2$ concentrations increased organic

carbon and decreased inorganic carbon content of *E. huxleyi* cells in the presence of *I. abyssalis*, but not *Brachybacterium* sp. Changes in phytoplankton communities due to variation in seawater acidity alter the composition of the organic ligands that these communities release into the surrounding environment (Samperio-Ramos et al., 2017). These ligands are crucial in the formation of metal complexes to acquire micronutrients, sequester toxic metals, and to establish electrochemical gradients resulting in changes in the speciation, the bioavailability, and the toxicity of trace metals (Harmesa et al., 2022; Santana-

Casiano et al., 2014). Iron is an essential micronutrient for phytoplankton involved in fundamental cellular processes, including respiration, photosynthesis, nitrogen uptake, and nitrogen fixation (Raven et al., 1999; Hogle et al., 2014). It controls the productivity, species composition, and trophic structure of microbial communities over large regions of the ocean (González et al., 2019; Hunter and Boyd, 2007). Iron concentrations in ocean waters are very low due to its low solubility and effective removal from the ocean surface by phytoplankton (Liu and Millero, 2002). Complexation with organic compounds is one of

the mechanisms for maintaining dissolved iron concentrations above its inorganic solubility, while potentially reducing the concentrations of soluble and bioavailable inorganic species (Hunter and Boyd, 2007; Shaked et al., 2020). A decrease in seawater pH from 8.1 to 7.4 will increase Fe(III) solubility by approximately 40%, which could have a large impact on biogeochemical cycles (Morel and Price, 2003; Millero et al., 2009). Organic matter exuded by marine microorganisms can form Fe(III) complexes that modify Fe(II) oxidation rates and promote the reduction of Fe(III) to Fe(II) in seawater. Under

acidifying conditions, some research work has shown that the residence time of the reduced form of essential trace metals increases as their oxidation rate decreases (Pérez-Almeida et al., 2022; Santana-Casiano et al., 2014).

Carbohydrates are one of the major components of the dissolved organic carbon (DOC) pool in marine environments, representing 3-50% of dissolved and colloidal organic matter (Giljan et al., 2023; Hassler et al., 2011; Wang et al., 2006). Phenolic compounds are secondary metabolites synthesized as a defense mechanism of organisms exposed to abiotic stresses

(salinity, metal toxicity, heat, acidification, cold, extreme light, nutrient deficiency, UV radiation) (López et al., 2015; Parvin et al., 2022). Their antioxidant nature enhances tolerance of organisms to adverse climatic conditions that induce an increase in reactive oxygen species (ROS) (Gauthier et al., 2020; Vázquez et al., 2022). Both types of compounds can influence the iron chemistry and bioavailability (Pérez-Almeida et al., 2022; Santana-Casiano et al., 2014). Catechin, sinapic acid and gallic acid were found to be weak Fe-binding ligands that increased the persistence of dissolved Fe, regenerating Fe(II) in seawater

from 0.05% to 11.92% (González et al., 2019). Hassler et al. (2011) reported that the addition of glucuronic acid to natural planktonic assemblages increased iron bioavailability for eukaryotic phytoplankton. Furthermore, iron bioavailability also increased when three different saccharides were used in cultured and natural eukaryotic phytoplankton populations, suggesting that this is a generalizable phenomenon.

The effect of pH changes on the speciation of metal-organic complexes and on the redox kinetics of metals in the marine environment is not as well characterized as that of inorganic ligands due to the heterogeneous composition of dissolved organic matter and their unknown structure. Therefore, studying the nature of these organic ligands will allow a comprehensive understanding of the consequences of acidification on ocean biogeochemical processes.

This work aimed to determine how marine acidification may affect the composition of cells and exudates from *Emiliania huxleyi*. Therefore, cultures of *E. huxleyi* were grown at four different pH (7.75, 7.90, 8.10, and 8.25) reached by bubbling $CO_2$ in the culture seawater (Samperio-Ramos et al., 2017). The four experimental scenarios represent interglacial, close to contemporary, and two future ocean acidification conditions based on the Intergovernmental Panel on Climate Change projections (IPCC, 2014). Intra- and extracellular phenolic compounds (gallic acid (GAL), protocatechuic acid (PCA), p-coumaric acid (COU), ferulic acid (FA), catechin (CAT), vanillic acid (VAN), epicatechin (ECAT), syringic acid (SYR), rutin (RU) and gentisic acid (GA)) were identified and quantified by RP-HPLC. The total carbohydrate content of cells was quantified using the phenol/sulfuric acid (PSA) assay (Dubois et al., 1956). The presence of antioxidants was assessed by the antioxidant activity of cells and exudates determined by DPPH radical scavenging activity test (RSA) and FRAP assay (Sethi et al., 2020).

## 2 Materials and methods

### 2.1 Chemicals

Methanol (HPLC gradient grade) was purchased from Scharlab (Barcelona, Spain). Formic acid (synthesis grade), $Fe_3Cl \cdot 6H_2O$, and $FeSO_4 \cdot 7H_2O$ were supplied by Panreac (Barcelona, Spain), D-Glucose (Glc), phenol, DPPH and 2,4,6-tri(2-pyridyl)-triazine (TPTZ) by Sigma-Aldrich (St. Louis, MO, USA). Polyphenol standards were supplied as follows: GAL, PCA, COU, FA, CAT, VAN, ECAT, and SYR by Sigma–Aldrich Chemie (Steinheim, Germany); RU and GA by Merck (Darmstadt, Germany. Ultrapure water was obtained from a Milli-Q system from Millipore (Bedford, MA, USA).

### 2.2 Cultures

Axenic cultures of *E. huxleyi* (strain RCC1238) were supplied by the Spanish Bank of Algae (BEA) in f/2 medium. *E. huxleyi* coccolithophore was cultured with an initial cell density of $10^6$ cells $L^{-1}$ at a constant temperature of 25 ºC, under complete photoperiod (24 h) with light intensity of 200 μmol photons $m^{-2}$ and under different $pCO_2$-controlled seawater pH conditions (7.75, 7.90, 8.10, and 8.25), measured on the free hydrogen ion scale $pH_F = -\log[H+]$ with a Ross Combination glass body electrode calibrated daily with TRIS buffer solutions. For this purpose, a gaseous mixture of $CO_2$-free air and pure $CO_2$ (up to $CO_2$ levels 900, 600, 350, and 225 μatm, respectively) was bubbled in the culture medium (sterile filtered (0.1 μm) North Atlantic seawater (S = 36.48) obtained at the ESTOC site (29º10' N, 15º30' W)) with an equipment that modulates the $CO_2$ flow once the desired pH is reached, keeping it constant (±0.02). To maintain suspended cells and homogeneity of $CO_2$ in solution, the cultures were shaken at 60 rpm with a Teflon-coated magnetic stirrer. All materials were cleaned following a

standard protocol (Achterberg et al., 2001) and subsequently autoclaved at 121 °C for 30 min prior to use. Once the seawater reached the desired $pCO_2$/pH value, coccolithophorides were inoculated into the batch cultures. Stock cultures were maintained in the experimental conditions for 48 h before starting each experiment, which allowed acclimation to the $CO_2$ concentration. Five experimental batch cultures were carried out at each pH treatment. Seawater enriched with exudates was filtered using acid-cleaned and combusted polycarbonate (Nucleopore) syringe-filters (0.45 μm) to prevent cell breakage. Cells were frozen and stored at -80ºC.

Carbonate chemistry was monitored continuously in the experimental media and determined from pH, total alkalinity (TA), and total dissolved inorganic carbon (DIC). pH in the treatments was measured on the free hydrogen ion scale (pH = −log [$H^+$]), by immersing Orion Ross combination glass electrodes in the experimental media. The electrodes were calibrated daily using TRIS buffer solutions. The equilibration of the gas in the media of each treatment was achieved after a maximum of 24 h, observed by the evolution of pH. TA and DIC were measured at the beginning and end of the experiment on days 0 and 8 utilizing a VINDTA 3C system (González-Dávila et al., 2011). TA was determined by potentiometric titration with hydrochloric acid until the endpoint of carbonic acid was reached. DIC was analyzed by coulometric procedure after phosphoric acid addition. Certified Reference Material (provided by A. Dickson at Scripps Institution of Oceanography) was employed to assess the performance of the titration system, yielding an accuracy of 1.5 and 1.0 μmol $kg^{-1}$ for TA and DIC, respectively. The Seawater Carbonate package (Seacarb version 3.0), developed for R Studio software (R Development Core Team), was employed to calculate the values of $pCO_2$, considering the carbonic acid dissociation constants. A more detailed description is given by Samperio-Ramos et al. (2017).

**2.3 Cells and exudates extract preparation prior to HPLC, carbohydrates, and antioxidant activities quantifications**

Cells cultured for 8 days in 700 mL of seawater were freeze-dried and suspended in 5 mL of methanol. The samples were sonicated for 10 min, stirred for 5 min, and centrifuged at 7000 rpm for 15 min. The supernatant was separated, and the residue was extracted with 15 mL of methanol for 30 min and centrifuged again. All supernatants were collected and evaporated to dryness, and the resulting residue was dissolved in 5 mL of methanol. These samples were used for phenolic compounds identification, DPPH inhibition and FRAP values determinations. For carbohydrates, freeze-dried cells were suspended in 5 mL of acidified water (pH=2), the samples were sonicated for 10 min, stirred for 5 min, and centrifuged at 7000 rpm for 15 min and the supernatant was separated.

Seawater samples (700 mL) enriched with exudates were previously subjected to solid phase extraction (SPE) at a flow rate of 2 mL $min^{-1}$ using Macherey-Nagel Chromabond Easy cartridges (500 mg). The retained analytes were eluted with 12 mL of methanol, which was subsequently evaporated on a rotary evaporator. The residue was dissolved in 300 μl of methanol and filtered through a 0.22 μm filter to be injected into the HPLC equipment. These samples were used to quantify phenolic compounds and DPPH inhibition values.

## 2.4 Chlorophyll *a*

Chlorophyll *a* (Chl *a*) was determined according to Branisa et al. (2014) with modifications. Frozen cells were suspended in 5 mL of acetone:hexane (3:2) and sonicated for 3 min in an ice water bath. Homogenates were centrifuged at 7500 rpm for 5 minutes and the absorbance (A) of supernatants was measured at 645 and 663 nm. Chl *a* was expressed as femtogram cell$^{-1}$ and quantified spectrophotometrically according to equation: Chl *a* (mg/100 mL) = $0.999 \times A_{663} - 0,0989 \times A_{645}$.

## 2.5 Phenolic profile of *E. huxleyi* cells and exudates

Methanol extract of cells (2 mL) prepared as described in section 2.3 was evaporated, the residue was dissolved in methanol (300 µL) and filtered with a syringe filter (0.2 µm). Chromatographic analysis of cells and exudates was made according to Santiago-Díaz et al. (2023), with a Jasco LC-4000 HPLC equipment provided with a PU-4180 quaternary pump, an AS-4150 autosampler, an MD-4015 photodiode array detector, LC-Net ll interface, a Phenomenex C18 column (250 mm x 4.6 mm, 5 µm) and a Phenomenex guard column maintained at 30ºC. The elution was performed with water containing 0.1% formic acid (phase A) and methanol (phase B), with a flow rate of 1 mL min$^{-1}$ and injection volume of 10 µL. The gradient elution method for A was 0 min, 75%; 30 min, 40%; 40 min, 40%; and finally, the column was washed and reconditioned. For quantification, simultaneous monitoring was set at 270 nm (GAL, PCA, CAT, VAN, RU, ECAT, and SYR) and 324 nm (GA, COU, and FA). Samples were analysed by triplicate, and the results were expressed as attomole cell$^{-1}$ (amol cell$^{-1}$).

## 2.6 Determination of carbohydrate content

The total carbohydrates contents in *E. huxleyi* cells were determined following the phenol sulfuric acid method (Dubois et al., 1956) with some modifications. The cells extract (1.5 mL) was concentrated, and the residue was dissolved in 0.8 mL of water. The reaction was carried out by mixing 80 µL of sample or standard with 150 µL of phenol (5%) and 1 mL of concentrated sulfuric acid. The resulting solution was heated at 100ºC for 5 min, and the absorbance was recorded at 490 nm in a UV-VIS spectrophotometer. The results were expressed as femtomole of glucose equivalent per cell (fmol Glc eq cell$^{-1}$) calculated from the calibration curve of 0.14 to 2.78 mM glucose, y=0.0032x + 0.0863; $R^2$=0.9971.

## 2.7 Antioxidant activities of algae and exudates

The antioxidant properties of the cell extracts were evaluated according to Sethi et al. (2020) with modifications. The ability to reduce complex TPTZ-Fe(III) to TPTZ-Fe(II) by donating an electron was measured as follows: The FRAP reagent was prepared with acetate buffer solution 0.3 M (pH 3.6), 10 mM of TPTZ in HCl (40 mM), and 2.5 mL of FeCl$_3$·6H$_2$O solution (20 mM) (in the ratio of 10:1:1). Methanol extracts (50 µL) were mixed with 1 mL of freshly prepared and pre-warmed FRAP reagent (37ºC) for 13 min. The mixture was introduced in an ice bath, and the absorbance was measured at 593 nm. Results were expressed as fmol of reduced Fe(III) per cell and calculated from a calibration curve constructed with FeSO$_4$·7H$_2$O

concentrations ranging from 0.15 to 2.20 mM (regression line equation y = 0.3942x + 0.1497, $R^2$ =0.996). The estimation was carried out in triplicate, and the results were averaged.

The capacity of cell extracts and exuded compounds of inhibiting DPPH radical was determined by mixing 0.8 mL of DPPH solution (0.053 mM) with 30 µL of cell extracts, and with 50 µL for each sample of exuded compounds. The absorbance was recorded after 20 min at 515 nm using a UV-visible spectrophotometer (Shimadzu Pharmaspec 1800). The total amount of inhibited DPPH radical per cell was calculated from a calibration curve of 0.03 to 0.1 mM DPPH, y = 11.987x - 0.1352; $R^2$=0.9996.

**2.8 Statistical analysis**

A Pearson's correlation test and a one-way ANOVA were performed to determine the degree of relationship between pairs of variables and statistically significant differences between measurements, respectively. Both studies were conducted using the Jamovi program (2022) and *p*-values of <0.05 were considered statistically significant.

**3 Results and discussion**

**3.1 Carbonate chemistry parameters**

Preliminary tests indicated good stability of carbonate chemistry parameters in media, obtained by the $CO_2$ regulation system (Table 1). After $CO_2$-equilibration, initial DIC values ranged from 1905±26 to 2215±16 at a TA mean value of 2386±16 µmol kg$^{-1}$ (ANOVA; F=1.729, *p*=0.2382 among treatments), corresponding to a $CO_2$ range of 225±1 and 914±13 µatm. Although carbonate chemistry in $CO_2$-manipulated experiments can be strongly affected by biological activity (Howes et al., 2017; Miller and Kelley, 2021) during our research TA and DIC remained fairly stable (t-tests; p>0.05) within treatments, over the 8-day experimental period.

**Table 1.** Carbonate chemistry parameters in experimental media for each pH treatment at day 0 and day 8: total alkalinity (TA), total dissolved inorganic carbon concentration (DIC) and estimated $p$CO$_2$ (µatm).

| pH-Treatments | TA (µmol kg$^{-1}$) | | DIC (µmol kg$^{-1}$) | | $p$CO$_2$ (µatm) | |
|---|---|---|---|---|---|---|
| | Day 0 | Day 8 | Day 0 | Day 8 | Day 0 | Day 8 |
| 8.25 | 2376±12 | 2335±25 | 1905±26 | 1869±61 | 225±1 | 221±4 |
| 8.10 | 2380±15 | 2329±28 | 2012±28 | 1971±44 | 353±2 | 349±5 |
| 7.90 | 2390±17 | 2347±40 | 2129±47 | 2085±36 | 616±12 | 599±8 |
| 7.75 | 2401±14 | 2365±26 | 2215±16 | 2178±39 | 914±18 | 925±27 |

Means and standard deviations were calculated from sampling (n = 3).

## 3.2 Cell growth

The acidification conditions did not significantly affect the final cell densities (Samperio-Ramos et al., 2017), which increased during 8 days from an initial value of $10^6$ cells $L^{-1}$ to 9.98 ($\pm$0.53) $\times 10^7$ (pH 8.25), 1.07 ($\pm$0.13) $\times 10^8$ (pH 8.10), 1.04 ($\pm$0.07) $\times 10^8$ (pH 7.90) and 9.15 ($\pm$0.41) $\times 10^7$ (pH 7.75) (Fig. 1). After 8 days, growth rates remained fairly constant in all experimental acidification conditions, indicating that the stationary phase had been reached. Heidenreich et al. (2019) also concluded that growth rates of both haploid and diploid *E. huxleyi* cells were unaffected by ocean acidification (culture medium adjusted to $p$CO$_2$ 400 vs. 1000 µatm). Fukuda et al. (2014) reported similar results for *E. huxleyi* harvested in CO$_2$-enriched seawater, whose growth was unaffected. However, the authors observed different responses depending on the acidification method, i.e. *E. huxleyi* showed severe growth damage in experiments carried out by acidification by HCl, which also changes the seawater alkalinity that does not happen when bubbling CO$_2$ is used, concluding that coccolithophore *E. huxleyi* has the ability to respond positively to CO$_2$ acidification. In contrast to these results, Vázquez et al. (2022) found that CO$_2$ enrichment aeration up to lower pH than those here (1200 µatm, pH 7.62) induced growth rates of the coccolithophore more negatively affected concerning the control (400 µatm, pH 8.10) than low pH reached without CO$_2$ enrichment.

Discrepancies found in the literature regarding acidification effects and responses of *E. huxleyi* coccolithophores may be due not only to different environmental factors and culture conditions, etc. (Gafar et al., 2019; Tong et al., 2017). Langer et al. (2009) observed substantial differences in sensitivity to acidification in four different *E. huxleyi* strains with different responses in all parameters tested.

**Figure 1** near here.

Bautista-Chamizo et al. (2019) exposed microalgae *Tetraselmis chuii* (*T. Chuii*), *Nannochloropsis gaditana* (*N. gaditana*) and *P. tricornutum* separately to pH 6.0, 7.4 and pH 8.0 as a control, observing growth inhibition of 12%, 61% and 66% at pH 6.0, respectively, with *T. chuii* being the most resistant species to CO$_2$ enrichment. At pH 7.4 only *P. tricornutum* showed a significant decrease in cell abundance (16%), while no differences were found for *T. chuii* and *N. gaditana*, demonstrating that sensitivity to acidification depends on the microalgae species. However, the growth rate of all three microalgae *Chlorella* sp., *P. tricornutum* and *Chaetocetos muelleri* (*C. muelleri*) harvested at pH levels 8.1, 7.8, and 7.5 was significantly enhanced (Jia et al., 2024). The differences observed in the growth behavior of microalgae, mainly of *P. tricornutun* diatoms, in these two last studies could be due to the culture conditions, e.g. in the first study, Bautista-Chamizo et al. (2019) used seawater collected in the Bay of Cádiz (Spain) (with initial pH 8.0, salinity of 34, temperature at 23 °C and initial cell density $3 \times 10^4$ cells mL$^{-1}$), and in the second study Jia et al. (2024) used artificial seawater prepared by mixing sea salt (Haizhixun, China) with purified water (salinity of 30 $\pm$ 1%, temperature maintained at 18 $\pm$ 0.5°C, initial pH at 8.1, no data on initial cell density).

### 3.3 Chlorophyl *a*

After 8 culture days, the concentration of Chl *a* per cell decreases with decreasing pH from 56.6±2.8 fmol cell$^{-1}$ (pH 8.25) to 26.8±1.4 fmol cell$^{-1}$ (pH 7.9). However, cells grown in the most acidic conditions (pH 7.75) show the highest amount of Chl *a* (67.3±2 fmol cell$^{-1}$) with a significant increase observed between pH 8.1 (45.1±3 fmol cell$^{-1}$) and 7.75 ($p<0.01$). These results agree with those reported by Vázquez et al. (2022), who observed significantly increased cellular concentration of Chl *a* in the high $pCO_2$ treatment of *E. huxleyi* cells (1200 µatm, pH 7.62) with respect to control (400 µatm, pH 8.10). Crawford et al. (2011) also studied the effect of elevated $CO_2$ on cultures of the diatom *Thalassiosira pseudonana* CCMP1335 in a pH range between 7.8 and 8.1 (760 µatm and 380 µatm, respectively), concluding that chlorophyll content increased in cells grown at pH 7.8 compared to those grown at pH 8.1, but photosynthetic efficiency remained constant in both experiments. Higher level of Chl *a* were also observed in microalgae *Chlorella* sp., *P. tricornutum,* and *C. muelleri* harvested at pH 7.8 and 7.5 than that quantified at pH 8.1, concluding that primary producers that can utilize $HCO_3^-$ as the carbon source benefit from elevated $CO_2$ concentration (Jia et al., 2024). However, Yu et al. (2022) found no significant changes in cellular and total content of Chl *a* of *E. huxleyi* cells harvested at similar pH than those here (between 7.72-7.76 under elevated $CO_2$ (1000 µatm) and between 8.03 and 8.07 under atmospheric $CO_2$). Mackey et al. (2015) concluded that photosynthetic responses to ocean acidification are highly variable throughout species and taxa, with different *E. huxleyi* strains exhibiting opposite responses to elevated $CO_2$ on maximum photosynthetic rates.

### 3.4 Phenolic contents of cells and exudates

The phenolic profile of cell extracts and seawater samples enriched with *E. huxleyi* exudates are summarized in Table 2. GAL, PA, ECAT, and RU were detected in both cells and exudates, whereas VAN and COU were only detected in cells. The other polyphenols were below the limit of quantification (LOQ). At all pH conditions, PA, ECAT, and RU were detected in the seawater enriched with exudates and VAN in cells.

**Table 2.** Amounts of intracellular and exuded phenolic compounds by cells of *E. huxleyi* grown under different pH conditions.

| Phenolic compound | pH 7.75 ($pCO_2$ 900µat) | | pH 7.90 ($pCO_2$ 600µat) | | pH 8.10 ($pCO_2$ 350µat) | | pH 8.25 ($pCO_2$ 225µat) | |
|---|---|---|---|---|---|---|---|---|
| | Cell | Exuded | Cell | Exuded | Cell | Exuded | Cell | Exuded |
| GAL[a] | 0.13±0.03 | - | - | - | 0.25±0.07 | - | - | 45±0 |
| PCA[a] | - | 151±12 | 0.16±0.01 | 34±2 | - | 26.3±0.2 | - | 298±22 |
| ECAT[a] | 1.2±0.1 | 34±1 | - | 40±5 | - | 70±2 | - | 77±6 |
| VAN[a] | 6.44±0.06 | - | 2.5±0.1 | - | 2.30±0.05 | - | 2.5±0.2 | - |
| COU[a] | - | - | - | - | 1.4±0.2 | - | - | - |
| RU[a] | 1.47±0 | 12.1±0.4 | | 20±2 | | 13.2±0.3 | | 11.2±0.8 |

| | | | | | | | | |
|---|---|---|---|---|---|---|---|---|
| Sum[a] | 9.24±0.19 | 197.1±12.3 | 2.66±0.11 | 94±9 | 3.95±0.32 | 109.5±2.5 | 2.5±0.2 | 431.2±28.8 |
| Concentration (nM)[b] | | 18.0±0.9 | | 9.6±0.8 | | 11.7±0.3 | | 43±3 |

[a]Results are expressed as attomole cell$^{-1}$ (means ± standard deviation of two measurements).

[b]Results are expressed as nanomole L$^{-1}$ (means ± standard deviation of two measurements).

Abbreviations: GAL: gallic acid; PCA: protocatechuic acid; ECAT: epicatechin; VAN: vanillic acid; COU: p-coumaric acid; RU: rutin.

Samperio et al. (2017) reported an increase in dissolved organic carbon (DOC) exudation by 19% and 15% during exponential and stationary phases, respectively, as $CO_2$ levels increased from 225 µatm (177.06 ± 10.95 fmol C cell$^{-1}$ day$^{-1}$) to 900 µatm (209.74 ± 50.00 fmol C day$^{-1}$ cell$^{-1}$) in the culture medium. The authors suggested that ocean acidification could significantly enhance the release of phenolic compounds when *E. huxleyi* is grown under low-iron conditions. They detected phenolic compounds only in the stationary phase and their release rate was affected by $CO_2$ conditions, observing a strong correlation

between the concentrations of produced phenolic compounds with exuded DOC, indicating that these compounds constituted a relatively constant fraction of the organic matter excreted by *E. huxleyi*. Extracellular release of phenolic compounds was statistically higher at pH 7.75 (0.41 ± 0.02 fmol C cell$^{-1}$ day$^{-1}$) than at pH 8.1 (0.36 ± 0.02 fmol C cell$^{-1}$day$^{-1}$; Tukey contrast: t value=2.495; $p<0.1$). While Samperio et al. (2017) studied the total polyphenol contents through the Arnow spectrophotometric assay, in this study individual polyphenols were identified (Table 2), concluding that the highest

concentrations of these polyphenols identified were exuded in the cultures at pH 8.25 (43±3 nM) and 7.75 (18.0±0.9 nM) (Fig. 2).

**Figure 2** near here.

The content of phenolic compounds inside the cells increased significantly ($p<0.01$) at pH 7.75, reaching the maximum level

(9.24±0.19 amol cell$^{-1}$) compared with that at pH 8.1 (Fig. 2). These results agree with those reported by Jin et al. (2015), who evidenced that phytoplankton grown under the $CO_2$ levels predicted for the end of this century showed accumulation of phenolic compounds, increased by 46–212% compared with that obtained at the current $CO_2$ level. Subsequently, zooplankton fed with phytoplankton grown in acidified seawater showed 28% to 48% higher phenolic content. This transfer of accumulated phenolic compounds to higher trophic levels could have serious consequences for the marine ecosystem and seafood quality.

Dupont et al. (2014) have shown that survival of adult boreal shrimp (*Pandalus borealis*) was affected when exposed for 3 weeks at pH 7.5 with an increase in mortality of 63% compared to those grown at pH 8.0 (pH at the sampling site), showing also changes in appearance and taste. Polyphenol accumulation has also been observed in terrestrial plants (Bai et al., 2019; Kaur et al., 2022; Lwalaba et al., 2020). In contrast, no significant changes were observed by Jia et al. (2024) in the three microalgae species *Chlorella* sp., *P. tricornutum,* and *C. muelleri* harvested under elevated $p$CO$_2$ (pH 8.1, 7.8, and 7.5), and

Arnold et al. (2012) also reported a loss of phenolics in the seagrasses *Cymodocea nodosa, Ruppia maritima*, and *Potamogeton perfoliatus* grown in acidified seawater, where the pH decreased up to 7.3, and the $CO_2$ level increased ten-fold.

Vázquez et al. (2022) showed that $CO_2$ enrichment aeration (1200 µatm, pH 7.62) induced metabolic stress and accumulation of ROS in the *E. huxleyi* strain used in their experiments. This could explain the accumulation of phenolic compounds at pH 7.75 ($9.24\pm0.19$ amol cell$^{-1}$), considered to play a significant role as a defense mechanism due to their antioxidant and ROS scavenging properties (Chvátalová et al., 2008; Gauthier et al., 2020; Shay et al., 2015). Previous studies on diatoms *P. tricornutum* grown under copper stress showed a significant correlation (r=0.9999; p<0.05) between accumulated phenolic compounds and malondialdehyde (MDA), commonly produced by an increase in ROS (Rico et al., 2024). ROS production was also correlated with acidification stress conditions in single and multispecies toxicity tests performed by Bautista-Chamizo et al. (2019) using *T. Chuii, N. gaditana* and *P. tricornutum* microalgae, where *P. tricornutum* and *N. gaditana* exhibited a significant increase in the percentage of intracellular ROS when exposed to pH 7.4 and pH 6.0, which was more pronounced for *N. gaditana* cells at pH 6.0. Consistent with the loss of cell membrane integrity observed after 48 h at pH 6.0, a 10% and 56% of non-viable cells were found for *P. tricornutum* and *N. gaditana*, respectively, while viable cells remained close to the control for *T. chuii* cells.

At the lowest pH 7.75 in our study, the intracellular phenolic content experienced the maximum increase with respect to the initial one (3.7 times higher). More interesting is the significant increase of VAN inside the cells (*p*<0.001) and its absence outside. VAN does not exhibit iron chelating ability due to the lack of catechol or galloyl moiety, but inhibits Fe(II) oxidation and scavenges radicals (Chvátalová et al., 2008). On the other hand, RU, ECAT and GAL detected inside cells only at this more acidic pH are also known radical scavenger with ability to neutralize ROS in the cells and with reducing power and metal chelating abilities (Chvátalová et al., 2008; Gauthier et al., 2020; Shay et al., 2015). Therefore, this increase of phenolics may be explained by their potential to scavenge harmful ROS, whose enhanced production has been linked to the pH decrease by Vázquez et al. (2022) and Bautista-Chamizo et al. (2019).

No correlation was found between phenolic compounds inside and outside the cells considering the pH 8.25 data. The identified exuded compounds significantly decreased (*p*<0.05) from $43 \pm 3$ nM at pH 8.25 to $9.6\pm0.8$ nM at pH 7.90 (Table 2). These results partially agree with those reported by López et al. (2015) for *Dunaliella tertiolecta* growing under stress conditions induced by high levels of copper, where the concentration of phenolic compounds declined from $9.4\pm0.6$ nM in seawater cultures without Cu addition to $8.4\pm0.4$ and $8.6\pm0.4$ nM in the two copper enrichment experiments, and increased 1.4 times concerning the control into the cells grown under the highest Cu level. However, the extracellular phenolic compound's behavior followed the same tendency to that described above for intracellular between pH 8.1 and 7.75, where a high correlation was found between intra- and extracellular phenolic compounds (*p*<0 .001). Total exuded phenolic compounds identified here were significantly enhanced in this pH interval (p<0.05) to $18.0\pm0.9$ nM at the predicted pH for future scenarios (pH 7.75) over those quantified at pH 8.1 ($11.7\pm0.3$). Significant differences were found in PCA and ECAT amounts (p<0.01 and 0.05, respectively), which together with RU were detected in all exudates. Wu et al. (2016) studied the interaction between Fe and ten phenols at pH 8.0, finding that Fe(II) was rapidly oxidized under alkali condition (pH $8.0\pm0.1$) even in the presence

of gentisic acid, syringic acid, p-coumaric acid, vanillic acid, and ferulic acid, which did not show any protection effect for ferrous iron in these conditions. However, caffeic acid, GAL and PCA protected 69%, 64% and 33% of the initial concentration of Fe(II), respectively, due to the chelating capacity of the catechol and galloyl groups with Fe(II). They reported the formation of relatively stable phenolic-Fe complexes under alkaline conditions helping to weaken iron precipitation. Therefore, the high level of polyphenols with chelating ability exuded at pH 8.25 in this study could be due to the redox chemistry of inorganic Fe, intimately linked to the pH (Pérez-Almeida et al., 2019), specifically the low Fe concentration under these conditions. When the pH increases, the oxidation rate constants of Fe(II) also increases and the solubility of inorganic Fe(III) decreases. Polyphenols modify Fe(II) oxidation rates by promoting the reduction of Fe(III) to Fe(II) in seawater. The effect of GAL on Fe oxidation and reduction was studied by Pérez-Almeida et al. (2022), concluding that it reduces Fe(III) to Fe(II) in seawater, with a more pronounced effect as pH decreases, allowing Fe(II) to be present for longer periods and improving its bioavailability. The authors found that 69.3% of the initial Fe(II) was oxidized after 10 min at pH 8.0 in the absence of GAL, while only 37.5% was oxidized in its presence at a concentration of 100 nM after 30 min. The reduction of Fe(III) to Fe(II) by GAL was faster as pH decreased. The same results were observed for catechin and sinapic acid, which also favoured the regeneration of Fe(II) in seawater, increasing the amount of regenerated Fe(II) as pH decreased, concluding that acidification may contribute to an increase in the level of reduced iron in the environment (Santana-Casiano et al., 2014). This could be the reason for the decrease of GAL and the remaining phenolic compounds in the extracellular medium until pH 7.9 is reached, where acidification does not seem to produce extreme adverse effects in *E. huxleyi*, as observed in other species (Bautista-Chamizo et al., 2019), and the increase of their presence inside the cell as the pH decreases. Changes in the bioavailability of Fe and other trace metals linked to pH variation could affect the control of phytoplankton growth and have a major influence on the biogeochemical cycling of carbon and other bioactive elements in the ocean (Shaked et al., 2020; Tagliabue et al., 2017).

**3.5 Total carbohydrate content of cells**

Total carbohydrates in cells and exudates of *E. huxleyi* are shown in Table 3. No correlation was found between intra- and extracellular contents. During the exponential phase, the contribution of dissolved carbohydrates to excreted DOC was higher (18–37%) than during the stationary phase (14–23%), significantly increased as time elapsed from the exponential to the stationary phase (Samperio et al., 2017). However, acidification of the culture medium with $CO_2$ did not affect the levels of carbohydrates exuded per cell in any of the three growth phases, as these levels did not change significantly in any of them as the pH dropped to pH 7.75. The amount of total intracellular carbohydrates also remained constant between pH 8.25 and pH 7.75. The coccolithophore *E. huxleyi* have shown diverse metabolic responses to ocean acidification and to combinations of ocean acidification with other environmental factors with significant differences between strains (Gafar et al., 2019; Langer et al., 2009; Mackey et al., 2015; Tong et al., 2017). Fukuda et al. (2014) observed increased production and storage of polysaccharides stimulated by acidification with $CO_2$ enrichment. Jones et al. (2013) compared the response of the coccolithophore species *E. huxleyi* cultured at two pH conditions reached by bubbling $CO_2$ (pH 7.94 and 7.47 at the time of the harvesting), when as little as 5% of DIC was consumed, indicating low chemical shift throughout the experiments. The

higher $CO_2$ level evidenced cellular responses to stress such as decreased growth rates, but proteins associated with many key metabolic processes remained unaltered, thus maintaining many biological functions. Xie et al. (2021) studied the effects of high and low DIC concentration (from 900 to 4,930 $\mu$molkg$^{-1}$) and reduced pH value (from 8.04 to 7.70) on physiological rhythms, element contents and macromolecules of the coccolithophore *Emiliania huxleyi*, concluding that its response is highly dependent on the DIC. Compared to high pH conditions, low pH and DIC concentration led to increases in particulate organic carbon (POC) and particulate organic nitrogen (PON) contents with less impact on protein and carbohydrate contents; however, high DIC and low pH reduced POC, PON, protein, and carbohydrate contents. These results partially agree with those reported by Jia et al. (2024), who found that lowering seawater pH up to 7.8 and 7.5 promoted protein synthesis in microalgae *Chlorella* sp., *P. tricornutum*, and *C. muelleri* compared to those harvested at pH 8.1, concluding that acidification improves the efficiency of carbon assimilation and provides more carbon skeletons for amino acids and protein synthesis.

**Table 3.** Intracellular and extracellular carbohydrates of *Emiliania huxleyi* harvested under different pH conditions

| pH | Intracellular carbohydrates[a] | Extracellular carbohydrates[b] |
|---|---|---|
| 8.25 ($p$CO$_2$ 225$\mu$at) | 27.0±2.2 | 44.43±2.98 |
| 8.10 ($p$CO$_2$ 350$\mu$at) | 30.4±3.2 | 45.74±4.93 |
| 7.90 ($p$CO$_2$ 600$\mu$at) | 30.3±3.7 | 47.39±7.36 |
| 7.75 ($p$CO$_2$ 900$\mu$at) | 23.8±1.9 | 46.95±3.92 |

[a]The results are expressed as femtomole glucose equivalent cell$^{-1}$

[b]Extracellular released carbohydrates from Samperio et al. (2017) (femtomole C cell$^{-1}$day$^{-1}$).

The results are expressed as means ± standard deviation of three measurements.

Grosse et al. (2020) investigated the effects of seawater acidification on dissolved and particulate amino acids and carbohydrates in arctic and sub-arctic planktonic communities in two large-scale experiments in a pH range similar to ours here (control mesocosm: $p$CO$_2$ 185 $\mu$atm /pH 8.32; mesocosm $p$CO$_2$ between 270 and 1420 $\mu$atm/pH 8.18–7.51). The authors concluded that the relative composition of amino acids and carbohydrates did not change as a direct consequence of increased $p$CO$_2$, and the observed changes depended mainly on the composition of the phytoplankton community. Araujo and Tabano (2005) reported that CO$_2$ addition to the culture seawater lowered the carbohydrates content of marine diatom *Chaetoceros* cf. *wighamii*. Thornton (2009) observed that the planktonic diatom *Chaetoceros muelleri* grown at pH 6.8, 7.4, 7.9, and 8.2 showed a decreased proportion of total carbohydrate within the cells and increased levels of dissolved exuded carbohydrates into the surrounding medium with the decrease in pH. Thornton used a different method to maintain pH, a biological buffer

(25 mM HEPES (Sigma-Aldrich, St. Louis, Mo, USA)) and daily titration with the addition of a small amount of HCl, which
changes the seawater alkalinity.

Similar effects to those observed here have been found in the response of macroalgae to elevating ocean acidification. Gao et al. (2017) reported that $pCO_2$ does not affect the carbohydrate content of algae *Ulva rigida* investigated under pH 7.95 and 7.55. Barakat et al. (2021) studied the effect of the acid stress on the green alga *Ulva fasciata* subjected to four levels of $pCO_2$, 280, 550, 750, and 1050 µatm (pH values 7.2, 7.6, 7.86, 8.1 (control)), and found similar production of carbohydrates at the
three lowest pH (46.13, 46.96 and 46.04% of dry weight respectively), while the control (pH 8.1) showed 42.37% of dry weight.

### 3.6 Antioxidant activities

Table 4 shows the antioxidant activities of extracts of cells *E. huxleyi* grown in seawater enriched with several levels of $CO_2$ and those of the compounds exuded by these cells. Cells with the highest phenolic content (pH 7.75) gave the most increased
scavenging activity (8.1±0.1 fmol cell$^{-1}$) ($p<0.01$), indicating that these cells grown in the most acidified media produce relevant amounts of antioxidants with ROS scavenging ability. In fact, a high correlation was found between the phenolics identified inside the cells and their antioxidant activity ($p<0.01$). These results partially agree with those reported by Santiago-Díaz et al. (2023), who found enhanced radical scavenging activities and production of polyphenols in diatoms *P. tricornutum* grown under Cu stress as the Cu level increased. However, the ability to reduce iron (FRAP) remains fast constant in the cells,
but also correlates with the intracellular phenolic content ($p<0.05$). The FRAP test is based on the transfer of an electron, whereas the DPPH test includes the transfer of both a hydrogen atom and an electron. These different chemical reactions mechanisms and their kinetics could explain the differences between FRAP and DPPH results. In addition, the antioxidant activity of extracts and exudates depends on their complex mixture, ratios of components and their interactions, which may be synergistic, additive or antagonistic, as well as the assay mechanism/kinetic (Šamec et al., 2021; Sethi et al., 2020). Concerning
the exudates, no significant differences were found in their RSA despite significant changes in the total exuded phenolic compounds per cell were detected ($p<0.05$).

**Table 4.** Antioxidant activities of exudates and extracts of *E. huxleyi* cells grown in seawater enriched with $CO_2$.

| pH | FRAP[a] | RSA | |
| --- | --- | --- | --- |
| | | Cells[b] | Exudates[c] |
| 8.25 ($pCO_2$ 225µat) | 9.5±0.6 | 3.4±0.2 | 26±1 |
| 8.10 ($pCO_2$ 350µat) | 9.3±0.3 | 2.3±0.3 | 37±2 |
| 7.90 ($pCO_2$ 600µat) | 9.3±0.7 | 6.0±0.2 | 40±2 |
| 7.75 ($pCO_2$ 900µat) | 10.2±0.5 | 8.1±0.1 | 26.1±0.4 |

[a]Results are expressed as fmol of reduced Fe(III) per cell.

[b]The results are expressed as fmol of inhibited DPPH per cell

[c]The results are expressed as nmol of inhibited DPPH L$^{-1}$

All results are expressed as means ± standard deviation of three measurements

Abbreviations: FRAP: ferric-reducing antioxidant power; RSA: radical scavenging activity

The results found in the literature on the effect of acid stress on the biochemical composition of microorganisms and their exudates are apparently contradictory. However, a large number of factors (initial cell density, grown phase, culture media composition, light photoperiod and light intensity applied during culture, pH, among others) strongly influence the toxicity of pollutants in algal bioassay stimulating and/or inhibiting the production of different metabolites through the regulation of metabolic pathways (Barcelos e Ramos et al., 2010; Santiago-Díaz et al., 2023; Singh and Shrivastava, 2016; Tandon et al.,

2017; Zhang et al., 2019) and making the responses of marine organisms to ocean acidification highly complex and dependent on these factors (Grosse et al., 2020). Changes in seawater pH influence the protonation of biological molecules and could modify their charge and negatively affect metabolic processes. Several mechanisms have been described to maintain the pH inside cells, such as changing the lipid composition of the cytoplasmic membrane or increasing the production of cyclopropane fatty acids to reduce proton permeability (Lund et al., 2020).

Changes in pH generate stress conditions, either because they drastically decrease the availability of trace metals such as iron, essential in the life cycle of microalgae, which modify the composition of their exudates to protect and internalise Fe(II) at alkaline pH where it is rapidly oxidised to Fe(III) (Wu et al., 2016) or because ROS are increased at acid pH (Bautista-Chamizo et al., 2019; Vázquez et al., 2022). On the other hand, the pH range for optimal growth varies depending on the microalgae species, and even the sensitivity of *E. huxleyi* to acidification appears to be strain-specific (Langer et al., 2009). Borchard and

Engel (2015) reported that *E. huxleyi* ability to acclimate to different $CO_2$ concentrations during the stationary phase of growth was responsible for the absence of a $CO_2$ effect found on primary production and exudation in their study. However, numerous experimental studies show that most coccolithophores cultured at elevated $CO_2$ reduce their level of calcification with a tendency to produce degraded or aberrant coccoliths that calcify even less and will spread in a future high $CO_2$ ocean (Langer et al., 2009; Lohbeck et al., 2012; Mackey et al., 2015). The ecological consequences of ocean acidification on the

competitiveness of coccolithophores in mixed phytoplankton communities are unclear. The microalgae species *T. Chuii*, *N. gaditana* and *P. tricornutum* showed a higher cell abundance when grown alone under acidification conditions than in multispecies toxicity tests (Bautista-Chamizo et al., 2019). In addition, although oxidative stress in *N. gaditana* was significantly higher in the single-species tests, the ROS levels were higher in *P. tricornutum* and *T. chuii* in the multispecies tests, suggesting that algal interactions had an effect on pH toxicity in these species, but in a species-specific way. However,

the effects of competition between these three species were recorded at pH 8, but they were eclipsed by the effect of $CO_2$ acidification. Several studies evidenced enhanced oxidative stress in cells in acidified seawater, through an increase in ROS that correlated significantly with the accumulation of polyphenols (Bautista-Chamizo et al., 2019; Vázquez et al., 2022). These changes may cause cascading effects in the marine food web, varying the macromolecular composition of consumers (Jia et

al., 2024). Ocean acidification will directly affect marine organisms, altering the structure and functions of ecosystems. The accumulation of phenolic compounds leads to functional consequences in primary and secondary producers, with the possibility that fishery industries could be influenced as a result of progressive ocean change (Gattuso et al., 2015; Jin et al., 2015; Trombetta et al., 2019).

**Conclusion**

Global environmental change influences the growth and metabolic functions of microalgae affecting their communities' structure and compositions depending on the sensitivity of different taxonomic groups, with implications for higher trophic levels and biodiversity loss. Limited research has been focused on the ecological consequences of joint action of seawater pH decrease, warming, and changes in salinity, among others. Acidification between the current pH of the oceans (8.1) and the future scenario of pH 7.75 leads to an increase in polyphenol production in *E. huxleyi* cells and their free radical inhibitory activity. More importantly, the change in the polyphenol profile between cells and exudates and between pH conditions should be closely related not only to their antioxidant activity under stress conditions, but also to the chemistry of iron and other trace metals and their bioavailability under different pH conditions. Intra- and extracellular carbohydrate levels did not show modifications with decreasing pH. These changes in metabolites with different capacity to inhibit radicals and complex metals, whose accumulation is associated with enhanced oxidative stress, are potential factors leading to readjustments in phytoplankton community structure and diversity and possible alteration in marine ecosystems.

*Data availability.* Data are available from the corresponding author upon reasonable request.

*Author contributions.* MR: design and conceptualization, methodology, supervision, SPE seawater concentration, preparation of samples for HPLC analysis, FRAP and DPPH activity assays, data analysis, and wrote the manuscript; PS-D: Carbohydrate and HPLC phenolic analysis; GS-R: cells grown study; JMS-C and MG-D: Design and conceptualization of cells culture, critically review of the manuscript, project management, funding and resourcing. All authors approved the submitted version.

*Financial support.* This study received financial support from the FeRIA project (PID2021-123997NB-100) by the Spanish Ministerio de Ciencia e Innovación. The participation of Paula Santiago was funded through a PhD scholarship from the

Universidad de Las Palmas de Gran Canaria (PIFULPGC-2019) to join the Ph.D. Program in Oceanography and Global Change (DOYCAG). The program is promoted by the Institute of Oceanography and Global Change (IOCAG).

*Competing interest.* The authors declare that they have no financial or non-financial interests to disclose. No conflicts apply
to this study.

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

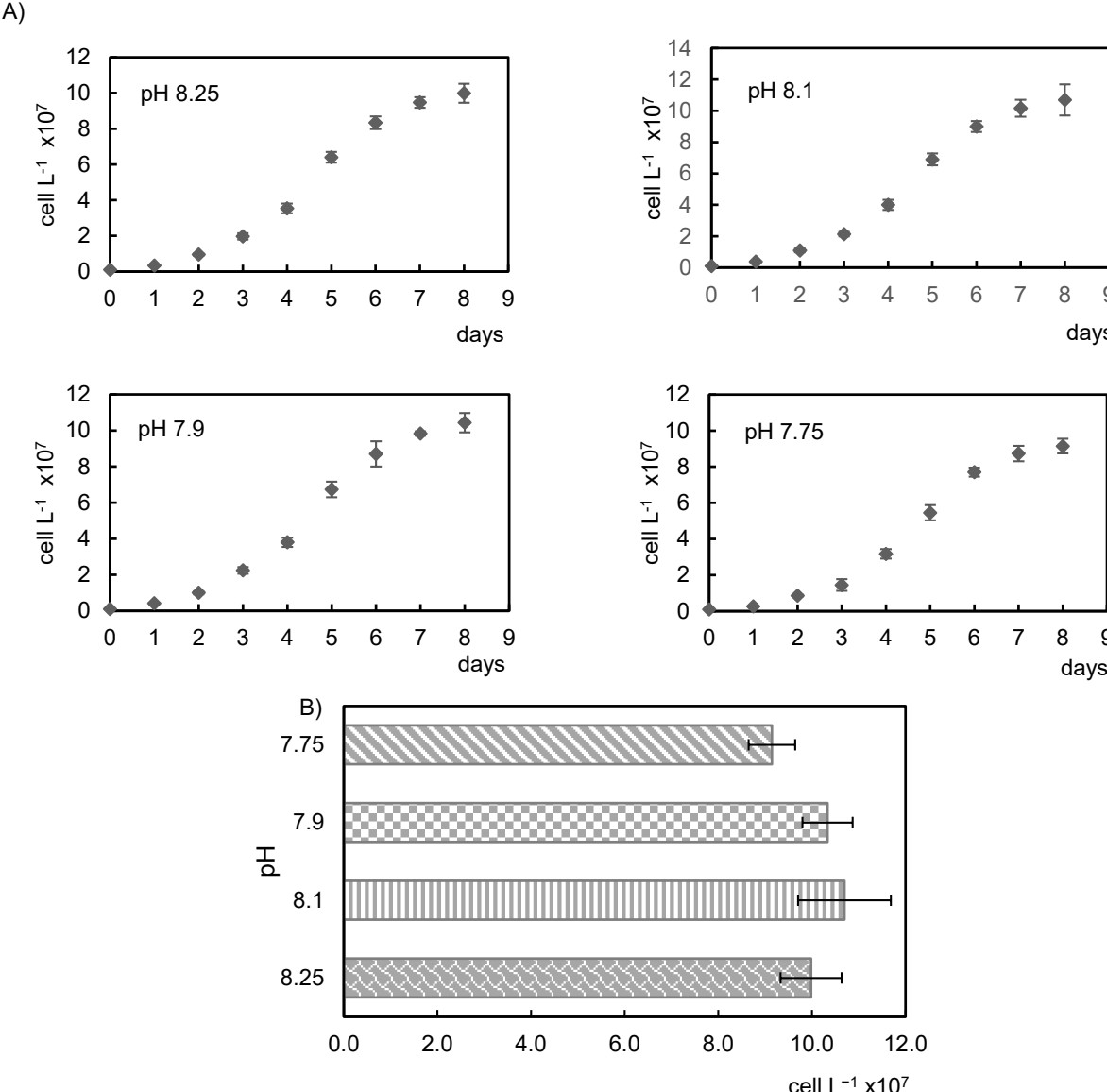

**Figure 1** Growth curves (A) and cell densities (B) of coccolithophore *Emiliania huxleyi* cultivated under four different pH conditions.

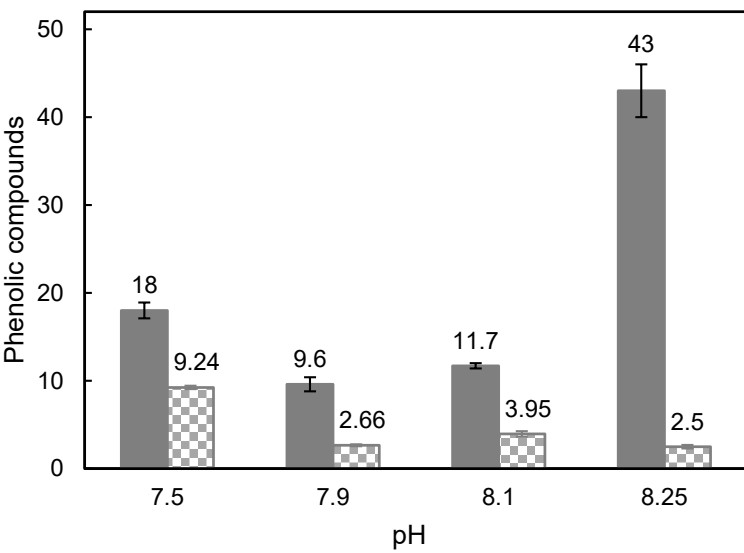

**Figure 2** Total sum of the identified intracellular polyphenols expressed as attomol cell$^{-1}$ (non-solid bars) and concentration (nM) of exuded polyphenols (solid bars) by *Emiliania huxleyi* cells grown under reduced pH conditions after eight culture days.
