# Peer review of "Variations of polyphenols and carbohydrates of *Emiliania huxleyi* grown under simulated ocean acidification conditions"

_Biogeosciences, 2024_

## Author Comment (AC1)

**'Point-by-point response to reviewers' file**

We really thank and appreciate important comments and suggestions made by Reviewers. Substantial changes have been made in the manuscript in accordance with their suggestions and have been highlighted in blue.

Reviewer's comments followed by our reply (blue color).

https://doi.org/10.5194/bg-2024-1-RC2

**Anonymous Referee #2, 25 Apr 2024**

The manuscript by Rico et al., demonstrated the responses of phenolics and carbohydrates of Emiliania huxleyi to ocean acidification conditions. They found that the intracellular phenolic compounds increased under low pH conditions, and these findings have certain implications for the marine food webs and biogeochemical cycles. However, I do have some concerns need to be addressed before it can be accepted for the publication in BG.

This study complements the one previously carried out by our research group and published by Samperio et al. (2017), who cultured the cells studied here (5 replicates) and monitored the changes in several parameters throughout different growth phases (growth, DOC, etc.). Some details of the growing conditions have been included in section 2.2 following suggestions from another reviewer.

- Samperio-Ramos, G., Santana-Casiano, J. M., González-Dávila, M., Ferreira, S., and Coimbra, M. A.: Variability in the organic ligands released by Emiliania huxleyi under simulated ocean acidification conditions, AIMS Environ. Sci., 4(6), https://doi.org/10.3934/environsci.2017.6.788, 2017.

New sections have been included and are listed at the end of this document:

- **2.4 Chlorophyll *a***
- **2.8 Statistical analysis**
- **3.1 Carbonate chemistry parameters**
- **3.3 Chlorophyl a**

Some references have been deleted or replaced by others that are more in line with the amended discussion.

General Comments:

**Abstract:** The Abstract was not well structed, and is wordy. It should be very concise and highlight the significance of the findings presented in this study. Please rephrase it to be more concise.

The Abstract has been modified as follows:

- **Abstract.** Cultures of the coccolithophore *Emiliania huxleyi* were grown under four different $CO_2$-controlled pH conditions (7.75, 7.90, 8.10, and 8.25) to improve understanding of its responses to ocean acidification scenarios. Acidification did not significantly affect final cell densities and carbohydrate contents. Intra- and extracellular phenolic compounds were identified and quantified by Reverse Phase-High Performance Liquid Chromatography (RP-HPLC), with the highest concentrations of total exuded phenolics at pH 8.25 (43±3 nM) and 7.75 (18.0±0.9 nM). Accumulation of intracellular phenolic compounds was observed in cells with decreasing pH, reaching the maximum level (9.24±0.19 attomole cell$^{-1}$) at the lowest pH (7.75). The phenolic profiles presented significant changes in exuded epicatechin and protocatechuic acid ($p<0.05$ and $0.01$, respectively), and intracellular vanillic acid ($p<0.001$), which play an essential role as antioxidants, and in the availability of trace metals. A significant increase in chlorophyll *a* content was observed in cells grown at the most acidic pH ($p<0.01$), which also showed significantly higher radical inhibition activity ($p<0.01$). However, no significant differences were found between the iron reducing activities and the radical scavenging activities of the compounds present in the exudates. The nature and concentration of the organic compounds present in the culture medium may favour or inhibit the local growth of specific algal species, and influence trace metal bioavailability affecting the biogeochemical cycling of carbon and affecting microbial functional diversity.

Discussion:

1: The discussion could better articulate how the observed changes in polyphenols and carbohydrates could impact broader ecological processes. Specifically, the implications for the marine food web and biogeochemical cycles could be explored in more depth.

Several sentences have been included to highlight the implications for the marine food web in more depth:

- Introduction (lines 46 to 63 discuss this topic): Changes in environmental pH could have consequences on the competitiveness of both sensitive and tolerant microalgae in mixed phytoplankton communities, modifying their structure, composition, and distribution, which are crucial in mitigating global environmental change by fixing and transporting carbon from the upper to the deep ocean in the major global carbon sink (Eltanahy and Torky, 2021; Kholssi et al., 2023, Marinov et al., 2010). Vasconcelos et al. (2002) found that exudates from *Phaeodactylumn tricornutum* (*P. tricornutum*) diatoms caused a toxic effect on *E. huxleyi*, while those from *Enteromorpha* spp. caused the enhancement of final cell yield, concluding that specific exudates produced by the bloom of one algal species may favour or inhibit the local growth of other species. Such changes could also affect the species at a higher trophic level, resulting in a potential shift in biodiversity (Jin and Kirk, 2018). Spisla et al. (2021) reported that extreme $CO_2$ events modify the composition of particulate organic matter, which leads to a substantial reorganization of the planktonic community, affecting multiple trophic levels from phytoplankton to primary and secondary consumers (Trombetta et al., 2019). Nelson et al. (2020) found modifications of planktonic and benthic communities in response to reduced seawater pH (from pH 8.1 to 7.8 and 7.4), concluding that a re-arrangement of the biofilm microbial communities occurred through a potential shift from autotrophic to heterotrophic dominated biofilms. In addition, microbial biofilms obtained under reduced pH altered settlement rates in invertebrate larvae of *Galeolaria hystrix*. Barcelos e Ramos et al. (2022) showed that coexistence with other microorganisms modifies the response of *E. huxleyi* to increased $CO_2$, markedly decreasing its growth rate at elevated $CO_2$ concentrations with bacteria *Idiomarina abyssalis* (*I. abyssalis*) and *Brachybacterium* sp. Moreover, elevated $CO_2$ concentrations increased organic carbon and decreased inorganic carbon content of *E. huxleyi* cells in the presence of *I. abyssalis*, but not *Brachybacterium* sp.

[revised manuscript text omitted]

- Sections 3.5: …. These results agree partially with those reported by Jia et al. (2024), who found that lowering seawater pH promoted protein synthesis in microalgae *Chlorella* sp., *P. tricornutum*, and *C. muelleri* grown at pH 7.8 and 7.5, compared to those harvested at pH 8.1, concluding that acidification improves the efficiency of carbon assimilation and provides more carbon skeletons for amino acids and protein synthesis.

2: The manuscript could benefit from a more detailed comparison with existing literature. Specifically, how do these results align or contrast with the findings of similar studies in terms of the magnitude and direction of changes in polyphenol and carbohydrate content?

We have included the following sentences where results of exposure to acidification of other microalgae species are discussed.

- Sections 2.4 and 3.3 with Chl *a* analysis and comparison with other species has also been included.

**2.4 Chlorophyll a**

[revised manuscript text omitted]

Regarding the magnitudes, section 3.3 cites the article by Lopez et al. (2015) in which the concentration of polyphenols is in the same range (between 9.4 and 8.4 nM). The contents in the cells are in the same range than here (attomole):

- These results partially agree with those reported by López et al. (2015) for $Dunaliella$ $tertiolecta$ growing under stress conditions induced by high levels of copper, where the concentration of phenolic compounds declined from 9.4±0.6 nM in seawater cultures without Cu addition to 8.4±0.4 and 8.6±0.4 nM in the copper enriched seawater, and increased 1.4 times concerning the control into the cells grown under the highest Cu level.

3: Broader Impacts: The results are discussed primarily in the context of $Emiliania$ $huxleyi$. Expanding the discussion to consider potential impacts on other phytoplankton species, could provide a more holistic view of ocean acidification impacts.

In addition to all the comparisons and references already included in the manuscript, the following sentences have been added

- Section 3.2: Bautista-Chamizo et al. (2019) exposed microalgae *Tetraselmis chuii* (*T. Chuii*), *Nannochloropsis gaditana* (*N. gaditana*) and *P. tricornutum* to pH 6.0, 7.4 and pH 8.0 as a control, observing growth inhibition of 12%, 61% and 66% at pH 6.0, respectively, in single toxicity tests, with *T. chuii* being the most resistant species to $CO_2$ enrichment. At pH 7.4 only *P. tricornutum* showed a significant decrease in cell abundance (16%), while no differences were found for *T. chuii* and *N. gaditana*, demonstrating that sensitivity to acidification depends on the microalgae species. However, the growth rate of all three microalgae *Chlorella* sp., *P. tricornutum* and *Chaetocetos muelleri* (*C. muelleri*) harvested at pH levels (8.1, 7.8, and 7.5) was significantly enhanced (Jia et al., 2024). The differences observed in the growth behavior of microalgae, mainly of *P. tricornutun* diatoms, in these two last studies could be due to the culture conditions, e.g. in the first study, Bautista-Chamizo et al. (2019) used seawater collected in the Bay of Cádiz (Spain) (with initial pH 8.0, salinity of 34, temperature at 23 °C and initial cell density $3 \times 10^4$ cells $mL^{-1}$), and in the second study Jai et al. (2024) used artificial seawater prepared by mixing sea salt (Haizhixun, China) with purified water (salinity of $30 \pm 1‰$, temperature maintained at $18 \pm 0.5$ °C, initial pH at 8.1, no data on initial cell density).

- Section 3.3 focuses on chlorophyll analysis and the results are compared with those obtained in other acidification studies with *E. huxleyi* and other microalgae species:

[revised manuscript text omitted]

Specific comments:

1: Line 37-39: Some controversial findings regarding the calcification of coccolithophore to ocean acidification should be mentioned.

We agree with the reviewer that a study of the controversial findings regarding the calcification would be interesting, but that part is beyond the scope of our research, which focuses more on the change in both polyphenols and carbohydrates with pH variation. However, as indicated in the previous paragraph we have included same references to this effect.

2: Line 82-84: There were at least two studies examined the responses of phenolic compounds in marine primary producers to ocean acidification (Arnold et al., 2012, PLOS ONE; Jin et al., 2015, Nature Communications, 6:8714), where are more relevant to the present study should be acknowledged here.

The authors agree and therefore cite the article twice in the manuscript, and completed the information as follows:

- Line 214:
- The contents of phenolic compounds inside the cells increased with the decline of pH up to pH 7.75, reaching the maximum level (9.24±0.19 amol cell⁻¹). These results agree with those reported by Jin et al. (2015), who evidenced that phytoplankton grown under the $CO_2$ levels predicted for the end of this century showed accumulation of phenolic compounds, increased by 46–212% compared with that obtained at the current $CO_2$ level. Subsequently, zooplankton fed with phytoplankton grown in acidified seawater showed 28% to 48% higher phenolic content. This transfer of accumulated phenolic compounds to higher trophic levels could have serious consequences for the marine ecosystem and seafood quality. Dupont et al. (2014) have shown that survival of adult boreal shrimp (*Pandalus borealis*) was affected when exposed for 3 weeks at pH 7.5 with an increase in mortality of 63% compared to those grown at pH 8.0 (at the sampling site), showing also changes in appearance and taste.

- Section 3.6, in the last paragraph:

  Several studies evidenced enhanced oxidative stress in cells in acidified seawater, through an increase in ROS that correlated significantly with the accumulation of polyphenols (Bautista-Chamizo et al., 2019; Vázquez et al., 2022). Ocean acidification will directly affect marine organisms, altering the structure and functions of ecosystems. These changes may cause cascading effects in the marine food web, varying the macromolecular composition of consumers (Jia et al., 2024). The accumulation of phenolic compounds leads to functional consequences in primary and secondary producers, with the possibility that fishery industries could be influenced as a result of progressive ocean change (Gattuso et al., 2015; Jin et al., 2015; Trombetta et al., 2019).

By other hand, Arnold et al. (2012) study is focused on marine plants. The authors consider it important to mention this effect seen in higher plants, but not to focus our discussion on them, but rather on microalgae.

 However, Arnold et al. (2012) reported a loss of phenolics in the seagrasses *Cymodocea nodosa, Ruppia maritima,* and *Potamogeton perfoliatus* grown in acidified seawater, where the pH decreased up to 7.3, and the $CO_2$ level increased ten-fold.

3: Line 115-125: Have you monitored the carbonate chemistry parameters over the 8 days experimental duration? These parameters are important for a typic OA research. Please clarify.

All these parameters were monitored, discussed and reported previously by Samperio et al. (2017). We have included the following paragraph in section 2.2 and a new section called 3.1 Carbonate chemistry parameters

- Section 2.2:

- Carbonate chemistry was monitored continuously in the experimental media and determined from pH, total alkalinity (TA), and total dissolved inorganic carbon (TC). pH in the treatments was measured on the free hydrogen ion scale (pH = −log [H⁺]), by immersing Orion Ross combination glass electrodes in the experimental media. The electrodes were calibrated daily using TRIS buffer solutions. The equilibration of the gas in the media of each treatment was achieved after a maximum of 24 h, observed by the evolution of pH. TA and TC were measured at the beginning and end of the experiment on days 0 and 8. TA was determined by potentiometric titration with hydrochloric acid until the endpoint of carbonic acid was reached utilizing a VINDTA 3C system (Mintrop et al., 2000). TC was analyzed by coulometric procedure after phosphoric acid addition (González-Dávila et al. 2011). Certified Reference Material (provided by A. Dickson at Scripps Institution of Oceanography) was employed to assess the performance of the titration system, yielding an accuracy of 1.5 and 1.0 µmol kg⁻¹ for TA and TC, respectively. The Seawater Carbonate package (Seacarb version 3.0), developed for R Studio software (R Development Core Team), was employed to calculate the values of pCO₂, considering the carbonic acid dissociation constants. A more detailed description is given by Samperio-Ramos et al. (2017).

- **3.1 Carbonate chemistry parameters**

- Preliminary tests indicated good stability of carbonate chemistry parameters in media, obtained by the $CO_2$ regulation system (Table 1). After $CO_2$-equilibration, initial TC values ranged from 1905 ± 26 to 2215 ± 16 at a TA mean value of 2386 ± 16 µmol kg⁻¹ (one-way ANOVA; F=1.729, p=0.2382 among treatments), corresponding to a $CO_2$ range of 225 ± 1 and 914 ± 13 µatm. Although carbonate chemistry in $CO_2$-manipulated experiments can be strongly affected by biological activity (Howes et al. 2017; Miller and Kelley 2021), during our experiments TA and TC remained fairly stable (t-tests; p>0.05) within treatments, over the 8-day experimental period.

**Table 1.** Carbonate chemistry parameters in experimental media for each pH treatment at day 0 and day 8: total alkalinity (TA), total dissolved inorganic carbon concentration (TC)) and estimated pCO₂ (µatm). Means and standard deviations were calculated from sampling (n = 3).

| pH-Treatments | TA (µmol kg⁻¹) | | TC (µmol kg⁻¹) | | *p*CO₂ (µatm) | |
|---|---|---|---|---|---|---|
| | Day 0 | Day 8 | Day 0 | Day 8 | Day 0 | Day 8 |
| 8.25 | 2376 ± 12 | 2335 ± 25 | 1905 ± 26 | 1869 ± 61 | 225 ± 1 | 221 ± 4 |
| 8.10 | 2380 ± 15 | 2329 ± 28 | 2012 ± 28 | 1971 ± 44 | 353 ± 2 | 349 ± 5 |
| 7.90 | 2390 ± 17 | 2347 ± 40 | 2129 ± 47 | 2085 ± 36 | 616 ± 12 | 599 ± 8 |
| 7.75 | 2401±14 | 2365 ± 26 | 2215 ± 16 | 2178 ± 39 | 914 ± 18 | 925 ± 27 |

In addition, the following paragraphs have been included in the sections 3.4 and 3.5

- Samperio et al. (2017) reported an increase in dissolved organic carbon exudation by 19% and 15% during exponential and stationary phases, respectively, as $CO_2$ levels increased from 225 µatm (177.06 ± 10.95 fmol C cell⁻¹ day⁻¹) to 900 µatm (209.74 ± 50.00 fmol C day⁻¹ cell⁻¹) in the culture medium. The authors suggested that ocean acidification could significantly enhance the release of phenolic compounds when *E. huxleyi is* grown

under low-iron conditions. They detected phenolic compounds only in the stationary phase and their release rate was affected by $CO_2$ conditions, observing a strong correlation between the concentrations of produced phenolic compounds with exuded dissolved organic carbon, indicating that these compounds constituted a relatively constant fraction of the organic matter excreted by *E. huxleyi*. Extracellular release of phenolic compounds was statistically higher at $p$CO$_2$ 900 µatm ($0.41 \pm 0.02$ fmol C cell$^{-1}$ day$^{-1}$) than at pCO$_2$ 350µatm ($0.36 \pm 0.02$ fmol C cell$^{-1}$day$^{-1}$; Tukey contrast: t value $= 2.495$; $p < 0.1$). While Samperio et al. (2017) studied the total polyphenol contents through the Arnow spectrophotometric assay, in our study we have identified some polyphenols (Table 1), concluding that the highest concentrations of these polyphenols identified were exuded in the cultures at pH 8.25 ($43\pm3$ nM) and 7.75 ($18.0\pm0.9$ nM). The high level of these exuded polyphenols at pH 8.25 could be due to the redox chemistry of inorganic Fe, intimately linked to the pH (Pérez-Almeida et al., 2019). When the pH increases the solubility of inorganic Fe(III) decreases and the oxidation rate constants of Fe(II) increases. Wu et al. (2016) studied the interaction between Fe and ten phenols at pH = 8.0, finding that only caffeic acid, gallic acid, and protocatechuic acid protected 69%, 64% and 33% of the initial iron (II), respectively, due to the chelating capacity of the catechol and galloyl groups with Fe(II).

- Samperio et al. (2017) reported higher contributions of dissolved carbohydrates to excreted DOC during the exponential phase (18–37%) than during the stationary phase (14–23%), significantly increased as time elapsed from the exponential to the stationary phase. However, acidification of the culture medium with $CO_2$ did not affect the levels of carbohydrates exuded per cell in any of the three growth phases, as these levels did not change significantly in any of them as the pH dropped to pH 7.75. The amount of total intracellular carbohydrates also remained constant between pH 8.25 and pH 7.75. The coccolithophore *E. huxleyi* have shown diverse metabolic responses to ocean acidification and to combinations of ocean acidification with other environmental factors with significant differences between strains (Langer et al., 2009; Tong et al. 2017; Gafar et al. 2019).

4: Line 170-175: Should it be possible to calculate the specific growth rates for the exponential phase of each growth curve and then compare them under different pH conditions?

No significant differences were found between the specific growth rates of five replicates:

| pH | Specific Growth Rate (day^-1) |
|---|---|
| 7.75 | 0.564±0.009 |
| 7.9 | 0.582±0.014 |
| 8.1 | 0.587±0.014 |
| 8.25 | 0.575±0.011 |
| ANOVA ($p=0.336$) | |

5: Line 210: diatoms?

This has been corrected

10: Line 255: Typo of the citation "Santana_Casiano et al., 2014"

This has been corrected to Santana-Casiano et al., 2014

A Pearson's correlation test was performed to determine the degree of relationship between pairs of variables and a one-way ANOVA to determine statistically significant differences between measurements. Both studies were conducted using the Jamovi program (2022) and p-values of <0.05 were considered statistically significant.

**Correlation Matrix between pH 8.25 – 7.75**

| | | Intracellular TCH | Extracellular TCH | Intra TPC | Extra TPC | RSA cells | RSA exudates | FRAP cells |
|---|---|---|---|---|---|---|---|---|
| Intracellular TCH | R de Pearson | — | | | | | | |
| | valor p | — | | | | | | |
| Extracellular TCH | R de Pearson | -0.257 | — | | | | | |
| | valor p | 0.731 | — | | | | | |
| Intra TPC | R de Pearson | -0.615 | 0.416 | — | | | | |
| | valor p | 0.948 | 0.153 | — | | | | |
| Extra TPC | R de Pearson | 0.396 | -0.810 | -0.073 | — | | | |
| | valor p | 0.166 | 0.993 | 0.568 | — | | | |
| RSA cells | R de Pearson | -0.280 | 0.728 * | 0.829 ** | -0.240 | — | | |
| | valor p | 0.749 | 0.020 | 0.005 | 0.716 | — | | |
| RSA exuded | R de Pearson | 0.213 | 0.456 | -0.567 | -0.730 | -0.249 | — | |
| | valor p | 0.306 | 0.128 | 0.929 | 0.980 | 0.724 | — | |
| FRAP | R de Pearson | -0.543 | 0.129 | 0.718 * | 0.133 | 0.527 | -0.565 | — |
| | valor p | 0.918 | 0.380 | 0.022 | 0.376 | 0.090 | 0.928 | — |

* $p < .05$, ** $p < .01$, *** $p < .001$

TCH: Total carbohydrates
TFC Total phenolic content
RSA: Radical Scavenging Activity
FRAP: Ferric Reducing Power Assay

**Correlation Matrix between pH 8.1 and 7.75**

| | | Intracellular TCH | Extracellular TCH | Intra TPC | Extra TPC | RSA cells | RSA exuded | FRAP |
|---|---|---|---|---|---|---|---|---|
| Intracellular TCH | R of Pearson | — | | | | | | |
| | p value | — | | | | | | |
| Extracellular TCH | R of Pearson | 0.463 | — | | | | | |
| | p value | 0.177 | — | | | | | |
| Intra TPC | R of Pearson | -0.567 | 0.307 | — | | | | |
| | p value | 0.879 | 0.277 | — | | | | |
| Extra TPC | R of Pearson | -0.662 | 0.104 | 0.965 *** | — | | | |
| | p value | 0.924 | 0.423 | < .001 | — | | | |
| RSA cells | R of Pearson | -0.083 | 0.803 * | 0.810 * | 0.665 | — | | |
| | p value | 0.562 | 0.027 | 0.025 | 0.075 | — | | |
| RSA exuded | R of Pearson | 0.736 * | -0.043 | -0.941 | -0.976 | -0.627 | — | |
| | p value | 0.048 | 0.532 | 0.997 | 1.000 | 0.909 | — | |
| FRAP | R of Pearson | -0.625 | 0.201 | 0.778 * | 0.786 * | 0.588 | -0.709 | — |
| | p value | 0.908 | 0.351 | 0.034 | 0.032 | 0.110 | 0.943 | — |

Nota. * $p < .05$, ** $p < .01$, *** $p < .001$

TCH: Total carbohydrates
TPC: Total phenolic content
RSA: Radical Scavenging Activity
FRAP: Ferric Reducing Power Assay

**One way factor ANOVA**

| | F | gl1 | gl2 | P | p |
|---|---|---|---|---|---|
| RSA cells | 290.89 | 3 | 2.07 | 0.003 | < .01 |
| RSA exudates | 30.38 | 3 | 1.90 | 0.037 | < .05 |
| FRAP cells | 1.26 | 3 | 2.12 | 0.464 | |
| Total exuded phenolics (nM) | 64.69 | 3 | 1.88 | 0.019 | < .05 |
| Exuded phenolics per cell | 14529.66 | 3 | 1.67 | < .001 | |
| Protocatechuic acid | 256.02 | 3 | 2.00 | 0.004 | < .01 |
| Epicatechin | 40.88 | 3 | 1.94 | 0.026 | < .05 |
| Rutin | 7.25 | 3 | 1.70 | 0.152 | |
| Intracellular-phenolics | 173.30 | 3 | 2.18 | 0.004 | < .01 |
| Intra-Vanillic acid | 1308.13 | 3 | 2.08 | < .001 | |

RSA: Radical Scavenging Activity
FRAP: Ferric Reducing Power Assay
TCH: Total carbohydrates
TPC: Total phenolic content

---

## Author Comment (AC2)

Las Palmas de Gran Canaria, June 3, 2024

**'Point-by-point response to reviewers' file**

**https://doi.org/10.5194/bg-2024-1-RC1**

We really thank and appreciate important comments and suggestions made by Reviewers. Substantial changes have been included in the manuscript in accordance with their suggestions and have been highlighted in blue. Some references have been deleted or replaced by others that are more in line with the amended discussion.

Reviewer's comments followed by our highlighted reply (blue color).

**Anonymous Referee #1, 30 Mar 2024**

The manuscript by M. Rico et al presents data from a cultivation of *Emiliania h*. under four pH conditions. results concern the growth, phenolic compounds and total carbohydrate in both cells and medium.

The study might be of interest if the authors add some other variables (pigments, coccoliths,). in its present state, this ms can not be accepted as publication, due to relevant weaknesses.

New sections have been included:

- Section 2.4 Chlorophyll *a*: Chlorophyll *a* (Chl *a*) was determined according to Branisa et al. (2014) with modifications. Frozen cells were suspended in 5 mL of acetone:hexane (3:2) and sonicated for 3 min in an ice water bath. Homogenates were centrifuged at 7500 rpm for 5 minutes and the absorbance (A) of supernatants was measured at 645 and 663 nm. Chl *a* was expressed as femtogram cell$^{-1}$ and quantified spectrophotometrically according to equation: Chl *a* (mg/100 mL) = 0.999×A$_{663}$–0,0989×A$_{645}$.

- Section 3.3 Chlorophyl *a*: After 8 culture days, the concentration of Chl *a* per cell decreases with decreasing pH from 56.6±2.8 fmol cell$^{-1}$ (pH 8.25) to 26.8±1.4 fmol cell$^{-1}$ (pH 7.9). However, cells grown in the most acidic conditions (pH 7.75) show the highest amount of Chl *a* (67.3±2 fmol cell$^{-1}$) with a significant increase observed between pH 8.1 (45.1±3 fmol cell$^{-1}$) and 7.75 ($p<0.01$). These results agree with those reported by Vázquez et al. (2022), who observed significantly increased cellular concentration of Chl *a* in the high $pCO_2$ treatment of *E. huxleyi* cells (1200 μatm, pH 7.62) with respect to control (400 μatm, pH 8.10). Crawford et al. (2011) also studied the effect of elevated $CO_2$ on cultures of the diatom *Thalassiosira pseudonana* CCMP1335 in a pH range between 7.8 and 8.1 (760 μatm and 380 μatm, respectively), concluding that chlorophyll content increased in cells grown at pH 7.8 compared to those grown at pH 8.1, but photosynthetic efficiency remained constant in both experiments. Higher level of Chl *a* were also observed in microalgae *Chlorella* sp., *P. tricornutum,* and *C. muelleri* harvested at pH 7.8 and 7.5 than that quantified at pH 8.1, concluding that primary producers that can utilize $HCO_3^-$ as the carbon source benefit from elevated $CO_2$ concentration (Jia et al., 2024). However, Yu et

al. (2022) found no significant changes in cellular and total content of Chl *a* of *E. huxleyi* cells harvested at similar pH than those here (between 7.72-7.76 under elevated $CO_2$ (1000 μatm) and between 8.03 and 8.07 under atmospheric $CO_2$). Mackey et al. (2015) concluded that photosynthetic responses to ocean acidification are highly variable throughout species and taxa, with different *E. huxleyi* strains exhibiting opposite responses to elevated $CO_2$ on maximum photosynthetic rates.

- The ms totally lacks of statistical analysis. It does seem that for most of the data, no significant variation was revealed among the different treatments. if this was true, the results and discussion section has to be thoroughly re-written.

This study complements the one previously carried out by our research group and published by Samperio et al. (2017), who cultured the cells studied here (5 replicates) and monitored the changes in several parameters throughout different growth phases (growth, DOC, etc.). Some details of the growing conditions have been included in section 2.2 following suggestions from another reviewer.

- Samperio-Ramos, G., Santana-Casiano, J. M., González-Dávila, M., Ferreira, S., and Coimbra, M. A.: Variability in the organic ligands released by Emiliania huxleyi under simulated ocean acidification conditions, AIMS Environ. Sci., 4(6), https://doi.org/10.3934/environsci.2017.6.788, 2017.

The cell densities (N) were determined from the average of 5 experimental batch cultures at each $pCO_2$ treatment. The goodness of the fit for each curve was estimated by the coefficient of correlation (r2 > 0.95).

The following new sections have been included in the manuscript. ANOVA studies and correlation matrix are included at the end of this document.

- **2.8 Statistical analysis.** A Pearson's correlation test was performed to determine the degree of relationship between pairs of variables and a one-way ANOVA to determine statistically significant differences between measurements. Both studies were conducted using the Jamovi program (2022) and *p*-values of <0.05 were considered statistically significant.
- **3.1 Carbonate chemistry parameters**

- the fig. 2 is emblematic:  in the legend, "diatoms" are mentioned while results cam from Emiliania h.; dark vs light color is referred to what?; these data correspond to which day ?

The title of Figure 2 has been changed as follows:

- Figure 2. Total sum of identified intracellular polyphenols expressed as attomole cell[-1] (light coloured bars) and concentration (nM) of exuded polyphenols (dark coloured bars) by *E. huxleyi* cells grown under reduced pH conditions after eight days of culture.

- fig.1: no SD bars

- The SD bars overlap in the growth curves and were therefore avoided. Figure 1.B) is related and shows the cell densities with the SDs.

[Figure]

However, Figure 1 has been changed as presented below.

A)

B)

- material and methods: no mention on light, photoperiod, temperature...: these factors are probably the most relevant to shape the microalgal physiology.

- The following sentences with the requested information have been included in the text:
- Axenic cultures of *E. huxleyi* (strain RCC1238) were supplied by the Spanish Bank of Algae (BEA) in f/2 medium. *E. huxleyi* coccolithophore was cultured with an initial cell density of $10^6$ cells $L^{-1}$ at a constant temperature of 25 ºC, under complete photoperiod (24 h) with light intensity of 200 µmol photons $m^{-2}$ and under different $p$CO$_2$-controlled seawater pH conditions (7.75, 7.90, 8.10, and 8.25), measured on the free hydrogen ion scale pH$_F$=-log[H+] with a Ross Combination glass body electrode calibrated daily with TRIS buffer solutions. For this purpose, a gaseous mixture of CO$_2$-free air and pure CO$_2$ (up to CO$_2$ levels 900, 600, 350, and 225 µatm, respectively) was bubbled in the culture medium (sterile filtered (0.1 µm) North Atlantic seawater (S = 36.48) obtained at the ESTOC site (29º10' N, 15º30' W) with an equipment that modulates the CO$_2$ flow once the desired pH is reached, keeping it constant (±0.02). Cells were frozen and stored at -80ºC.
- Carbonate chemistry was monitored continuously in the experimental media and determined from pH, total alkalinity (TA), and total dissolved inorganic carbon (DIC). pH in the treatments was measured on the free hydrogen ion scale (pH = −log [H$^+$]), by immersing Orion Ross combination glass electrodes in the experimental media. The electrodes were calibrated daily using TRIS buffer solutions. The equilibration of the gas in the media of each treatment was achieved after a maximum of 24 h, observed by the evolution of pH. TA and DIC were determined using a VINDTA 3C system (González-Dávila et al., 2011). TA was determined by potentiometric titration with hydrochloric acid until the endpoint of carbonic acid was reached. DIC was analyzed by coulometric procedure after phosphoric acid addition. Certified Reference Material (provided by A. Dickson at Scripps Institution of Oceanography) was employed to assess the performance of the titration system, yielding an accuracy of 1.5 and 1.0 µmol kg$^{-1}$ for TA and DIC, respectively. The Seawater Carbonate package (Seacarb version 3.0), developed for R Studio software (R Development Core Team), was employed to calculate the values of $p$CO$_2$, considering the carbonic acid dissociation constants. A more detailed description is given by Samperio-Ramos et al. (2017).

- material and methods: it is not clear: seawater or medium?

- Experimental cultures were grown in sterile filtered (0.1 µm) North Atlantic seawater (S = 36.48) obtained at the ESTOC site (29º10' N, 15º30' W).
- The following sentences have been included in the text:

Axenic cultures of *E. huxleyi* (strain RCC1238) were supplied by the Spanish Bank of Algae (BEA) in f/2 medium. *E. huxleyi* coccolithophore was cultured with an initial cell density of $10^6$ cells $L^{-1}$ at a constant temperature of 25 ºC, under complete photoperiod (24 h) with light intensity of 200 µmol photons $m^{-2}$ and under different $p$CO$_2$-controlled seawater pH conditions (7.75, 7.90, 8.10, and 8.25), measured on the free hydrogen ion scale pH$_F$=-log[H+] with a Ross Combination glass body electrode calibrated daily with TRIS buffer solutions. For this purpose, a gaseous mixture of CO$_2$-free air and pure CO$_2$ (up to CO$_2$ levels 900, 600, 350, and 225 µatm, respectively) was bubbled in the culture medium (sterile filtered (0.1 µm) North Atlantic seawater (S = 36.48) obtained at the ESTOC site (29º10' N, 15º30' W) with an equipment that modulates the CO$_2$ flow once the desired pH is reached, keeping it constant (±0.02).

- which was the target of pH manipulation?

- The main aim of this work is stated in the first sentence of the last paragraph of the introduction:

This work aimed to determine how marine acidification may affect the composition of cells and exudates from *Emiliania huxleyi*.

- **Abstract:** Cultures of the coccolithophore *Emiliania huxleyi* were grown under four different $CO_2$-controlled pH conditions (7.75, 7.90, 8.10, and 8.25) to improve understanding of its responses to ocean acidification scenarios.

- Changes in pH generate stress conditions, either because at high pH drastically decrease the availability of trace metals such as Fe(II), a restrictive element for primary productivity (Wu et al., 2016) or because ROS are increased at acid pH (Bautista-Chamizo et al., 2019; Vázquez et al., 2022). The characterization of compounds exuded into the environment under stress conditions has allowed the development of important new lines of research in iron chemistry, for example, in different acidification scenarios. These compounds are crucial ligands in the formation of metal complexes to acquire micronutrients, sequester toxic metals, and to establish electrochemical gradients resulting in changes in the speciation, the bioavailability, and the toxicity of trace metals. Our research team has tested several compounds identified in the exudates of microalgae to study the effects on copper and iron chemistry in seawater:

  - Pérez-Almeida et al. (2022): Ocean Acidification Effect on the Iron-Gallic Acid Redox Interaction in Seawater, Front. Mar. Sci., Sec. Marine Biogeochemistry, 9. https://doi.org/10.3389/fmars.2022.837363
  - Arnone et al. (2024): Distribution of copper-binding ligands in Fram Strait and influences from the Greenland Shelf (GEOTRACES GN05), Science of The Total Environment, 909, 168162, https://doi.org/10.1016/j.scitotenv.2023.168162
  - González et al. (2018): Iron complexation by phenolic ligands in seawater, Chem. Geol. 511, 380-388, https://doi.org/10.1016/j.chemgeo.2018.10.017
  - López, et al. (2015): Phenolic profile of Dunaliella tertiolecta growing under high levels of copper and iron, Environ. Sci. Pollut. Res. 22 (19) 14820-14828, 10.1007/s11356-015-4717-y
  - Santana-Casiano et al. (2014): Characterization of phenolic exudates from Phaeodactylum tricornutum and their effects on the chemistry of Fe(II)-Fe(III). Mar. Chem. 158, 10-16. https://doi.org/10.1016/j.marchem.2013.11.001

- material and methods: 48h (line 124)? does it mean that the experiment started after 48 h of cultivation of E.h. under the different conditions? in the fig.1 the day 0 corresponded to this time? which was the cell concentration at this time?

- These two days correspond to the log phase and are included in the 8 days of culture monitoring.
- We distinguish the three stages in the growth curves of *E. huxleyi*: initial or log phase (these two days, until 2nd day = the first 48h mentioned above), exponential (EP, from 3rd to 5th day) and steady (SP, from 6th to 8th day) phases.

- material and methods: please explain why a first extraction in acetone and then in methanol. what is the role of acetone?

This is a mistake. The described procedure was used for chlorophyll measurement with a mixture of acetone and hexane as described in section 2.4. For the extraction of polyphenols, and for the DPPH and FRAP assays, methanol was used following the usual protocol in our laboratory (López et al., 2015; Santiago-Díaz et al., 2023) and that used by Vicente et al. (2021), and for the extraction of carbohydrates, 5 mL of acidified water (pH = 2) was used, 1.5 mL was freeze-dried and the residue was dissolved as described in section 2.3.

- López et al.: Phenolic profile of Dunaliella tertiolecta growing under high levels of copper and iron, Environ. Sci. Pollut. Res. 22, 14820–14828. https://doi.org/10.1007/s11356-015-4717-y, 2015.
- Santiago-Díaz et al.: Copper toxicity leads to accumulation of free amino acids and polyphenols in Phaeodactylum tricornutum diatoms, Environ. Sci. Pollut. Res., 30, 51261–51270, https://doi.org/10.1007/s11356-023-25939-0, 2023.
- Vicente et al.: Production and bioaccessibility of *Emiliania huxleyi* biomass and bioactivity of its aqueous and ethanolic extracts, J. Appl. Phycol. 33, 3719–3729, https://doi.org/10.1007/s10811-021-02551-8, 2021

- quantification of phenols: did the authors used some pure standards?

We used the standards described in the following sections:

- Introduction (lines 100-102)
  Intra- and extracellular phenolic compounds (gallic acid (GAL), protocatechuic acid (PCA), p-coumaric acid (COU), ferulic acid (FA), catechin (CAT), vanillic acid (VAN), epicatechin (ECAT), syringic acid (SYR), rutin (RU) and gentisic acid (GA)) were identified and quantified by RP-HPLC.

- 2.1 Chemicals (lines 110-111)
  Polyphenol standards were supplied as follows: GAL, PCA, COU, FA, CAT, VAN, ECAT, and SYR by Sigma–Aldrich Chemie (Steinheim, Germany); RU and GA by Merck (Darmstadt, Germany

- The description of the chromatographic analysis can be found in section 2.5, where the standards were cited (lines 166-167): For quantification, simultaneous monitoring was set at 270 nm (GAL, PCA, CAT, VAN, RU, ECAT, and SYR) and 324 nm (GA, COU, and FA).

- calibration curve: please explain why there is a "b" factor (y = ax + b; for carbohydrates, frap) also especially when it is negative (dpph)?

**DPPH assay:** We prepare a calibration curve of DPPH in methanol at different concentrations. The absorbance corresponding to each concentration is measured and the following calibration curve is constructed:

[Figure]

The absorbances of the algal samples correspond to the concentration of not inhibited DPPH. Therefore, subtracting the concentrations obtained through the calibration curve from the initial DPPH concentration allows us to calculate the amount of inhibited DPPH.

**Calibration curve for FRAP determination:**

[Figure]

This method has been performed according to the literature, where this factor is always present, probably due to parallel reactions or impurities in the reagents, which are present in all standards and samples, and therefore do not affect the final result. Regardless of whether the reducing power is measured as reduced Fe(III) from a calibration curve prepared with Fe(II) or whether it is measured with a standard such as Trolox, this factor b is present.

**Glucose calibration curve:**

[Figure]

The following articles corroborates this:
- *Noreen et al., 2017: $y=0.000\ 06x+0.1887$. https://doi.org/10.1016/j.apjtm.2017.07.024.*
- *Alam et al. 2014: $y=5.4901x+0.2547$. https://doi.org/10.1155/2014/296063.*
- *Mukherjee et al., 2019. https://link.springer.com/article/10.1007/s12649-017-0053-4*

- line 165: cell extracts? how they have been done?

- this aspect has been clarified above.

- LINE 173: The highest peak? but NO SIGNIFICANT!

- This sentence has been deleted.

- lines 186-189: re-write!

- These lines have been rewritten as follows and placed at the end of section 3.2:

> Discrepancies found in the literature regarding acidification effects and responses of *E. huxleyi* coccolithophores may be due not only to different environmental factors and culture conditions, etc. (Tong et al. 2017; Gafar et al. 2019). Langer et al. (2009) observed substantial differences in sensitivity to acidification in four different *E. huxleyi* strains with different responses in all parameters tested.

- table 1: unit of cell and exuded? seems to be different, also from the "levels" (nM).

- The title of Table 1 has been changed as follows and a mistake was corrected in the amount of PCA exuded at pH 7.75.

**Table 1.** Amounts of intracellular and exuded phenolic compounds by cells of *E. huxleyi* grown under different pH conditions.

| Phenolic compound | pH 7.75 (pCO$_2$ 900µat) | | pH 7.90 (pCO$_2$ 600µat) | | pH 8.10 (pCO$_2$ 350µat) | | pH 8.25 (pCO$_2$ 225µat) | |
|---|---|---|---|---|---|---|---|---|
| | Cell | Exuded | Cell | Exuded | Cell | Exuded | Cell | Exuded |
| GAL[a] | 0.13±0.03 | - | - | - | 0.25±0.07 | - | - | 45±0 |
| PCA[a] | - | 151±12 | 0.16±0.01 | 34±2 | - | 26.3±0.2 | - | 298±22 |
| ECAT[a] | 1.2±0.1 | 34±1 | - | 40±5 | - | 70±2 | - | 77±6 |
| VAN[a] | 6.44±0.06 | - | 2.5±0.1 | - | 2.30±0.05 | - | 2.5±0.2 | - |
| COU[a] | - | - | - | - | 1.4±0.2 | - | - | - |
| RU[a] | 1.47±0 | 12.1±0.4 | | 20±2 | | 13.2±0.3 | | 11.2±0.8 |
| Sum[a] | 9.24±0.19 | 197.1±12.3 | 2.66±0.11 | 94±9 | 3.95±0.32 | 109.5±2.5 | 2.5±0.2 | 431.2±28.8 |
| Concentration (nM)[b] | | 18.0±0.9 | | 9.6±0.8 | | 11.7±0.3 | | 43±3 |

[a]Results are expressed as attomole cell$^{-1}$ (means ± standard deviation of three measurements).
[b]Results are expressed as nanomole L$^{-1}$ (means ± standard deviation of three measurements).
Abbreviations: GAL: gallic acid; PCA: protocatechuic acid; ECAT: epicatechin; VAN: vanillic acid; COU: p-coumaric acid; RU: rutin.

All quantities are expressed as attomole cell$^{-1}$, specified with the superscripts[a], except for the nanomolar concentration, specified with the superscripts[b] and footnotes in the table.

- I suggest also to the authors to give a look on 10.1080/07388551.2021.1874284, that might help for the discussion.

The aim of the study suggested by the reviewer is to provide an overview of current knowledge on phenolic compounds in microalgae. The article focuses on factors that influence the variation in total polyphenol and flavonoid content: microalgal biodiversity, chemodiversity between groups, different analytical methodologies, physiological state based on how the cells are maintained or cultured, e.g. light or other factors. The study cites two of our previously reported manuscripts (references 19 and 23):

- 19. Rico et al.: Variability of the phenolic profile in the diatom *Phaeodactylum tricornutum* growing under copper and iron stress. Limnol Oceanogr. 2013; 58: 144–152.
- 23. López et al.: Phenolic profile of *Dunaliella tertiolecta* growing under high levels of copper and iron. Environ Sci Pollut Res Int. 2015; 22: 14820–14828.

However, pH is the only variable modified in our study focused on the effect of acidification on *E. huxleyi* through $CO_2$ acidification, so changes in organic matter should be linked only to the effect of this pH change and its consequences (changes in the availability of essential metals such as iron). We used the same strains as well as the cultivation conditions (lighting, seawater, nutrients, temperature, etc.) so the influence of all these factors should be the same in all cultures.

- legend: three measurements: three replicates? (different cultures? or technical replicates?)
- Five experimental batch cultures were carried out at each pH treatment. Three replicates refer to different cultures.

- lack of significativity tests in all the studies

The version 2.3 of jamovi program (2022) has been used for statistical analyses (retrieved from https://www.jamovi.org).

A Pearson's correlation test was performed to determine the degree of relationship between pairs of variables and a one-way ANOVA to determine statistically significant differences between measurements. Both studies were conducted using the Jamovi program (2022) and *p*-values of <0.05 were considered statistically significant.

**Correlation Matrix between pH 8.25 – 7.75**

|  |  | Intracellular TCH | Extracellular TCH | Intra TPC | Extra TPC | RSA cells | RSA exudates | FRAP cells |
|---|---|---|---|---|---|---|---|---|
| Intracellular TCH | R de Pearson | — |  |  |  |  |  |  |
|  | valor p | — |  |  |  |  |  |  |
| Extracellular TCH | R de Pearson | -0.257 | — |  |  |  |  |  |
|  | valor p | 0.731 | — |  |  |  |  |  |
| Intra TPC | R de Pearson | -0.615 | 0.416 | — |  |  |  |  |
|  | valor p | 0.948 | 0.153 | — |  |  |  |  |
| Extra TPC | R de Pearson | 0.396 | -0.810 | -0.073 | — |  |  |  |
|  | valor p | 0.166 | 0.993 | 0.568 | — |  |  |  |
| RSA cells | R de Pearson | -0.280 | 0.728 * | 0.829 ** | -0.240 | — |  |  |
|  | valor p | 0.749 | 0.020 | 0.005 | 0.716 | — |  |  |
| RSA exuded | R de Pearson | 0.213 | 0.456 | -0.567 | -0.730 | -0.249 | — |  |
|  | valor p | 0.306 | 0.128 | 0.929 | 0.980 | 0.724 | — |  |
| FRAP | R de Pearson | -0.543 | 0.129 | 0.718 * | 0.133 | 0.527 | -0.565 | — |
|  | valor p | 0.918 | 0.380 | 0.022 | 0.376 | 0.090 | 0.928 | — |

$* p < .05, ** p < .01, *** p < .001$

TCH: Total carbohydrates
TFC Total phenolic content
RSA: Radical Scavenging Activity
FRAP: Ferric Reducing Power Assay

**Correlation Matrix between pH 8.1 and 7.75**

| | | Intracellular TCH | Extracellular TCH | Intra TPC | Extra TPC | RSA cells | RSA exuded | FRAP |
|---|---|---|---|---|---|---|---|---|
| Intracellular TCH | R of Pearson | — | | | | | | |
| | p value | — | | | | | | |
| Extracellular TCH | R of Pearson | 0.463 | — | | | | | |
| | p value | 0.177 | — | | | | | |
| Intra TPC | R of Pearson | -0.567 | 0.307 | — | | | | |
| | p value | 0.879 | 0.277 | — | | | | |
| Extra TPC | R of Pearson | -0.662 | 0.104 | 0.965 *** | — | | | |
| | p value | 0.924 | 0.423 | < .001 | — | | | |
| RSA cells | R of Pearson | -0.083 | 0.803 * | 0.810 * | 0.665 | — | | |
| | p value | 0.562 | 0.027 | 0.025 | 0.075 | — | | |
| RSA exuded | R of Pearson | 0.736 * | -0.043 | -0.941 | -0.976 | -0.627 | — | |
| | p value | 0.048 | 0.532 | 0.997 | 1.000 | 0.909 | — | |
| FRAP | R of Pearson | -0.625 | 0.201 | 0.778 * | 0.786 * | 0.588 | -0.709 | — |
| | p value | 0.908 | 0.351 | 0.034 | 0.032 | 0.110 | 0.943 | — |

Nota. * $p < .05$, ** $p < .01$, *** $p < .001$

TCH: Total carbohydrates
TPC: Total phenolic content
RSA: Radical Scavenging Activity
FRAP: Ferric Reducing Power Assay

**One way factor ANOVA**

| | F | gl1 | gl2 | P | p |
|---|---|---|---|---|---|
| RSA cells | 290.89 | 3 | 2.07 | 0.003 | < .01 |
| RSA exudates | 30.38 | 3 | 1.90 | 0.037 | < .05 |
| FRAP cells | 1.26 | 3 | 2.12 | 0.464 | |
| Total exuded phenolics (nM) | 64.69 | 3 | 1.88 | 0.019 | < .05 |
| Exuded phenolics per cell | 14529.66 | 3 | 1.67 | < .001 | |
| Protocatechuic acid | 256.02 | 3 | 2.00 | 0.004 | < .01 |
| Epicatechin | 40.88 | 3 | 1.94 | 0.026 | < .05 |
| Rutin | 7.25 | 3 | 1.70 | 0.152 | |
| Intracellular-phenolics | 173.30 | 3 | 2.18 | 0.004 | < .01 |
| Intra-Vanillic acid | 1308.13 | 3 | 2.08 | < .001 | |

RSA: Radical Scavenging Activity
FRAP: Ferric Reducing Power Assay
TCH: Total carbohydrates
TPC: Total phenolic content

New sections and paragraphs included and/or changed in the manuscript:

**Abstract.** Cultures of the coccolithophore *Emiliania huxleyi* were grown under four different $CO_2$-controlled pH conditions (7.75, 7.90, 8.10, and 8.25) to improve understanding of its responses to ocean acidification scenarios. Acidification did not significantly affect final cell densities and carbohydrate contents. Intra- and extracellular phenolic compounds were identified and quantified by Reverse Phase-High Performance Liquid Chromatography (RP-HPLC), with the highest concentrations of total exuded phenolics at pH 8.25 (43±3 nM) and 7.75 (18.0±0.9 nM). Accumulation of intracellular phenolic compounds was observed in cells with decreasing pH, reaching the maximum level (9.24±0.19 attomole cell$^{-1}$) at the lowest pH (7.75). The phenolic profiles presented significant changes in exuded epicatechin and protocatechuic acid ($p<0.05$ and 0.01, respectively), and intracellular vanillic acid ($p<0.001$), which play an essential role as antioxidants, and in the availability of trace metals. A significant increase in chlorophyll *a* content was observed in cells grown at the most acidic pH ($p<0.01$), which also showed significantly higher radical inhibition activity ($p<0.01$). However, no significant differences were found between the iron reducing activities and the radical scavenging activities of the compounds present in the exudates. The nature and concentration of the organic compounds present in the culture medium may favour or inhibit the local growth of specific algal species, and influence trace metal bioavailability affecting the biogeochemical cycling of carbon and affecting microbial functional diversity.

**2.3 Cells and exudates extract preparation prior to HPLC, carbohydrates, and antioxidant activities quantifications**

Cells cultured for 8 days in 700 mL of seawater were freeze-dried and suspended in 5 mL of methanol. The samples were sonicated for 10 min, stirred for 5 min, and centrifuged at 7000 rpm for 15 min. The supernatant was separated, and the residue was extracted with 15 mL of methanol for 30 min and centrifuged again. All supernatants were collected and evaporated to dryness, and the resulting residue was dissolved in 5 mL of methanol. These samples were used for phenolic compounds identification, DPPH inhibition and FRAP values determinations. For carbohydrates, freeze-dried cells were suspended in 5 mL of acidified water (pH=2), the samples were sonicated for 10 min, stirred for 5 min, and centrifuged at 7000 rpm for 15 min and the supernatant was separated.

Seawater samples (700 mL) enriched with exudates were previously subjected to solid phase extraction (SPE) at a flow rate of 2 mL min$^{-1}$ using Macherey-Nagel Chromabond Easy cartridges (500 mg). The retained analytes were eluted with 12 mL of methanol, which was subsequently evaporated on a rotary evaporator. The residue was dissolved in 300 µl of methanol and filtered through a 0.22 µm filter to be injected into the HPLC equipment. These samples were used to quantify phenolic compounds and DPPH inhibition values.

**3.1 Carbonate chemistry parameters**

Preliminary tests indicated good stability of carbonate chemistry parameters in media, obtained by the $CO_2$ regulation system (Table 1). After $CO_2$-equilibration, initial DIC values ranged from 1905±26 to 2215±16 at a TA mean value of 2386±16 µmol kg$^{-1}$ (ANOVA; F=1.729, $p=0.2382$ among treatments), corresponding to a $CO_2$ range of 225±1 and 914±13 µatm. Although carbonate chemistry in $CO_2$-manipulated experiments can be strongly affected by biological activity (Howes et al., 2017; Miller and Kelley, 2021) during our research TA and DIC remained fairly stable (t-tests; p>0.05) within treatments, over the 8-day experimental period.

**Table 1.** Carbonate chemistry parameters in experimental media for each pH treatment at day 0 and day 8: total alkalinity (TA), total dissolved inorganic carbon concentration (DIC) and estimated $pCO_2$ (µatm).

| pH-Treatments | TA (µmol kg$^{-1}$) | | DIC (µmol kg$^{-1}$) | | $pCO_2$ (µatm) | |
|---|---|---|---|---|---|---|
| | Day 0 | Day 8 | Day 0 | Day 8 | Day 0 | Day 8 |
| 8.25 | 2376 ± 12 | 2335 ± 25 | 1905 ± 26 | 1869 ± 61 | 225 ± 1 | 221 ± 4 |
| 8.10 | 2380 ± 15 | 2329 ± 28 | 2012 ± 28 | 1971 ± 44 | 353 ± 2 | 349 ± 5 |
| 7.90 | 2390 ± 17 | 2347 ± 40 | 2129 ± 47 | 2085 ± 36 | 616 ± 12 | 599 ± 8 |
| 7.75 | 2401±14 | 2365 ± 26 | 2215 ± 16 | 2178 ± 39 | 914 ± 18 | 925 ± 27 |

Means and standard deviations were calculated from sampling (n = 3).

---

## Referee Report (RR1)

Comments on "Variations of polyphenols and carbohydrates of *Emiliania huxleyi* grown under simulated ocean acidification conditions."

This study reports the effect of ocean acidification on phenols and carbohydrates contents of *Emiliania huxleyi*. The idea is nice and the experimental setup is good. The data shown here are collected and analyzed clearly, and provide useful information for the biogeochemical cycling of carbon in future ocean acidification. I only have a few minor comments:

(1) Can you explain the relationships between carbohydrates and phenols in the introduction section of this manuscript?

(2) Lines 216-218: "In contrast to these results, .........". This sentence is too long, please rewrite it.

(3) Lines 237-240: The unit of Chl *a* is "fmol cell$^{-1}$" here. Please also show it in "pg cell$^{-1}$" in a bracket, such as 56.6±2.8 fmol cell$^{-1}$ (**±** pg cell$^{-1}$).

(4) For figure 2, the first point in the x-axis should be "7.75" rather than "7.5". Please change it.

(5) Lines 292-296: "ROS production was also correlated with ……..for *N. gaditana* cells at pH 6.0". It is so difficult to understand this sentence. Please rewrite it.

(6) In the introduction and discussion sections, such as in lines 70-79 and

317-337, the authors talk about the contents of Fe. It is better to measure the Fe concentration in the seawater at the beginning and end of the incubations in future study.

(7) Lines 415-416, there are logistic problems about this sentence "Engel (2015) reported that …… in their study". Please rewrite it.

---

## Author Response (AR2)

Las Palmas de Gran Canaria, July 31, 2024

**Point-by-point response to reviewers**

We really appreciate suggestions made by Reviewers.

Our answers and clarifications are written in blue. Modified texts in the manuscript are highlighted.

**Report #1**

**Checklist for reviewers**

| | |
|---|---|
| **1) Scientific significance**
Does the manuscript represent a substantial contribution to scientific progress within the scope of this journal (substantial new concepts, ideas, methods, or data)? | Excellent **Good** Fair Poor |
| **2) Scientific quality**
Are the scientific approach and applied methods valid? Are the results discussed in an appropriate and balanced way (consideration of related work, including appropriate references)? | Excellent **Good** Fair Poor |
| **3) Presentation quality**
Are the scientific results and conclusions presented in a clear, concise, and well structured way (number and quality of figures/tables, appropriate use of English language)? | Excellent **Good** Fair Poor |

**For final publication, the manuscript should be**

**accepted as is**

accepted subject to **technical corrections**

**accepted subject to minor revisions**

reconsidered after **major revisions**

**rejected**

**Were a revised manuscript to be sent for another round of reviews:**

**I would be willing to review the revised manuscript.**

I would not be willing to review the revised manuscript.

**Suggestions for revision or reasons for rejection**

(visible to the public if the article is accepted and published)
The authors have made substantial revisions on the manuscript, however, they are still some questions which need authors to address.
1. I am not satisfied with the current abstract, it need further refinement, some statements should be omitted.
2. The photoperiod was 24 h? Why no dark phase

**Point by point response**

1. I am not satisfied with the current abstract, it need further refinement, some statements should be omitted.

The abstract has been changed as follows:

**Abstract.** Cultures of the coccolithophore *Emiliania huxleyi* were grown under four different $CO_2$-controlled pH conditions (7.75, 7.90, 8.10, and 8.25) to explore variations in intra- and extracellular polyphenols and carbohydrates in response to different ocean acidification scenarios. Acidification did not significantly affect final cell densities and carbohydrate contents. Intra- and extracellular phenolic compounds were identified and quantified by Reverse Phase-High Performance Liquid Chromatography (RP-HPLC), with the highest concentrations of total exuded phenolics at pH 8.25 (43±3 nM) and 7.75 (18.0±0.9 nM). Accumulation of intracellular phenolic compounds was observed in cells with decreasing pH, reaching the maximum level (9.24±0.19 attomole $cell^{-1}$) at the lowest pH (7.75). The phenolic profiles presented significant changes in exuded epicatechin and protocatechuic acid ($p<0.05$ and 0.01, respectively), and intracellular vanillic acid ($p<0.001$), which play an essential role in the availability of trace metals. A significant increase in chlorophyll *a* content was observed in cells grown at the most acidic pH ($p<0.01$), which also showed significantly higher radical inhibition activity ($p<0.01$). The nature and concentration of these organic compounds present in the culture medium may influence trace metal bioavailability affecting the biogeochemical cycling of carbon and microbial functional diversity.

The photoperiod was 24 h? Why no dark phase

Even that no dark phase was applied in our studies, *E. huxleyi* has often been considered extremely light-tolerant (Jakob et al., 2018; Loebl et al., 2010). Xing et al. (2015) reported that *E. huxleyi* grown under indoor constant light showed higher specific growth rate than those grown under fluctuating outdoor solar radiation. Therefore, we applied a photoperiod of 24 h to provide maximum organic matter production, keeping this condition in all the studies. Moreover, pH, through $CO_2$ acidification, is the only variable modified in our study focused on the effect of acidification on *E. huxleyi*, so changes in organic matter should be linked only to the effect of this pH change and its consequences (changes in the availability of essential metals such as iron). We used the same strains as well as the cultivation conditions (lighting, seawater, nutrients, temperature, etc.) so the influence of all these factors should be the same in all cultures.

Axenic cultures of *E. huxleyi* (strain RCC1238) were supplied by the Spanish Bank of Algae (BEA) (member of the European Culture Collections Organization (ECCO) located

in Taliarte, SE coast of Gran Canaria), and cultured following its recommendations to increase the productivity. This microalgae Collection is included in the World Data Centre for Microorganisms (WFCC-MIRCEN) and is recognized by the World Intellectual Property Organization (WIPO) as international authority for the deposit of microorganisms (algae) through the Budapest Treaty. It is also included in the European Consortium MIRRI.

- Jakob, I., Weggenmann, F., Posten, C., Cultivation of Emiliania huxleyi for coccolith production, Algal Research, 31, 47-59, https://doi.org/10.1016/j.algal.2018.01.013, 2018.
- Loebl M., Cockshutt A. M., Campbell, D. A., Finkel, Z. V.: Physiological basis for high resistance to photoinhibition under nitrogen depletion in *Emiliania huxleyi*, *Limnology and Oceanography*, 55, 1807-2229, https://doi.org/10.4319/lo.2010.55.5.2150, 2010.
- Xing, T., Gao, K. and Beardall, J.: Response of Growth and Photosynthesis of *Emiliania huxleyi* to Visible and UV Irradiances under Different Light Regimes. Photochem Photobiol, 91: 343-349. https://doi.org/10.1111/php.12403, 2015.

**Report #2**

**Anonymous during peer-review: Yes** No
**Anonymous in acknowledgements of published article: Yes** No

**Checklist for reviewers**

| | |
|---|---|
| **1) Scientific significance**
Does the manuscript represent a substantial contribution to scientific progress within the scope of this journal (substantial new concepts, ideas, methods, or data)? | Excellent Good **Fair** Poor |
| **2) Scientific quality**
Are the scientific approach and applied methods valid? Are the results discussed in an appropriate and balanced way (consideration of related work, including appropriate references)? | Excellent Good Fair **Poor** |
| **3) Presentation quality**
Are the scientific results and conclusions presented in a clear, concise, and well structured way (number and quality of figures/tables, appropriate use of English language)? | Excellent Good Fair **Poor** |

**For final publication, the manuscript should be**

**accepted as is**

accepted subject to **technical corrections**

accepted subject to **minor revisions**

reconsidered after **major revisions**

**rejected**

**Were a revised manuscript to be sent for another round of reviews:**

I would be willing to review the revised manuscript.

**I would not be willing to review the revised manuscript.**

**Suggestions for revision or reasons for rejection**

(visible to the public if the article is accepted and published)

Overview

Technically, I am not inclined to provide support for this published paper in this journal. The focus of this work is on physiological variations of E. hux, which have already been extensively reported. Therefore, this manuscript does not offer anything new except for the testing of four different pH levels. One major concern is the poorly written manuscript with unclear descriptions. For instance, the method section regarding the carbonate system needs to be rewritten, and there is no mention of the $CO_2$ calculation, among other things. In my opinion, it would be more beneficial to include fewer molecular approaches and focus on identifying function genes related to toxicity and polyphenols. Additionally, significant improvements in writing style are necessary before submitting elsewhere.

Down below are fewer suggestions/questions.

Abstract
The response of OA to E. hux has already been extensively documented both physiologically and metabolically. This sentence should be revised to align with the aim of this work, which is to explore variations in polyphenols and carbohydrates. The abstract now appears to be more results-oriented rather than providing an overview. It should be revised to incorporate references to published articles.

Introduction
Reference needed.

Line 43: State as "OA" with a lowering of pH.

The main aim of this study is to investigate ocean acidification (OA). Therefore, a small introduction explaining the definition of OA would be very useful for readers to understand what ocean acidification is.

Lines 40 to 80: This section is too vague and could benefit from reducing and splitting the paragraphs.

Methods
Line 116-120. Rewrite. Too vague.

Do all variations of the conditions have the same temperature and incubation time in the culture incubator? Not clearly written.

Don't you think 200 µmol is too much for E. hux?
Please provide a clearer explanation of how CO2 was introduced into the cultures. Was it done using a single cylinder in different incubators or with multiple cylinders?

Results
How were the values calculated? Did they use the standard protocol and calculation methods, such as CO2SYS software, etc.? A clear description is needed.

The authors discussed other published papers that reported on E. hux at different pH levels, but they examined the concentration of pCO2 injected rather than the pH changes. This raises the question of how much concentration was injected to achieve each pH variation in this study.

Line 435: Is "limited research" completely wrong? Numerous studies have already been reported, not only on E. hux, but also on other phytoplankton, regarding topics such as warming, acidification, etc.

There are many concerns troughout the results and discussion.

Conclusion has to be completely based on the author's own results.

Point by point response:

Studying the nature of the organic ligands in oceanic waters will allow a comprehensive understanding of the consequences of acidification on ocean biogeochemical processes. Catechin, sinapic acid and gallic acid were found to increase the persistence of dissolved Fe, regenerating Fe(II) in seawater from 0.05% to 11.92% (González et al., 2019).

- González, A. G, Cadena-Aizaga, M. I., Sarthou, G., González-Dávila, M., and Santana-Casiano, J. M.: Iron complexation by phenolic ligands in seawater, Chem. Geol., 511, 380–388, https://doi.org/10.1016/j.chemgeo.2018.10.017, 2019.

Introduction
Reference needed.

Line 43: State as "OA" with a lowering of pH.

The introduction has been modified to address this and the following suggestion as follows (lines 40-42):

The absorption of anthropogenic $CO_2$ into seawater alters the natural chemical balance of the $CO_2$-carbonate system resulting in a decrease of the chemical bases in seawater, increasing protons ($H^+$) and lowering its pH in a process termed "ocean acidification" with adverse consequences for marine ecosystems and human societies (Bates et al., 2014; Jiang et al., 2023; Lida et al., 2021). In fact, pre-industrial seawater pH (8.25) has already dropped to 8.10, and is expected to reach a pH of 7.85 in this century (Jacobson, 2005).

The reference Gruber et al. (2023) has been deleted and replaced with Bates et al. (2014), from which the definition of OA has been taken.

Gruber, N., Bakker, D. C. E., DeVries, T., Gregor, L., Hauck, J., Landschützer, P., McKinley G. A.: Trends and variability in the ocean carbon sink, Nat. Rev. Earth Environ., 4, 119–134, https://doi.org/10.1038/s43017-022-00381-x, 2023.

Bates, N. R., Astor, Y. M., Church, M. J., Currie, K., Dore, J. E., González-Dávila, M., Lorenzoni, L., Muller-Karger, F., Olafsson, J., & Santana-Casiano, J. M. (2014). A Time-Series View of Changing Surface Ocean Chemistry Due to Ocean Uptake of Anthropogenic $CO_2$ and Ocean Acidification, Oceanography, 27(1), 126–141, http://www.jstor.org/stable/24862128, 2014.

The main aim of this study is to investigate ocean acidification (OA). Therefore, a small introduction explaining the definition of OA would be very useful for readers to understand what ocean acidification is.

The definition has been included as follows (lines 40-42):

The absorption of anthropogenic $CO_2$ into seawater alters the natural chemical equilibrium of $CO_2$-carbonate system resulting in a decrease of the chemical bases in seawater, increasing protons ($H^+$) and lowering its pH in a process termed "ocean

acidification" with adverse consequences for marine ecosystems and human societies (Bates et al., 2014; Jiang et al., 2023; Lida et al., 2021). In fact, pre-industrial seawater pH (8.25) has already dropped to 8.10, and is expected to reach a pH of 7.85 in this century (Jacobson, 2005).

Lines 40 to 80: This section is too vague and could benefit from reducing and splitting the paragraphs.

Several sentences were included in this section according to the suggestion of previous reviewers. However, the introduction has been changed as follows:

Global environmental changes, in particular those related to increasing temperature and decreasing pH, profoundly affect ocean ecosystems at many levels, as these are the two main variables controlling all chemical and biological cycles, with a major impact on the growth and metabolic functions of microalgae (Berge et al., 2010; Dedman et al., 2023; Kholssi et al., 2023; Lu et al., 2013). The absorption of anthropogenic $CO_2$ into seawater alters the natural chemical equilibrium of $CO_2$-carbonate system resulting in a decrease of the chemical bases, increasing protons concentration ($H^+$) and lowering its pH in a process termed "ocean acidification" lowers its pH with adverse consequences for marine ecosystems and human societies (Bates et al., 2014; Gruber et al., 2023; Jiang et al., 2023; Lida et al., 2021). In fact, pre-industrial seawater pH (8.25) has already dropped to 8.10, and is expected to reach a pH of 7.85 in this century (Jacobson, 2005).

For instance, pH homeostasis, which regulates the pH inside and outside the cell, is critical for the growth and metabolism of most microorganisms, including microalgae (; Guan and Liu, 2020; Lund et al., 2020). Different algal species show different optimal pH ranges for maximum growth (Hoppe et al., 2011; Kholssi et al., 2023). Changes in environmental pH could have consequences on the competitiveness of both sensitive and tolerant microalgae in mixed phytoplankton communities, modifying their structure, composition, and distribution, which are crucial in mitigating global environmental change by fixing and transporting carbon from the upper to the deep ocean in the major global carbon sink (Eltanahy and Torky, 2021; Kholssi et al., 2023, Marinov et al., 2010). Vasconcelos et al. (2002) found that exudates from *Phaeodactylumn tricornutum* (*P. tricornutum*) diatoms caused a toxic effect on *E. huxleyi*, while those from *Enteromorpha* spp. caused an increase in final cell yield, concluding that specific exudates produced by the bloom of one algal species may favour or inhibit the local growth of other species. Such changes could also affect species at a higher trophic level, resulting in a potential shift in biodiversity (Jin and Kirk, 2018). Spisla et al. (2021) reported that extreme $CO_2$ events modify the composition of particulate organic matter in the ocean, which leads to a substantial reorganization of the planktonic community, affecting multiple trophic levels from phytoplankton to primary and secondary consumers (Nelson et al., 2020; Trombetta et al., 2019;). Nelson et al. (2020) found modifications of planktonic and benthic communities in response to reduced seawater pH (from pH 8.1 to 7.8 and 7.4), concluding that a re-arrangement of the biofilm microbial communities occurred through a potential shift from autotrophic to heterotrophic dominated biofilms. In addition, microbial biofilms obtained under reduced pH altered settlement rates in invertebrate larvae of *Galeolaria hystrix*. Barcelos e Ramos et al. (2022) showed that coexistence with

other microorganisms modifies the response of *E. huxleyi* subjected to high $CO_2$ concentration, markedly decreasing its growth rate and cellular organic carbon and increasing its organic carbon in the presence of at elevated $CO_2$ concentrations with the bacteria *Idiomarina abyssalis* and *Brachybacterium* sp. Moreover, elevated $CO_2$ concentrations increased organic carbon and decreased inorganic carbon content of *E. huxleyi* cells in the presence of *I. abyssalis*, but not *Brachybacterium* sp.

Changes in phytoplankton communities due to variation in seawater acidity alter the composition of the organic ligands that these communities released into the surrounding environment (Samperio-Ramos et al., 2017). These ligands are crucial in the formation of metal complexes for acquiring micronutrients, sequestering toxic metals, and establishing electrochemical gradients that result in changes in speciation, bioavailability, and toxicity of trace metals (Harmesa et al., 2022; Santana-Casiano et al., 2014). Iron is an essential micronutrient for phytoplankton involved in fundamental cellular processes, including respiration, photosynthesis, nitrogen uptake, and nitrogen fixation (Raven et al., 1999; Hogle et al., 2014), controlling productivity, species composition and trophic structure of microbial communities over large regions of the ocean (González et al., 2019; Hunter and Boyd, 2007). Iron concentrations in ocean waters are very low due to its low solubility and effective removal from the ocean surface by phytoplankton (Liu and Millero, 2002). Complexation with organic compounds is one of the mechanisms for maintaining dissolved iron concentrations above its inorganic solubility, while potentially reducing the concentrations of soluble and bioavailable inorganic species (Hunter and Boyd, 2007; Shaked et al., 2020). A decrease in seawater pH from 8.1 to 7.4 will increase Fe(III) solubility by approximately 40%, which could have a large impact on biogeochemical cycles (Morel and Price, 2003; Millero et al., 2009). Organic matter exuded by marine microorganisms can form Fe(III) complexes that modify Fe(II) oxidation rates and promote the reduction of Fe(III) to Fe(II) in seawater. In addition, some research work has shown that the residence time of the reduced form of essential trace metals increases as their oxidation rate decreases under acidifying conditions (Pérez-Almeida et al., 2022; Santana-Casiano et al., 2014).

Methods
Line 116-120. Rewrite. Too vague.

The highlighted information has been included (line 110):

Axenic cultures of *E. huxleyi* (strain RCC1238) were supplied by the Spanish Bank of Algae (BEA) in f/2 medium. *E. huxleyi* coccolithophore was cultured for 8 days in an incubator clean chamber (Friocell FC111) at a constant temperature of 25 ºC with an initial cell density of $10^6$ cells $L^{-1}$, under complete photoperiod (24 h) with light intensity of 200 µmol photons $m^{-2}$ and under different $p$$CO_2$-controlled seawater pH conditions (7.75, 7.90, 8.10, and 8.25), measured on the free hydrogen ion scale $pH_F = -\log[H+]$ with a Ross Combination glass body electrode calibrated daily with TRIS buffer solutions. A gaseous mixture of $CO_2$-free air and pure $CO_2$ was bubbled in the culture medium to $CO_2$ levels of 900 µatm (pH 7.75), 600 µatm (pH 7.90), 350 µatm (pH 8.10), and 225 µatm (pH 8.25). Two gas cylinders were used for each incubator. To ensure quasi-constant seawater carbonate chemistry, a solenoid valve connected to both gas cylinders (pure

CO₂-free air cylinder and pure CO₂ cylinder) and a pH controller modulates the CO₂ flow rate once the desired pH is reached, keeping it constant (±0.02) (Samperio-Ramos et al., 2017). The culture medium was sterile filtered (0.1 μm) North Atlantic seawater (S = 36.48) obtained at the ESTOC site (29°10' N, 15°30' W).

Do all variations of the conditions have the same temperature and incubation time in the culture incubator? Not clearly written.

All experiments have the same incubation time (8 days) and temperature (25°C). This has been clarified in section 2.2 as follows:

- Axenic cultures of *E. huxleyi* (strain RCC1238) were supplied by the Spanish Bank of Algae (BEA) in f/2 medium. *E. huxleyi* coccolithophore was cultured for 8 days in an incubator clean chamber (Friocell FC111) at a constant temperature of 25 °C with an initial cell density of $10^6$ cells $L^{-1}$,…

Furthermore, in the same section (2.2 culture), it is stated (lines 132-133):

- Gas equilibrium in the media of each treatment was reached after a maximum of 24 h, as observed by the pH evolution. TA and DIC were measured at the beginning and end of the experiment on days 0 and 8 using a VINDTA 3C system (González-Dávila et al.…

Don't you think 200 μmol is too much for E. hux?

Coccolithophore growth rates usually increase with increased light intensity, level off at saturated light intensity and decline at inhibiting high light intensity (Zhang et al., 2019). Growth rates of *E. huxleyi* increased with elevated light intensity up to 200 μmol photons $m^{-2}\,s^{-1}$ and significantly declined thereafter (all $P < 0.001$) (The following figure has been extracted from Zhang et al., 2019).

[Figure]

Growth rate of *Emiliania huxleyi* as a function of light intensities at low $p$CO₂ (LC, hollow) and high $p$CO₂ levels (HC, solid) under **a** high dissolved inorganic nitrogen (DIN) and phosphate (DIP) concentrations (HNHP), **b** low DIN and high DIP concentrations (LN), and **c** high DIN and low DIP concentrations (LP). The lines in each panel were fitted using the model provided by Eilers and Peeters (1988). The values represent the mean ± standard deviation for four replicates

In addition, axenic cultures of *E. huxleyi* (strain RCC1238) were supplied by the Spanish Bank of Algae (BEA) and cultured following its recommendations. This microalgae Collection is member of the European Culture Collections Organization (ECCO) located in Taliarte, SE coast of Gran Canaria. It is included in the World Data Centre for Microorganisms (WFCC-MIRCEN) and is recognized by the World Intellectual Property Organization (WIPO) as international authority for the deposit of microorganisms (algae) through the Budapest Treaty. It is also included in the European Consortium MIRRI.

- Zhang, Y., Fu, F., Hutchins, D.A . *et al.* Combined effects of $CO_2$ level, light intensity, and nutrient availability on the coccolithophore *Emiliania huxleyi*. *Hydrobiologia* **842**, 127–141 (2019). https://doi.org/10.1007/s10750-019-04031-0

Please provide a clearer explanation of how CO2 was introduced into the cultures. Was it done using a single cylinder in different incubators or with multiple cylinders?

Two cylinders were used to ensure quasi-constant seawater carbonate chemistry. When the seawater pH values reached the target value, a solenoid valve, connected to both gas cylinders (pure $CO_2$-free air cylinder and pure $CO_2$ cylinder) and a pH controller, modulated the $CO_2$ flux, maintaining the set pH ($\pm 0.02$).

To clarify this, section 2.2 has been modified as follows (lines 113-119):

A gaseous mixture of $CO_2$-free air and pure $CO_2$ was bubbled into the culture medium to $CO_2$ levels of 900 µatm (pH 7.75), 600 µatm (pH 7.90), 350 µatm (pH 8.10), and 225 µatm (pH 8.25). To ensure quasi-constant seawater carbonate chemistry a solenoid valve connected to a pH controller and both gas cylinders (pure $CO_2$-free air cylinder and pure $CO_2$ cylinder) modulates the $CO_2$ flow rate once the desired pH is reached, keeping it constant ($\pm 0.02$) (Samperio-Ramos et al., 2017). The culture medium was sterile filtered (0.1 µm) North Atlantic seawater (S = 36.48) obtained at the ESTOC site (29º10' N, 15º30' W))

[Figure]

**Figure 1.** CO₂/pH perturbation experiment set-up, indicating the components.

(Figure extracted from Samperio et al. (2017))

All protocols and methods are described in detail in Samperio et al. (2017) and González Dávila et al. (2011), cited in the manuscript.

The authors discussed other published papers that reported on E. hux at different pH levels, but they ₑₓₐₘᵢₙₑ𝒹 the concentration of pCO₂ injected rather than the pH changes. This raises the question of how much concentration was injected to achieve each pH variation in this study.

We simulated in our study past, nowadays and future $CO_2$ concentrations. Injections of $CO_2$ do not change alkalinity (TA) but increases partial pressure of $CO_2$ ($p$CO₂) and total dissolved inorganic carbon (DIC) and decreases pH (e.g., Bates et al., 2014). Therefore, using the initial measured TA, the pH is controlled by the $CO_2$ injected. Moreover, the pH was continuously monitored during the experiment. The pair TA and DIC was also measured at the end of the study to confirm the concentrations and values were maintained.

**Table 1.** Carbonate chemistry parameters in experimental media for each pH treatment at day 0 and day 8: total alkalinity (TA), total dissolved inorganic carbon concentration (DIC) and estimated $p$CO₂ (µatm).

| pH-Treatments | TA (µmol kg⁻¹) | | DIC (µmol kg⁻¹) | | $p$CO₂ (µatm) | |
|---|---|---|---|---|---|---|
| | Day 0 | Day 8 | Day 0 | Day 8 | Day 0 | Day 8 |
| 8.25 | 2376±12 | 2335±25 | 1905±26 | 1869±61 | 225±1 | 221±4 |
| 8.10 | 2380±15 | 2329±28 | 2012±28 | 1971±44 | 353±2 | 349±5 |
| 7.90 | 2390±17 | 2347±40 | 2129±47 | 2085±36 | 616±12 | 599±8 |
| 7.75 | 2401±14 | 2365±26 | 2215±16 | 2178±39 | 914±18 | 925±27 |

Means and standard deviations were calculated from sampling (n = 3).

Results
How were the values calculated? Did they use the standard protocol and calculation methods, such as CO2SYS software, etc.? A clear description is needed.

The measurements and calculation of the carbon dioxide system for this study was previously described by Samperio et al. (2017). The Seawater Carbonate package (Seacarb version 3.0), developed for R Studio software (R Development Core Team), was employed to calculate the values of $p\text{CO}_2$, using the experimental results of pH, dissolved inorganic carbon and total alkalinity, and considering the carbonic acid dissociation constants of Millero et al. (2006).

This information has been included in section 2.2 (lines 138-141) and the bibliography cited in the references section.

Line 435: Is "limited research" completely wrong? Numerous studies have already been reported, not only on E. hux, but also on other phytoplankton, regarding topics such as warming, acidification, etc.
This sentence has been deleted following the last recommendation here.

There are many concerns troughout the results and discussion.
Conclusion has to be completely based on the author's own results.
The conclusion has been modified as follows to focus on our results following this recommendation:

Acidification between the current pH of the oceans (8.1) and the future scenario of pH 7.75 leads to an increase in polyphenol production in *E. huxleyi* cells and their free radical inhibitory activity. More importantly, the change in the polyphenol profile between cells and exudates and between pH conditions should be closely related not only to their antioxidant activity under stress conditions, but also to the chemistry of iron and other trace metals and their bioavailability under different pH conditions. Intra- and extracellular carbohydrate levels did not show modifications with decreasing pH. These changes in metabolites with different capacity to inhibit radicals and complex metals, whose accumulation is associated with enhanced oxidative stress, are potential factors leading to readjustments in phytoplankton community structure and diversity and possible alteration in marine ecosystems.

**Report #3**

**Anonymous during peer-review: Yes** No
**Anonymous in acknowledgements of published article: Yes** No

**Checklist for reviewers**

| | |
|---|---|
| **1) Scientific significance**
Does the manuscript represent a substantial contribution to scientific progress within the scope of this journal (substantial new concepts, ideas, methods, or data)? | Excellent **Good** Fair Poor |
| **2) Scientific quality**
Are the scientific approach and applied methods valid? Are the results discussed in an appropriate and balanced way (consideration of related work, including appropriate references)? | Excellent **Good** Fair Poor |
| **3) Presentation quality**
Are the scientific results and conclusions presented in a clear, concise, and well structured way (number and quality of figures/tables, appropriate use of English language)? | Excellent **Good** Fair Poor |

**For final publication, the manuscript should be**

**accepted as is**

accepted subject to **technical corrections**

**accepted subject to minor revisions**

reconsidered after **major revisions**

**rejected**

**Were a revised manuscript to be sent for another round of reviews:**

I would be willing to review the revised manuscript.

**I would not be willing to review the revised manuscript.**

**Suggestions for revision or reasons for rejection**
(visible to the public if the article is accepted and published)

Comments on "Variations of polyphenols and carbohydrates of Emiliania huxleyi grown under simulated ocean acidification conditions."

This study reports the effect of ocean acidification on phenols and carbohydrates contents of Emiliania huxleyi. The idea is nice and the experimental setup is good. The data shown here are collected and analyzed clearly, and provide useful information for the biogeochemical cycling of carbon in future ocean acidification. I only have a few minor comments:

(1) Can you explain the relationships between carbohydrates and phenols in the introduction section of this manuscript?

(2) Lines 216-218: "In contrast to these results, ………". This sentence is too long,

please rewrite it.

(3) Lines 237-240: The unit of Chl a is "fmol cell–1" here. Please also show it in "pg cell–1" in a bracket, such as 56.6±2.8 fmol cell–1 (**±** pg cell–1).

(4) For figure 2, the first point in the x-axis should be "7.75" rather than "7.5". Please change it.

(5) Lines 292-296: "ROS production was also correlated with ……..for N. gaditana cells at pH 6.0". It is so difficult to understand this sentence. Please rewrite it.

(6) In the introduction and discussion sections, such as in lines 70-79 and 317-337, the authors talk about the contents of Fe. It is better to measure the Fe concentration in the seawater at the beginning and end of the incubations in future study.

(7) Lines 415-416, there are logistic problems about this sentence "Engel (2015) reported that …… in their study". Please rewrite it.

Point by point response:

(1) Can you explain the relationships between carbohydrates and phenols in the introduction section of this manuscript?

The authors are not sure which relationship the reviewer is referring to. These compounds were selected because they may influence the chemistry of iron and its bioavailability, as indicated in the introduction (lines 72-83) and in lines 318-338.

The antioxidant activities of complex carbohydrates have been attributed mainly to phenolic and protein components, rather than to carbohydrate molecules. Covalent and non-covalent interactions between carbohydrates and phenols are possible. Polysaccharides with covalently bound phenolic compounds acquire metal reducing properties, ability to inhibit oxidative enzymes and enhanced metal chelating properties due to the contribution of the electronegative character of the polyhydroxylated phenolic aromatic ring. Polysaccharides can also interact with phenolic compounds by means of hydrophobic effects, hydrogen bonding and Van der Waals interactions. Multiple binding sites along the polysaccharide backbone results in the formation of highly stable carbohydrate/phenolic complexes. (Fernandes et al., 2023)

The antioxidant activity of polysaccharides is highly dependent on several factors (solubility, molecular weight, occurrence of positive or negatively charged groups among others), being the presence of linked phenolic compounds the major contribution, which allows them to show RSA and metal reducing ability that polysaccharides devoid of phenolic and proteins groups do not exhibit (Chen et al. 2024; Fernandes and Coimbra, 2023). The antioxidant properties increase with the degree of polysaccharides substitution with phenolic compounds.

The following sentence and the cited reference have been included in the introduction (lines 83-86) and in the reference section respectively:

The metal reducing and chelating properties of polysaccharides are highly dependent on several factors, with the presence of covalently and non-covalently bound phenolic and protein components being the main contributing factor, allowing them to exhibit radical scavenging activity (RSA) and metal reduction capacity that they would not exhibit if devoid of these components (Fernandes and Coimbra, 2023).

- Fernandes P. A. R., Coimbra, M. A.: The antioxidant activity of polysaccharides: A structure-function relationship overview, Carbohyd. Polym., 314, 120965, https://doi.org/10.1016/j.carbpol.2023.120965, 2023.

(2) Lines 216-218: "In contrast to these results, ………". This sentence is too long, please rewrite it.

This sentence (lines 215-218) has been changed as follows:

In contrast to these results, Vázquez et al. (2022) found that acidification with $CO_2$ (1200 µatm, pH 7.62) induced lower coccolithophore growth rates than acidification reached without $CO_2$ enrichment compared to the control (400 µatm, pH 8.10). In addition, elevated $CO_2$ affected cell viability and promoted the ROS accumulation, effects not observed under low pH without $CO_2$ additions.

(3) Lines 237-240: The unit of Chl a is fmol cell$^{-1}$" here. Please also show it in "pg cell$^{-1}$" in a bracket, such as 56.6±2.8 fmol cell$^{-1}$ (**±** pg cell$^{-1}$).

There is an error in the amounts indicated in the manuscript that has been corrected. In the experimental section, the units are correctly stated: "Chl *a* was expressed as femtogram cell$^{-1}$ and quantified spectrophotometrically according to the equation: Chl *a* (mg/100 mL) = 0.999×$A_{663}$-0.0989×$A_{645}$". From the equation applied for quantification, the quantity is obtained directly in mass units. The quantities given in section 3.3 are calculated correctly, the error affects the units. The paragraph has been changed as follows

After 8 culture days, the concentration of Chl *a* per cell decreases with decreasing pH from 56.6±2.8 fg cell$^{-1}$ (pH 8.25) to 26.8±1.4 fg cell$^{-1}$ (pH 7.9). However, cells grown in the most acidic conditions (pH 7.75) show the highest amount of Chl *a* (67.3±2.0 fg cell$^{-1}$) with a significant increase observed between pH 8.1 (45.1±3.0 fg cell$^{-1}$) and 7.75 ($p<0.01$).

(4) For figure 2, the first point in the x-axis should be "7.75" rather than "7.5". Please change it.

This has been corrected.

(5) Lines 292-296: "ROS production was also correlated with ……..for N. gaditana cells at pH 6.0". It is so difficult to understand this sentence. Please rewrite it.

The sentence (lines 292-298) has been changed as follows:

Bautista-Chamizo et al. (2019) reported an increase in ROS production by decreasing the pH of the culture medium in single- and multispecies toxicity assays conducted with microalgae *T. Chuii*, *N. gaditana* and *P. tricornutum*. The species *P. tricornutum* and *N.*

*gaditana* exposed to pH 7.4 and pH 6.0 exhibited a significant increase in the percentage of intracellular ROS, which was more pronounced for *N. gaditana* cells at pH 6.0.

(6) In the introduction and discussion sections, such as in lines 70-79 and 317-337, the authors talk about the contents of Fe. It is better to measure the Fe concentration in the seawater at the beginning and end of the incubations in future study.

We agree with the reviewer that Fe concentration should be measured at the beginning and at the end of the incubations. In this study, iron was added to seawater from a stock solution (1 mM) of ferric chloride (Sigma) obtaining an initial concentration of 2.5 nM to avoid iron deficiency.

- Wei Jin, C., You, G. Y., & Zheng, S. J.: The iron deficiency-induced phenolics secretion plays multiple important roles in plant iron acquisition underground. Plant Signaling & Behavior, 3(1), 60–61. https://doi.org/10.4161/psb.3.1.4902, 2008.

(7) Lines 415-416, there are logistic problems about this sentence "Engel (2015) reported that …… in their study". Please rewrite it.

The sentence has been rewritten as follows (lines 414-416):

Borchard and Engel (2015) found no significant differences between growth rates and primary production (composed of dissolved and particulate organic carbon) of *E. huxleyi* grown at current and high $CO_2$ concentrations due to its ability to acclimate.

---

## Author Response (AR3)

Las Palmas de Gran Canaria, August 19, 2024

**Point-by-point response to reviewers of the article**

**Variations of polyphenols and carbohydrates of *Emiliania huxleyi* grown under simulated ocean acidification conditions**

Milagros Rico[1,2], Paula Santiago-Díaz[1,2], Guillermo Samperio-Ramos[3], Melchor González-Dávila[1,2], Juana Magdalena Santana-Casiano[1,2]

[1]Departmento de Química, Universidad de Las Palmas de Gran Canaria, Campus de Tafira, 35017 Las Palmas de Gran Canaria, Spain.
[2]Instituto de Oceanografía y Cambio Global (IOCAG), Universidad de Las Palmas de Gran Canaria, Unidad Asociada ULPGC-CSIC, Las Palmas de Gran Canaria, Spain.
[3]Instituto de Investigaciones Oceanológicas. Universidad Autónoma de Baja California. Carretera Ensenada-Tijuana nº 3917, Fracc. Playitas, Ensenada Baja California. C.P. 22860.

The authors really appreciate suggestions made by Reviewers.

Our answers and clarifications are written in blue. Modified texts in the manuscript are highlighted.

**Major revision 2**

**Report #1**

1. I am not satisfied with the current abstract, it need further refinement, some statements should be omitted.

The abstract has been changed as follows:

**Abstract.** Cultures of the coccolithophore *Emiliania huxleyi* were grown under four different $CO_2$-controlled pH conditions (7.75, 7.90, 8.10, and 8.25) to explore variations in intra- and extracellular polyphenols and carbohydrates in response to different ocean acidification scenarios.

The following highlighted sentences has been deleted:

However, no significant differences were found between the iron reducing activities and the radical scavenging activities of the compounds present in the exudates. The nature and concentration of these organic compounds present in the culture medium may favour or inhibit the local growth of specific algal species, and influence trace metal bioavailability…

The photoperiod was 24 h? Why no dark phase

Even that no dark phase was applied in our studies, *E. huxleyi* has often been considered extremely light-tolerant (Jakob et al., 2018; Loebl et al., 2010). Xing et al. (2015) reported that *E. huxleyi* grown under indoor constant light showed higher specific growth rate than those grown under fluctuating outdoor solar radiation. Therefore, we applied a photoperiod of 24 h to provide maximum organic matter production, keeping this condition in all the studies. Moreover, pH, through $CO_2$ acidification, is the only variable modified in our study focused on the effect of acidification on *E. huxleyi*, so changes in organic matter should be linked only to the effect of this pH change and its consequences (changes in the availability of essential metals such as iron). We used the same strains as well as the cultivation conditions (lighting, seawater, nutrients, temperature, etc.) so the influence of all these factors should be the same in all cultures.

- Jakob, I., Weggenmann, F., Posten, C., Cultivation of Emiliania huxleyi for coccolith production, Algal Research, 31, 47-59, https://doi.org/10.1016/j.algal.2018.01.013, 2018.
- Loebl M., Cockshutt A. M., Campbell, D. A., Finkel, Z. V.: Physiological basis for high resistance to photoinhibition under nitrogen depletion in *Emiliania huxleyi*, *Limnology and Oceanography*, 55, 1807-2229, https://doi.org/10.4319/lo.2010.55.5.2150, 2010.
- Xing, T., Gao, K. and Beardall, J.: Response of Growth and Photosynthesis of *Emiliania huxleyi* to Visible and UV Irradiances under Different Light Regimes. Photochem Photobiol, 91: 343-349. https://doi.org/10.1111/php.12403, 2015.

**Report #2**

Technically, I am not inclined to provide support for this published paper in this journal. The focus of this work is on physiological variations of E. hux, which have already been extensively reported. Therefore, this manuscript does not offer anything new except for the testing of four different pH levels.

Studying the nature of the organic ligands in oceanic waters will allow a comprehensive understanding of the consequences of acidification on ocean biogeochemical processes. Catechin, sinapic acid and gallic acid were found to increase the persistence of dissolved Fe, regenerating Fe(II) in seawater from 0.05% to 11.92% (González et al., 2019).

- González, A. G, Cadena-Aizaga, M. I., Sarthou, G., González-Dávila, M., and Santana-Casiano, J. M.: Iron complexation by phenolic ligands in seawater, Chem. Geol., 511, 380–388, https://doi.org/10.1016/j.chemgeo.2018.10.017, 2019.

Introduction
Reference needed.

Line 43: State as "OA" with a lowering of pH.

The introduction has been modified including the following sentence to address this:

The absorption of anthropogenic $CO_2$ into seawater alters the natural chemical balance of the $CO_2$-carbonate system resulting in a decrease of the chemical bases in seawater, increasing protons ($H^+$) and lowering its pH in a process termed "ocean acidification" with adverse consequences for marine ecosystems and human societies (Bates et al., 2014; Jiang et al., 2023; Lida et al., 2021).

The reference Gruber et al. (2023) has been deleted and replaced by Bates et al. (2014), from which the definition of OA has been taken.

Bates, N. R., Astor, Y. M., Church, M. J., Currie, K., Dore, J. E., González-Dávila, M., Lorenzoni, L., Muller-Karger, F., Olafsson, J., & Santana-Casiano, J. M. (2014). A Time-Series View of Changing Surface Ocean Chemistry Due to Ocean Uptake of Anthropogenic $CO_2$ and Ocean Acidification, Oceanography, 27(1), 126–141, http://www.jstor.org/stable/24862128, 2014.

The main aim of this study is to investigate ocean acidification (OA). Therefore, a small introduction explaining the definition of OA would be very useful for readers to understand what ocean acidification is.

The definition has been included as described above.

Lines 40 to 80: This section is too vague and could benefit from reducing and splitting the paragraphs.

Several sentences were included in this section according to the suggestion of previous reviewers. However, the introduction has been changed as follows:

Global environmental changes, in particular those related to increasing temperature and decreasing pH, profoundly affect ocean ecosystems at many levels, as these are the two main variables controlling all chemical and biological cycles, with a major impact on the growth and metabolic functions of microalgae (Berge et al., 2010; Dedman et al., 2023; Kholssi et al., 2023; Lu et al., 2013). The absorption of anthropogenic $CO_2$ into seawater alters the natural chemical equilibrium of $CO_2$-carbonate system resulting in a decrease of the chemical bases, increasing protons concentration ($H^+$) and lowering its pH in a process termed "ocean acidification" lowers its pH with adverse consequences for marine ecosystems and human societies (Bates et al., 2014; Gruber et al., 2023; Jiang et al., 2023; Lida et al., 2021). In fact, pre-industrial seawater pH (8.25) has already dropped to 8.10, and is expected to reach a pH of 7.85 in this century (Jacobson, 2005).

For instance, pH homeostasis, which regulates the pH inside and outside the cell, is critical for the growth and metabolism of most microorganisms, including microalgae (; Guan and Liu, 2020; Lund et al., 2020). Different algal species show different optimal pH ranges for maximum growth (Hoppe et al., 2011; Kholssi et al., 2023). Changes in environmental pH could have consequences on the competitiveness of both sensitive and tolerant microalgae in mixed phytoplankton communities, modifying their structure, composition, and distribution, which are crucial in mitigating global environmental change by fixing and transporting carbon from the upper to the deep ocean in the major global carbon sink (Eltanahy and Torky, 2021; Kholssi et al., 2023, Marinov et al., 2010). Vasconcelos et al. (2002) found that exudates from *Phaeodactylumn tricornutum* (*P. tricornutum*) diatoms caused a toxic effect on *E. huxleyi*, while those from *Enteromorpha* spp. caused an increase in final cell yield, concluding that specific exudates produced by the bloom of one algal species may favour or inhibit the local growth of other species. Such changes could also affect species at a higher trophic level, resulting in a potential shift in biodiversity (Jin and Kirk, 2018). Spisla et al. (2021) reported that extreme $CO_2$ events modify the composition of particulate organic matter in the ocean, which leads to a substantial reorganization of the planktonic community, affecting multiple trophic levels from phytoplankton to primary and secondary consumers (Nelson et al., 2020; Trombetta et al., 2019;). Nelson et al. (2020) found modifications of planktonic and benthic communities in

response to reduced seawater pH (from pH 8.1 to 7.8 and 7.4), concluding that a re-arrangement of the biofilm microbial communities occurred through a potential shift from autotrophic to heterotrophic dominated biofilms. In addition, microbial biofilms obtained under reduced pH altered settlement rates in invertebrate larvae of *Galeolaria hystrix*. Barcelos e Ramos et al. (2022) showed that coexistence with other microorganisms modifies the response of *E. huxleyi* subjected to high $CO_2$ concentration, markedly decreasing its growth rate and cellular organic carbon and increasing its organic carbon in the presence of at elevated $CO_2$ concentrations with the bacteria *Idiomarina abyssalis* and *Brachybacterium* sp. Moreover, elevated $CO_2$ concentrations increased organic carbon and decreased inorganic carbon content of *E. huxleyi* cells in the presence of *I. abyssalis*, but not *Brachybacterium* sp.

Changes in phytoplankton communities due to variation in seawater acidity alter the composition of the organic ligands that these communities released into the surrounding environment (Samperio-Ramos et al., 2017). These ligands are crucial in the formation of metal complexes for acquiring micronutrients, sequestering toxic metals, and establishing electrochemical gradients that result in changes in speciation, bioavailability, and toxicity of trace metals (Harmesa et al., 2022; Santana-Casiano et al., 2014). Iron is an essential micronutrient for phytoplankton involved in fundamental cellular processes, including respiration, photosynthesis, nitrogen uptake, and nitrogen fixation (Raven et al., 1999; Hogle et al., 2014), controlling productivity, species composition and trophic structure of microbial communities over large regions of the ocean (González et al., 2019; Hunter and Boyd, 2007). Iron concentrations in ocean waters are very low due to its low solubility and effective removal from the ocean surface by phytoplankton (Liu and Millero, 2002). Complexation with organic compounds is one of the mechanisms for maintaining dissolved iron concentrations above its inorganic solubility, while potentially reducing the concentrations of soluble and bioavailable inorganic species (Hunter and Boyd, 2007; Shaked et al., 2020). A decrease in seawater pH from 8.1 to 7.4 will increase Fe(III) solubility by approximately 40%, which could have a large impact on biogeochemical cycles (Morel and Price, 2003; Millero et al., 2009). Organic matter exuded by marine microorganisms can form Fe(III) complexes that modify Fe(II) oxidation rates and promote the reduction of Fe(III) to Fe(II) in seawater. In addition, some research work has shown that the residence time of the reduced form of essential trace metals increases as their oxidation rate decreases under acidifying conditions (Pérez-Almeida et al., 2022; Santana-Casiano et al., 2014).

Methods
Line 116-120. Rewrite. Too vague.

The highlighted information extracted from Samperio et al. (2017) has been included:

Axenic cultures of *E. huxleyi* (strain RCC1238) were supplied by the Spanish Bank of Algae (BEA) in f/2 medium. *E. huxleyi* coccolithophore was cultured for 8 days in an incubator clean chamber (Friocell FC111) at a constant temperature of 25 ºC with an initial cell density of $10^6$ cells $L^{-1}$, under complete photoperiod (24 h) with light intensity of 200 µmol photons $m^{-2}$ and under different $p$CO$_2$-controlled seawater pH conditions (7.75, 7.90, 8.10, and 8.25), measured on the free hydrogen ion scale pH$_F$= $-\log$[H+] with a Ross Combination glass body electrode calibrated daily with TRIS buffer solutions. A gaseous mixture of $CO_2$-free air and pure $CO_2$ was bubbled in the culture medium to $CO_2$ levels of 900 µatm (pH 7.75), 600 µatm (pH 7.90), 350 µatm (pH 8.10), and 225 µatm (pH 8.25). Two gas cylinders were used for each incubator. To ensure quasi-constant seawater carbonate chemistry, a solenoid valve connected to both gas cylinders (pure $CO_2$-free air cylinder and pure $CO_2$

cylinder) and a pH controller modulates the $CO_2$ flow rate once the desired pH is reached, keeping it constant ($\pm0.02$) (Samperio-Ramos et al., 2017). The culture medium was sterile filtered (0.1 µm) North Atlantic seawater (S = 36.48) obtained at the ESTOC site (29°10' N, 15°30' W).

Do all variations of the conditions have the same temperature and incubation time in the culture incubator? Not clearly written.

All experiments have the same incubation time (8 days) and temperature (25ºC). This has been clarified in section 2.2 as described above. Furthermore, in the same section (2.2 culture), it is stated:

- Gas equilibrium in the media of each treatment was reached after a maximum of 24 h, as observed by the pH evolution. TA and DIC were measured at the beginning and end of the experiment on days 0 and 8 using a VINDTA 3C system (González-Dávila et al.…

Don't you think 200 µmol is too much for E. hux?

Coccolithophore growth rates usually increase with increased light intensity, level off at saturated light intensity and decline at inhibiting high light intensity (Zhang et al., 2019). Growth rates of *E. huxleyi* increased with elevated light intensity up to 200 µmol photons $m^{-2}$ $s^{-1}$ and significantly declined thereafter (all $P < 0.001$).

- Zhang, Y., Fu, F., Hutchins, D.A . *et al.* Combined effects of $CO_2$ level, light intensity, and nutrient availability on the coccolithophore *Emiliania huxleyi. Hydrobiologia* **842**, 127–141 (2019). https://doi.org/10.1007/s10750-019-04031-0

Please provide a clearer explanation of how $CO_2$ was introduced into the cultures. Was it done using a single cylinder in different incubators or with multiple cylinders?

To clarify this, section 2.2 has been modified as follows:

A gaseous mixture of $CO_2$-free air and pure $CO_2$ was bubbled into the culture medium to $CO_2$ levels of 900 µatm (pH 7.75), 600 µatm (pH 7.90), 350 µatm (pH 8.10), and 225 µatm (pH 8.25). To ensure quasi-constant seawater carbonate chemistry a solenoid valve connected to a pH controller and both gas cylinders (pure $CO_2$-free air cylinder and pure $CO_2$ cylinder) modulates the $CO_2$ flow rate once the desired pH is reached, keeping it constant ($\pm0.02$) (Samperio-Ramos et al., 2017). The culture medium was sterile filtered (0.1 µm) North Atlantic seawater (S = 36.48) obtained at the ESTOC site (29°10' N, 15°30' W))

All protocols and methods are described in detail in Samperio et al. (2017) and González Dávila et al. (2011), cited in the manuscript. The following Figure has been extracted from Samperio et al. (2017).

[Figure]

**Figure 1.** $CO_2$/pH perturbation experiment set-up, indicating the components.

The authors discussed other published papers that reported on E. hux at different pH levels, but they examined the concentration of $pCO_2$ injected rather than the pH changes. This raises the question of how much concentration was injected to achieve each pH variation in this study.

We simulated in our study past, nowadays and future $CO_2$ concentrations. Injections of $CO_2$ do not change alkalinity (TA) but increases partial pressure of $CO_2$ ($pCO_2$) and total dissolved inorganic carbon (DIC) and decreases pH (e.g., Bates et al., 2014). Therefore, using the initial measured TA, the pH is controlled by the $CO_2$ injected. Moreover, the pH was continuously monitored during the experiment. The pair TA and DIC was also measured at the end of the study to confirm the concentrations and values were maintained.

**Table 1.** Carbonate chemistry parameters in experimental media for each pH treatment at day 0 and day 8: total alkalinity (TA), total dissolved inorganic carbon concentration (DIC) and estimated $pCO_2$ (µatm).

| pH-Treatments | TA (µmol kg$^{-1}$) | | DIC (µmol kg$^{-1}$) | | $pCO_2$ (µatm) | |
|---|---|---|---|---|---|---|
| | Day 0 | Day 8 | Day 0 | Day 8 | Day 0 | Day 8 |
| 8.25 | 2376±12 | 2335±25 | 1905±26 | 1869±61 | 225±1 | 221±4 |
| 8.10 | 2380±15 | 2329±28 | 2012±28 | 1971±44 | 353±2 | 349±5 |
| 7.90 | 2390±17 | 2347±40 | 2129±47 | 2085±36 | 616±12 | 599±8 |
| 7.75 | 2401±14 | 2365±26 | 2215±16 | 2178±39 | 914±18 | 925±27 |

Means and standard deviations were calculated from sampling (n = 3).

Results
How were the values calculated? Did they use the standard protocol and calculation methods, such as CO2SYS software, etc.? A clear description is needed.

The measurements and calculation of the carbon dioxide system for this study was previously described by Samperio et al. (2017). The Seawater Carbonate package (Seacarb version 3.0), developed for R Studio software (R Development Core Team), was employed to calculate the values of $p$CO$_2$, using the experimental results of pH, dissolved inorganic carbon and total alkalinity, and considering the carbonic acid dissociation constants of Millero et al. (2006).

This information has been included in section 2.2 and the bibliography cited in the references section.

Line 435: Is "limited research" completely wrong? Numerous studies have already been reported, not only on E. hux, but also on other phytoplankton, regarding topics such as warming, acidification, etc.

This sentence has been deleted.

There are many concerns throughout the results and discussion.
Conclusion has to be completely based on the author's own results.

The conclusion has been modified by eliminating the following sentences:

Global environmental change influences the growth and metabolic functions of microalgae affecting their communities' structure and compositions depending on the sensitivity of different taxonomic groups, with implications for higher trophic levels and biodiversity loss. Limited research has been focused on the ecological consequences of joint action of seawater pH decrease, warming, and changes in salinity, among others.

**Report #3**

(1) Can you explain the relationships between carbohydrates and phenols in the introduction section of this manuscript?

The authors are not sure which relationship the reviewer is referring to. These compounds were selected because they may influence the chemistry of iron and its bioavailability, as indicated in the introduction and in section 3.4.

The antioxidant activities of complex carbohydrates have been attributed mainly to phenolic and protein components, rather than to carbohydrate molecules. Covalent and non-covalent interactions between carbohydrates and phenols are possible. Polysaccharides with covalently bound phenolic compounds acquire metal reducing properties, ability to inhibit oxidative enzymes and enhanced metal chelating properties due to the contribution of the electronegative character of the polyhydroxylated phenolic aromatic ring. Polysaccharides can also interact with phenolic compounds by means of hydrophobic effects, hydrogen bonding and Van der Waals interactions. Multiple binding sites along the polysaccharide backbone results in the formation of highly stable carbohydrate/phenolic complexes. (Fernandes et al., 2023)
The antioxidant activity of polysaccharides is highly dependent on several factors (solubility, molecular weight, occurrence of positive or negatively charged groups among others), being the

presence of linked phenolic compounds the major contribution, which allows them to show RSA and metal reducing ability that polysaccharides devoid of phenolic and proteins groups do not exhibit (Chen et al. 2024; Fernandes and Coimbra, 2023). The antioxidant properties increase with the degree of polysaccharides substitution with phenolic compounds.

The following sentence and the cited reference have been included in the introduction (lines 83-86) and in the reference section respectively:

The metal reducing and chelating properties of polysaccharides are highly dependent on several factors, with the presence of covalently and non-covalently bound phenolic and protein components being the main contributing factor, allowing them to exhibit radical scavenging activity (RSA) and metal reduction capacity that they would not exhibit if devoid of these components (Fernandes and Coimbra, 2023).

- Fernandes P. A. R., Coimbra, M. A.: The antioxidant activity of polysaccharides: A structure-function relationship overview, Carbohyd. Polym., 314, 120965, https://doi.org/10.1016/j.carbpol.2023.120965, 2023.

(2) Lines 216-218: "In contrast to these results, ………". This sentence is too long, please rewrite it.

This sentence (lines 215-218) has been changed as follows:

In contrast to these results, Vázquez et al. (2022) found that acidification with $CO_2$ (1200 µatm, pH 7.62) induced lower coccolithophore growth rates than acidification reached without $CO_2$ enrichment compared to the control (400 µatm, pH 8.10). In addition, elevated $CO_2$ affected cell viability and promoted the ROS accumulation, effects not observed under low pH without $CO_2$ additions.

(3) Lines 237-240: The unit of Chl a is fmol cell$^{-1}$" here. Please also show it in "pg cell$^{-1}$" in a bracket, such as 56.6±2.8 fmol cell$^{-1}$ (**±** pg cell$^{-1}$).

There is an error in the amounts indicated in the manuscript that has been corrected. In the experimental section, the units are correctly stated: "Chl $a$ was expressed as femtogram cell$^{-1}$ and quantified spectrophotometrically according to the equation: Chl $a$ (mg/100 mL) = 0.999×$A_{663}$-0.0989×$A_{645}$". From the equation applied for quantification, the quantity is obtained directly in mass units. The quantities given in section 3.3 are calculated correctly, the error affects the units. The paragraph has been changed as follows

After 8 culture days, the concentration of Chl $a$ per cell decreases with decreasing pH from 56.6±2.8 fg cell$^{-1}$ (pH 8.25) to 26.8±1.4 fg cell$^{-1}$ (pH 7.9). However, cells grown in the most acidic conditions (pH 7.75) show the highest amount of Chl $a$ (67.3±2.0 fg cell$^{-1}$) with a significant increase observed between pH 8.1 (45.1±3.0 fg cell$^{-1}$) and 7.75 ($p<0.01$).

(4) For figure 2, the first point in the x-axis should be "7.75" rather than "7.5". Please change it.

This has been corrected.

(5) Lines 292-296: "ROS production was also correlated with ……..for N. gaditana cells at pH 6.0". It is so difficult to understand this sentence. Please rewrite it.

The sentence has been changed as follows:

Bautista-Chamizo et al. (2019) reported an increase in ROS production by decreasing the pH of the culture medium in single- and multispecies toxicity assays conducted with microalgae *T. Chuii*, *N. gaditana* and *P. tricornutum*. The species *P. tricornutum* and *N. gaditana* exposed to pH 7.4 and pH 6.0 exhibited a significant increase in the percentage of intracellular ROS, which was more pronounced for *N. gaditana* cells at pH 6.0.

(6) In the introduction and discussion sections, such as in lines 70-79 and 317-337, the authors talk about the contents of Fe. It is better to measure the Fe concentration in the seawater at the beginning and end of the incubations in future study.

We agree with the reviewer that Fe concentration should be measured at the beginning and at the end of the incubations. In this study, iron was added to seawater from a stock solution (1 mM) of ferric chloride (Sigma) obtaining an initial concentration of 2.5 nM to avoid iron deficiency.

- Wei Jin, C., You, G. Y., & Zheng, S. J.: The iron deficiency-induced phenolics secretion plays multiple important roles in plant iron acquisition underground. Plant Signaling & Behavior, 3(1), 60–61. https://doi.org/10.4161/psb.3.1.4902, 2008.

(7) Lines 415-416, there are logistic problems about this sentence "Engel (2015) reported that …… in their study". Please rewrite it.

The sentence has been rewritten as follows (lines 414-416):

Borchard and Engel (2015) found no significant differences between growth rates and primary production (composed of dissolved and particulate organic carbon) of *E. huxleyi* grown at current and high $CO_2$ concentrations due to its ability to acclimate.

**Major revision 1**

We really thank and appreciate important comments and suggestions made by Reviewers. Substantial changes have been included in the manuscript in accordance with their suggestions and have been highlighted. The following references have been deleted or replaced by others that are more in line with the amended discussion.

- Antosiewicz, J. M., and Kane, P. M.: Editorial: Intracellular Molecular Processes Affected by pH. Front. Mol. Biosci. Sec. Biophysics, 9, 891533, https://doi: 10.3389/fmolb.2022.891533, 2022.
- Casey, J., Grinstein, S., and Orlowski, J.: Sensors and regulators of intracellular pH, Nat. Rev. Mol. Cell. Biol., 11, 50–61. https://doi.org/10.1038/nrm2820, 2010.
- Salam, U., Ullah, S., Tang, Z.-H., Elateeq, A.A., Khan, Y., Khan, J., Khan, A., and Ali, S.: Plant Metabolomics: An Overview of the Role of Primary and Secondary Metabolites against Different Environmental Stress Factors, Life, 13, 706, https://doi.org/10.3390/life13030706, 2023.
- Santschi, P. H., Hung, C.-C., Schultz, G., Alvarado-Quiroz, N., Guo, L., Pinckney, J., and Walsh, I.: Control of acid polysaccharide production and 234Th and POC export fluxes by marine organisms, Geophys. Res. Lett., 30, 1044, https://doi.org/10.1029/2002GL016046, 2003.
- Shaked, Y., and Lis, H.: Disassembling iron availability to phytoplankton. Front. Microbiol., 3, 123, https://doi.org/10.3389/fmicb.2012.00123, 2012.

The following paragraphs have been deleted:

**In section 3.2 Cell growth**

The highest peaks of algae biomass, 1.07 ($\pm$ 0.10) and 1.04 ($\pm$ 0.07) $\times 10^8$ cells L$^{-1}$, were recorded in the microcosms with intermediate $CO_2$ levels 350 µat and 600 µat (pH 8.10 and 7.90 respectively).

**In section 3.4 Phenolic contents of cells and exudates**

- The phenolic profile differences found inside and outside the cells could be explained by the different mechanisms to counter pH acidification inside (intrinsic buffers such as ionizable groups on amino acids, phosphates and other molecules; Na$^+$–H$^+$ exchangers and bicarbonate transporters, membrane permeability, among others) and the changes of metabolic pathways involved (Barcelos e Ramos et al., 2010; Casey et al., 2010) and outside, limited to membrane permeability and exuded material.

**Sections 3.6 Antioxidant activities and Conclusion have been rewritten**

**Report #1**

The manuscript by M. Rico et al presents data from a cultivation of *Emiliania h*. under four pH conditions. results concern the growth, phenolic compounds and total carbohydrate in both cells and medium.

The study might be of interest if the authors add some other variables (pigments, coccoliths,). in its present state, this ms can not be accepted as publication, due to relevant weaknesses.

New sections have been included:

- Section 2.4 Chlorophyll *a*
- Section 3.3 Chlorophyll *a*

- The ms totally lacks of statistical analysis. It does seem that for most of the data, no significant variation was revealed among the different treatments. if this was true, the results and discussion section has to be thoroughly re-written.

This study complements the one previously carried out by our research group and published by Samperio et al. (2017), who cultured the cells studied here (5 replicates) and monitored the changes in several parameters throughout different growth phases (growth, DOC, etc.). Some details of the growing conditions have been included in section 2.2 following suggestions from reviewer 2.

- Samperio-Ramos, G., Santana-Casiano, J. M., González-Dávila, M., Ferreira, S., and Coimbra, M. A.: Variability in the organic ligands released by Emiliania huxleyi under simulated ocean acidification conditions, AIMS Environ. Sci., 4(6), https://doi.org/10.3934/environsci.2017.6.788, 2017.

The cell densities (N) were determined from the average of 5 experimental batch cultures at each $p\text{CO}_2$ treatment. The goodness of the fit for each curve was estimated by the coefficient of correlation ($r2 > 0.95$).

The following new sections have been included in the manuscript. ANOVA studies and correlation matrix are included at the end of this document.

- **2.8 Statistical analysis.** A Pearson's correlation test and a one-way ANOVA were performed to determine the degree of relationship between pairs of variables and statistically significant differences between measurements, respectively. Both studies were conducted using the Jamovi program (2022) and *p*-values of <0.05 were considered statistically significant.
- **3.1 Carbonate chemistry parameters**

- the fig. 2 is emblematic:  in the legend, "diatoms" are mentioned while results cam from Emiliania h.; dark vs light color is referred to what?; these data correspond to which day ?

Figure 2 and its title have been changed as follows:

[Figure]

**Figure 2** Total sum of the identified intracellular polyphenols expressed as attomol cell$^{-1}$ (non-solid bars) and concentration (nM) of exuded polyphenols (solid bars) by *Emiliania huxleyi* cells grown under reduced pH conditions after eight culture days.

- fig.1: no SD bars

The SD bars overlap in the growth curves and were therefore avoided.

[Figure]

However, Fig. 1 has been changed as presented below.

A)

[Figure]

**Figure 1** Growth curves (A) and cell densities (B) of coccolithophore *Emiliania huxleyi* cultivated under four different pH conditions.

- material and methods:

- no mention on light, photoperiod, temperature...: these factors are probably the most relevant to shape the microalgal physiology.
- it is not clear: seawater or medium?

The highlighted sentences with the requested information have been included in the text:

Axenic cultures of *E. huxleyi* (strain RCC1238) were supplied by the Spanish Bank of Algae (BEA) in f/2 medium. *E. huxleyi* coccolithophore was cultured with an initial cell density of $10^6$ cells $L^{-1}$ at a constant temperature of 25 ºC, under complete photoperiod (24 h) with light

intensity of 200 µmol photons m$^{-2}$ and under different $p$CO$_2$-controlled seawater pH conditions (7.75, 7.90, 8.10, and 8.25), measured on the free hydrogen ion scale pH$_F$=-log[H+] with a Ross Combination glass body electrode calibrated daily with TRIS buffer solutions. For this purpose, a gaseous mixture of CO$_2$-free air and pure CO$_2$ (up to CO$_2$ levels 900, 600, 350, and 225 µatm, respectively) was bubbled in the culture medium (sterile filtered (0.1 µm) North Atlantic seawater (S = 36.48) obtained at the ESTOC site (29º10' N, 15º30' W) with an equipment that modulates the CO$_2$ flow once the desired pH is reached, keeping it constant (±0.02). Cells were frozen and stored at -80ºC.

Carbonate chemistry was monitored continuously in the experimental media and determined from pH, total alkalinity (TA), and total dissolved inorganic carbon (DIC). pH in the treatments was measured on the free hydrogen ion scale (pH $= -\log$ [H$^+$]), by immersing Orion Ross combination glass electrodes in the experimental media. The electrodes were calibrated daily using TRIS buffer solutions. The equilibration of the gas in the media of each treatment was achieved after a maximum of 24 h, observed by the evolution of pH. TA and DIC were determined using a VINDTA 3C system (González-Dávila et al., 2011). TA was determined by potentiometric titration with hydrochloric acid until the endpoint of carbonic acid was reached. DIC was analyzed by coulometric procedure after phosphoric acid addition. Certified Reference Material (provided by A. Dickson at Scripps Institution of Oceanography) was employed to assess the performance of the titration system, yielding an accuracy of 1.5 and 1.0 µmol kg$^{-1}$ for TA and DIC, respectively. The Seawater Carbonate package (Seacarb version 3.0), developed for R Studio software (R Development Core Team), was employed to calculate the values of $p$CO$_2$, considering the carbonic acid dissociation constants. A more detailed description is given by Samperio-Ramos et al. (2017).

- which was the target of pH manipulation?

The main aim of this work is stated in the first sentence of the last paragraph of the introduction:

This work aimed to determine how marine acidification may affect the composition of cells and exudates from *Emiliania huxleyi*.

**Abstract:** Cultures of the coccolithophore *Emiliania huxleyi* were grown under four different CO$_2$-controlled pH conditions (7.75, 7.90, 8.10, and 8.25) to improve understanding of its responses to ocean acidification scenarios.

Changes in pH generate stress conditions, either because at high pH drastically decrease the availability of trace metals such as Fe(II), a restrictive element for primary productivity (Wu et al., 2016), or because ROS are increased at acid pH (Bautista-Chamizo et al., 2019; Vázquez et al., 2022). The characterization of compounds exuded into the environment under stress conditions has allowed the development of important new lines of research in iron chemistry, for example, in different acidification scenarios. These compounds are crucial ligands in the formation of metal complexes to acquire micronutrients, sequester toxic metals, and to establish electrochemical gradients resulting in changes in the speciation, the bioavailability, and the toxicity of trace metals. Our research team has tested several compounds identified in the exudates of microalgae to study the effects on copper and iron chemistry in seawater:

- Pérez-Almeida et al., 2022. Ocean Acidification Effect on the Iron-Gallic Acid Redox Interaction in Seawater. Front. Mar. Sci., Sec. Marine Biogeochemistry, 9. https://doi.org/10.3389/fmars.2022.837363

- Arnone et al. (2024). Distribution of copper-binding ligands in Fram Strait and influences from the Greenland Shelf (GEOTRACES GN05): Science of The Total Environment, 909, 168162, https://doi.org/10.1016/j.scitotenv.2023.168162
- González et al., 2018. Iron complexation by phenolic ligands in seawater. Chemical Geology. 511, 380-388, 2018.https://doi.org/10.1016/j.chemgeo.2018.10.017
- A. López, et al., 2015. Phenolic profile of Dunaliella tertiolecta growing under high levels of copper and ironEnvironmental Science Pollution Res. 22 (19) 14820-14828, 2015. 10.1007/s11356-015-4717-y
- J. M. Santana-Casiano et al. 2014, Characterization of phenolic exudates from *Phaeodactylum tricornutum* and their effects on the chemistry of Fe(II)-Fe(III). Mar. Chem., 158, 10-16. https://doi.org/10.1016/j.marchem.2013.11.001

- material and methods: 48h (line 124)? does it mean that the experiment started after 48 h of cultivation of E.h. under the different conditions? in the fig.1 the day 0 corresponded to this time? which was the cell concentration at this time?

These two days correspond to the log phase and are included in the 8 days of culture monitoring. We distinguish the three stages in the growth curves of *E. huxleyi*: initial or log phase (these two days, until 2nd day = the first 48h mentioned above), exponential (EP, from 3rd to 5th day) and steady (SP, from 6th to 8th day) phases.

- material and methods: please explain why a first extraction in acetone and then in methanol. what is the role of acetone?

The described procedure was used for chlorophyll measurement with a mixture of acetone and hexane as described in section 2.4. For the extraction of polyphenols, and for the DPPH and FRAP assays, methanol was used following the usual protocol in our laboratory (López et al., 2015; Santiago-Díaz et al., 2023) and that used by Vicente et al. (2021), and for the extraction of carbohydrates, 5 mL of acidified water (pH = 2) was used, 1.5 mL was freeze-dried and the residue was dissolved as described in section 2.3.

- López et al.: Phenolic profile of *Dunaliella tertiolecta* growing under high levels of copper and iron, Environ. Sci. Pollut. Res. 22, 14820–14828. https://doi.org/10.1007/s11356-015-4717-y, 2015.

- Santiago-Díaz et al.: Copper toxicity leads to accumulation of free amino acids and polyphenols in Phaeodactylum tricornutum diatoms, Environ. Sci. Pollut. Res., 30, 51261–51270, https://doi.org/10.1007/s11356-023-25939-0, 2023.

- Vicente et al. Production and bioaccessibility of *Emiliania huxleyi* biomass and bioactivity of its aqueous and ethanolic extracts, J. Appl. Phycol. 33, 3719–3729, https://doi.org/10.1007/s10811-021-02551-8, 2021

- quantification of phenols: did the authors used some pure standards?

We used the standards described in the following sections:

- Introduction
  Intra- and extracellular phenolic compounds (gallic acid (GAL), protocatechuic acid (PCA), p-coumaric acid (COU), ferulic acid (FA), catechin (CAT), vanillic acid (VAN), epicatechin (ECAT), syringic acid (SYR), rutin (RU) and gentisic acid (GA)) were identified and quantified by RP-HPLC.

- 2.1 Chemicals
  Polyphenol standards were supplied as follows: GAL, PCA, COU, FA, CAT, VAN, ECAT, and SYR by Sigma–Aldrich Chemie (Steinheim, Germany); RU and GA by Merck (Darmstadt, Germany

- The description of the chromatographic analysis can be found in section 2.5, where the standards were cited: For quantification, simultaneous monitoring was set at 270 nm (GAL, PCA, CAT, VAN, RU, ECAT, and SYR) and 324 nm (GA, COU, and FA).

- calibration curve: please explain why there is a "b" factor (y = ax + b; for carbohydrates, frap) also especially when it is negative (dpph)?

DPPH assay: We prepare a calibration curve of DPPH in methanol at different concentrations. The absorbance corresponding to each concentration is measured and the following calibration curve is constructed:

[Figure]

The absorbances of the algal samples correspond to the concentration of not inhibited DPPH. Therefore, subtracting the concentrations obtained through the calibration curve from the initial DPPH concentration allows us to calculate the amount of inhibited DPPH.

**Calibration curve for FRAP determination:**

[Figure]

This method has been performed according to the literature, where this factor is always present, probably due to parallel reactions or impurities in the reagents, which are present in all standards and samples, and therefore do not affect the final result. Regardless of whether the reducing power is measured as reduced Fe(III) from a calibration curve prepared with Fe(II) or whether it is measured with a standard such as Trolox, this factor b is present.

**Glucose calibration curve:**

[Figure]

The following articles corroborates this:
- *Noreen et al., 2017:* $y=0.00006x+0.1887$. https://doi.org/10.1016/j.apjtm.2017.07.024.
- *Alam et al. 2014:* $y=5.4901x+0.2547$. https://doi.org/10.1155/2014/296063.
- *Mukherjee et al.,* 2019. https://link.springer.com/article/10.1007/s12649-017-0053-4

- line 165: cell extracts? how they have been done?

- this aspect has been clarified above.

- LINE 173: The highest peak? but NO SIGNIFICANT!

- This sentence has been deleted.

- lines 186-189: re-write!

These lines have been rewritten as follows:

Discrepancies found in the literature regarding acidification effects and responses of *E. huxleyi* coccolithophores may be due not only to different environmental factors and culture conditions, etc. (Gafar et al., 2019; Tong et al., 2017). Langer et al. (2009) observed substantial differences in sensitivity to acidification in four different *E. huxleyi* strains with different responses in all parameters tested.

- table 1: unit of cell and exuded? seems to be different, also from the "levels" (nM).

The title of Table 1 has been changed as follows and a mistake was corrected in the amount of PCA exuded at pH 7.75. The total concentration was correct.

**Table 1.** Amounts of intracellular and exuded phenolic compounds by cells of *E. huxleyi* grown under different pH conditions.

| Phenolic compound | pH 7.75 (pCO$_2$ 900µat) | | pH 7.90 (pCO$_2$ 600µat) | | pH 8.10 (pCO$_2$ 350µat) | | pH 8.25 (pCO$_2$ 225µat) | |
|---|---|---|---|---|---|---|---|---|
| | Cell | Exuded | Cell | Exuded | Cell | Exuded | Cell | Exuded |
| GAL[a] | 0.13±0.03 | - | - | - | 0.25±0.07 | - | - | 45±0 |
| PCA[a] | - | 151±12 | 0.16±0.01 | 34±2 | - | 26.3±0.2 | - | 298±22 |
| ECAT[a] | 1.2±0.1 | 34±1 | - | 40±5 | - | 70±2 | - | 77±6 |
| VAN[a] | 6.44±0.06 | - | 2.5±0.1 | - | 2.30±0.05 | - | 2.5±0.2 | - |
| COU[a] | - | - | - | - | 1.4±0.2 | - | - | - |
| RU[a] | 1.47±0 | 12.1±0.4 | | 20±2 | | 13.2±0.3 | | 11.2±0.8 |
| Sum[a] | 9.24±0.19 | 197.1±12.3 | 2.66±0.11 | 94±9 | 3.95±0.32 | 109.5±2.5 | 2.5±0.2 | 431.2±28.8 |
| Concentration (nM)[b] | 18.0±0.9 | | 9.6±0.8 | | 11.7±0.3 | | 43±3 | |

[a]Results are expressed as attomole cell$^{-1}$ (means ± standard deviation of three measurements).

[b]Results are expressed as nanomole L$^{-1}$ (means ± standard deviation of three measurements).

Abbreviations: GAL: gallic acid; PCA: protocatechuic acid; ECAT: epicatechin; VAN: vanillic acid; COU: p-coumaric acid; RU: rutin.

All quantities are expressed as attomole cell$^{-1}$, specified with the superscripts[a], except for the nanomolar concentration, specified with the superscripts[b] and footnotes in the table.

- I suggest also to the authors to give a look on 10.1080/07388551.2021.1874284, that might help for the discussion.

The aim of the study suggested by the reviewer is to provide an overview of current knowledge on phenolic compounds in microalgae. The article focuses on factors that influence the variation in total polyphenol and flavonoid content: microalgal biodiversity, chemodiversity between groups, different analytical methodologies, physiological state based on how the cells are maintained or cultured, e.g. light or other factors. The study cites two of our previously reported manuscripts (references 19 and 23):

- 19. Rico et al.: Variability of the phenolic profile in the diatom *Phaeodactylum tricornutum* growing under copper and iron stress. Limnol Oceanogr. 2013; 58: 144–152.
- 23. López et al.: Phenolic profile of *Dunaliella tertiolecta* growing under high levels of copper and iron. Environ Sci Pollut Res Int. 2015; 22: 14820–14828.

However, pH is the only variable modified in our study focused on the effect of acidification on *E. huxleyi* through $CO_2$ acidification, so changes in organic matter should be linked only to the effect of this pH change and its consequences (changes in the availability of essential metals such as iron). We used the same strains as well as the cultivation conditions (lighting, seawater, nutrients, temperature, etc.) so the influence of all these factors should be the same in all cultures.

- legend: three measurements: three replicates? (different cultures? or technical replicates?)

Five experimental batch cultures were carried out at each pH treatment. Three replicates refer to different cultures.

- lack of significativity tests in all the studies

The version 2.3 of jamovi program (2022) has been used for statistical analyses (retrieved from https://www.jamovi.org).

A Pearson's correlation test was performed to determine the degree of relationship between pairs of variables and a one-way ANOVA to determine statistically significant differences between measurements. Both studies were conducted using the Jamovi program (2022) and $p$-values of $<0.05$ were considered statistically significant.

**Correlation Matrix between pH 8.25 – 7.75**

| | | Intracellular TCH | Extracellular TCH | Intra TPC | Extra TPC | RSA cells | RSA exudates | FRAP cells |
|---|---|---|---|---|---|---|---|---|
| Intracellular TCH | R de Pearson | — | | | | | | |
| | valor p | — | | | | | | |
| Extracellular TCH | R de Pearson | -0.257 | — | | | | | |
| | valor p | 0.731 | — | | | | | |
| Intra TPC | R de Pearson | -0.615 | 0.416 | — | | | | |
| | valor p | 0.948 | 0.153 | — | | | | |
| Extra TPC | R de Pearson | 0.396 | -0.810 | -0.073 | — | | | |
| | valor p | 0.166 | 0.993 | 0.568 | — | | | |
| RSA cells | R de Pearson | -0.280 | 0.728 * | 0.829 ** | -0.240 | — | | |
| | valor p | 0.749 | 0.020 | 0.005 | 0.716 | — | | |
| RSA exuded | R de Pearson | 0.213 | 0.456 | -0.567 | -0.730 | -0.249 | — | |
| | valor p | 0.306 | 0.128 | 0.929 | 0.980 | 0.724 | — | |
| FRAP | R de Pearson | -0.543 | 0.129 | 0.718 * | 0.133 | 0.527 | -0.565 | — |
| | valor p | 0.918 | 0.380 | 0.022 | 0.376 | 0.090 | 0.928 | — |

$* p < .05$, $** p < .01$, $*** p < .001$

TCH: Total carbohydrates
TFC Total phenolic content
RSA: Radical Scavenging Activity
FRAP: Ferric Reducing Power Assay

**Correlation Matrix between pH 8.1 and 7.75**

|  |  | Intracellular TCH | Extracellular TCH | Intra TPC | Extra TPC | RSA cells | RSA exuded | FRAP |
|---|---|---|---|---|---|---|---|---|
| Intracellular TCH | R of Pearson | — |  |  |  |  |  |  |
|  | p value | — |  |  |  |  |  |  |
| Extracellular TCH | R of Pearson | 0.463 | — |  |  |  |  |  |
|  | p value | 0.177 | — |  |  |  |  |  |
| Intra TPC | R of Pearson | -0.567 | 0.307 | — |  |  |  |  |
|  | p value | 0.879 | 0.277 | — |  |  |  |  |
| Extra TPC | R of Pearson | -0.662 | 0.104 | 0.965 *** | — |  |  |  |
|  | p value | 0.924 | 0.423 | < .001 | — |  |  |  |
| RSA cells | R of Pearson | -0.083 | 0.803 * | 0.810 * | 0.665 | — |  |  |
|  | p value | 0.562 | 0.027 | 0.025 | 0.075 | — |  |  |
| RSA exuded | R of Pearson | 0.736 * | -0.043 | -0.941 | -0.976 | -0.627 | — |  |
|  | p value | 0.048 | 0.532 | 0.997 | 1.000 | 0.909 | — |  |
| FRAP | R of Pearson | -0.625 | 0.201 | 0.778 * | 0.786 * | 0.588 | -0.709 | — |
|  | p value | 0.908 | 0.351 | 0.034 | 0.032 | 0.110 | 0.943 | — |

Nota. * $p < .05$, ** $p < .01$, *** $p < .001$

TCH: Total carbohydrates
TPC: Total phenolic content
RSA: Radical Scavenging Activity
FRAP: Ferric Reducing Power Assay

One way factor ANOVA

| | F | gl1 | gl2 | p | |
|---|---|---|---|---|---|
| RSA cells | 290.89 | 3 | 2.07 | 0.003 | <.01 |
| RSA exudates | 30.38 | 3 | 1.90 | 0.037 | <.05 |
| FRAP cells | 1.26 | 3 | 2.12 | 0.464 | |
| Total exuded phenolics (nM) | 64.69 | 3 | 1.88 | 0.019 | <.05 |
| Exuded phenolics per cell | 14529.66 | 3 | 1.67 | <.001 | |
| Protocatechuic acid | 256.02 | 3 | 2.00 | 0.004 | <.01 |
| Epicatechin | 40.88 | 3 | 1.94 | 0.026 | <.05 |
| Rutin | 7.25 | 3 | 1.70 | 0.152 | |
| Intracellular-phenolics | 173.30 | 3 | 2.18 | 0.004 | <.01 |
| Intra-Vanillic acid | 1308.13 | 3 | 2.08 | <.001 | |

RSA: Radical Scavenging Activity
FRAP: Ferric Reducing Power Assay
TCH: Total carbohydrates
TPC: Total phenolic content

**Report #2**

The manuscript by Rico et al., demonstrated the responses of phenolics and carbohydrates of Emiliania huxleyi to ocean acidification conditions. They found that the intracellular phenolic compounds increased under low pH conditions, and these findings have certain implications for the marine food webs and biogeochemical cycles. However, I do have some concerns need to be addressed before it can be accepted for the publication in BG.

This study complements the one previously carried out by our research group and published by Samperio et al. (2017), who cultured the cells studied here (5 replicates) and monitored the changes in several parameters throughout different growth phases (growth, DOC, etc.). Some details of the growing conditions have been included in section 2.2 following suggestions from another reviewer.

- Samperio-Ramos, G., Santana-Casiano, J. M., González-Dávila, M., Ferreira, S., and Coimbra, M. A.: Variability in the organic ligands released by Emiliania huxleyi under simulated ocean acidification conditions, AIMS Environ. Sci., 4(6), https://doi.org/10.3934/environsci.2017.6.788, 2017.

New sections have been included and are listed at the end of this document:

- **2.4 Chlorophyll *a***
- **2.8 Statistical analysis**
- **3.1 Carbonate chemistry parameters**
- **3.3 Chlorophyll *a***

General Comments:

**Abstract:** The Abstract was not well structed, and is wordy. It should be very concise and highlight the significance of the findings presented in this study. Please rephrase it to be more concise.

The Abstract has been modified as follows:

**Abstract.** Cultures of the coccolithophore *Emiliania huxleyi* were grown under four different $CO_2$-controlled pH conditions (7.75, 7.90, 8.10, and 8.25) to improve understanding of its responses to ocean acidification scenarios. Acidification did not significantly affect final cell densities and carbohydrate contents. Intra- and extracellular phenolic compounds were identified and quantified by Reverse Phase-High Performance Liquid Chromatography (RP-HPLC), with the highest concentrations of total exuded phenolics at pH 8.25 (43±3 nM) and 7.75 (18.0±0.9 nM). Accumulation of intracellular phenolic compounds was observed in cells with decreasing pH, reaching the maximum level (9.24±0.19 attomole cell$^{-1}$) at the lowest pH (7.75). The phenolic profiles presented significant changes in exuded epicatechin and protocatechuic acid ($p<0.05$ and 0.01, respectively), and intracellular vanillic acid ($p<0.001$), which play an essential role as antioxidants, and in the availability of trace metals. A significant increase in chlorophyll *a* content was observed in cells grown at the most acidic pH ($p<0.01$), which also showed significantly higher radical inhibition activity ($p<0.01$). However, no significant differences were found between the iron reducing activities and the radical scavenging activities of the compounds present in the exudates. The nature and concentration of the organic compounds present in the culture medium may favour or inhibit the local growth

of specific algal species, and influence trace metal bioavailability affecting the biogeochemical cycling of carbon and microbial functional diversity.

Discussion:

1: The discussion could better articulate how the observed changes in polyphenols and carbohydrates could impact broader ecological processes. Specifically, the implications for the marine food web and biogeochemical cycles could be explored in more depth.

Several sentences have been included to highlight the implications for the marine food web in more depth:

[revised manuscript text omitted]

- Sections 3.5: …. These results agree partially with those reported by Jia et al. (2024), who found that lowering seawater pH promoted protein synthesis in microalgae *Chlorella* sp., *P. tricornutum*, and *C. muelleri* grown at pH 7.8 and 7.5, compared to those harvested at pH 8.1, concluding that acidification improves the efficiency of carbon assimilation and provides more carbon skeletons for amino acids and protein synthesis.

2: The manuscript could benefit from a more detailed comparison with existing literature. Specifically, how do these results align or contrast with the findings of similar studies in terms of the magnitude and direction of changes in polyphenol and carbohydrate content?

We have included the following sentences where results of exposure to acidification of other microalgae species are discussed.

- Sections 2.4 and 3.3 focused on Chlorophyll *a* analysis and comparison with other species have also been included.

- Section 3.4 has been completely changed: …In contrast, no significant changes were observed by Jia et al. (2024) in the three microalgae species *Chlorella* sp., *P. tricornutum,* and *C. muelleri harvested under elevated pCO2* (pH 8.1, 7.8, and 7.5), and

- …. Previous studies on diatoms *P. tricornutum* grown under copper stress showed a significant correlation (r=0.9999; p<0.05) between accumulated phenolic compounds and malondialdehyde (MDA), commonly produced by an increase in ROS (Rico et al., 2024). ROS production was also correlated with acidification stress conditions in single and multispecies toxicity tests performed by Bautista-Chamizo et al. (2019) using *T. Chuii*, *N. gaditana* and *P. tricornutum* microalgae, where *P. tricornutum* and *N. gaditana* exhibited a significant increase in the percentage of intracellular ROS when exposed to pH 7.4 and pH 6.0, which was more pronounced for *N. gaditana* cells at pH 6.0. Consistent with the loss of cell membrane integrity observed after 48 h at pH 6.0, a 10% and 56% of non-viable cells were found for *P. tricornutum* and *N. gaditana*, respectively, while viable cells remained close to the control for *T. chuii* cells.
  Section 3.5: During the exponential phase, the contribution of dissolved carbohydrates to excreted DOC was higher (18–37%) than during the stationary phase (14–23%), significantly increased as time elapsed from the exponential to the stationary phase (Samperio et al., 2017). However, acidification of the culture medium with $CO_2$ did not affect the levels of carbohydrates exuded per cell in any of the three growth phases, as these levels did not change significantly in any of them as the pH dropped to pH 7.75. The amount of total intracellular carbohydrates also remained constant between pH 8.25 and pH 7.75. The coccolithophore *E. huxleyi* have shown diverse metabolic responses to ocean acidification and to combinations of ocean acidification with other environmental factors with significant differences between strains (Gafar et al., 2019; Langer et al., 2009; Mackey et al., 2015; Tong et al. 2017;).

- ….. These results partially agree with those reported by Jia et al. (2024), who found that lowering seawater pH up to 7.8 and 7.5 promoted protein synthesis in microalgae *Chlorella* sp., *P. tricornutum*, and *C. muelleri* compared to those harvested at pH 8.1, concluding that acidification improves the efficiency of carbon assimilation and provides more carbon skeletons for amino acids and protein synthesis.

- Grosse et al. (2020) investigated the effects of seawater acidification on dissolved and particulate amino acids and carbohydrates in arctic and sub-arctic planktonic communities in two large-scale experiments in a pH range similar to ours here (control mesocosm: *p*$CO_2$ 185 µatm /pH 8.32; mesocosm *p*$CO_2$ between 270 and 1420 µatm/pH 8.18–7.51). The authors concluded that the relative composition of amino acids and carbohydrates did not change as a direct consequence of increased *p*$CO_2$, and the observed changes depended mainly on the composition of the phytoplankton community.

Regarding the magnitudes, Lopez et al. (2015) reported similar concentrations of polyphenols (between 9.4 and 8.4 nM) and the contents in the cells are in the same range than here (attomole):

- These results partially agree with those reported by López et al. (2015) for *Dunaliella tertiolecta* growing under stress conditions induced by high levels of copper, where the concentration of phenolic compounds declined from 9.4±0.6 nM in seawater cultures without Cu addition to 8.4±0.4 and 8.6±0.4 nM in the

copper enriched seawater, and increased 1.4 times concerning the control into the cells grown under the highest Cu level.

3: Broader Impacts: The results are discussed primarily in the context of *Emiliania huxleyi*. Expanding the discussion to consider potential impacts on other phytoplankton species, could provide a more holistic view of ocean acidification impacts.

In addition to all the comparisons and references already included in the manuscript, the following sentences have been added

- Section 3.2: Bautista-Chamizo et al. (2019) exposed microalgae *Tetraselmis chuii* (*T. Chuii*), *Nannochloropsis gaditana* (*N. gaditana*) and *P. tricornutum* to pH 6.0, 7.4 and pH 8.0 as a control, observing growth inhibition of 12%, 61% and 66% at pH 6.0, respectively, in single toxicity tests, with *T. chuii* being the most resistant species to $CO_2$ enrichment. At pH 7.4 only *P. tricornutum* showed a significant decrease in cell abundance (16%), while no differences were found for *T. chuii* and *N. gaditana*, demonstrating that sensitivity to acidification depends on the microalgae species. However, the growth rate of all three microalgae *Chlorella* sp., *P. tricornutum* and *Chaetocetos muelleri (C. muelleri)* harvested at pH levels (8.1, 7.8, and 7.5) was significantly enhanced (Jia et al., 2024). The differences observed in the growth behavior of microalgae, mainly of *P. tricornutun* diatoms, in these two last studies could be due to the culture conditions, e.g. in the first study, Bautista-Chamizo et al. (2019) used seawater collected in the Bay of Cádiz (Spain) (with initial pH 8.0, salinity of 34, temperature at 23 °C and initial cell density $3 \times 10^4$ cells $mL^{-1}$), and in the second study Jai et al. (2024) used artificial seawater prepared by mixing sea salt (Haizhixun, China) with purified water (salinity of 30 ± 1‰, temperature maintained at 18 ± 0.5 °C, initial pH at 8.1, no data on initial cell density).

Section 3.3 focuses on chlorophyll *a* analysis and the results are compared with those obtained in other acidification studies with *E. huxleyi* and other microalgae species:

[revised manuscript text omitted]

Specific comments:

1: Line 37-39: Some controversial findings regarding the calcification of coccolithophore to ocean acidification should be mentioned.

We agree with the reviewer that a study of the controversial findings regarding the calcification would be interesting, but that part is beyond the scope of our research, which focuses more on the change in both polyphenols and carbohydrates with pH variation. However, as indicated in the previous paragraph we have included same references to this effect.

2: Line 82-84: There were at least two studies examined the responses of phenolic compounds in marine primary producers to ocean acidification (Arnold et al., 2012, PLOS ONE; Jin et al., 2015, Nature Communications, 6:8714), where are more relevant to the present study should be acknowledged here.

The authors agree and therefore cite the article twice in the manuscript, and completed the information as follows:

- This transfer of accumulated phenolic compounds to higher trophic levels could have serious consequences for the marine ecosystem and seafood quality. Dupont et al. (2014) have shown that survival of adult boreal shrimp (*Pandalus borealis*) was affected when exposed for 3 weeks at pH 7.5 with an increase in mortality of 63% compared to those grown at pH 8.0 (at the sampling site), showing also changes in appearance and taste.

- Section 3.6, in the last paragraph:

  Several studies evidenced enhanced oxidative stress in cells in acidified seawater, through an increase in ROS that correlated significantly with the accumulation of polyphenols (Bautista-Chamizo et al., 2019; Vázquez et al., 2022). Ocean acidification will directly affect marine organisms, altering the structure and functions of ecosystems. These changes may cause cascading effects in the marine food web, varying the macromolecular composition of consumers (Jia et al., 2024). The accumulation of phenolic compounds leads to functional consequences in primary and secondary producers, with the possibility that fishery industries could be influenced as a result of progressive ocean change (Gattuso et al., 2015; Jin et al., 2015; Trombetta et al., 2019).

By other hand, Arnold et al. (2012) study is focused on marine plants. The authors consider it important to mention this effect seen in higher plants, but not to focus our discussion on them, but rather on microalgae.

- Line 284: However, Arnold et al. (2012) reported a loss of phenolics in the seagrasses *Cymodocea nodosa, Ruppia maritima,* and *Potamogeton perfoliatus* grown in acidified seawater, where the pH decreased up to 7.3, and the $CO_2$ level increased ten-fold.

3: Line 115-125: Have you monitored the carbonate chemistry parameters over the 8 days experimental duration? These parameters are important for a typic OA research. Please clarify.

All these parameters were monitored, discussed and reported previously by Samperio et al. (2017). We have included the following paragraph in section 2.2 and a new section called 3.1 Carbonate chemistry parameters.

Section 2.2:

- Carbonate chemistry was monitored continuously in the experimental media and determined from pH, total alkalinity (TA), and total dissolved inorganic carbon (TC). pH in the treatments was measured on the free hydrogen ion scale (pH = −log [H$^+$]), by immersing Orion Ross combination glass electrodes in the experimental media. The electrodes were calibrated daily using TRIS buffer solutions. The equilibration of the gas in the media of each treatment was achieved after a maximum of 24 h, observed by the evolution of pH. TA and TC were measured at the beginning and end of the experiment on days 0 and 8. TA was determined by potentiometric titration with hydrochloric acid until the endpoint of carbonic acid was reached utilizing a VINDTA 3C system (Mintrop et al., 2000). TC was analyzed by coulometric procedure after phosphoric acid addition (González-Dávila et al. 2011). Certified Reference Material (provided by A. Dickson at Scripps Institution of Oceanography) was employed to assess the performance of the titration system, yielding an accuracy of 1.5 and 1.0 µmol kg$^{-1}$ for TA and TC, respectively. The Seawater Carbonate package (Seacarb version 3.0), developed for R Studio software (R Development Core Team), was employed to calculate the values of pCO$_2$, considering the carbonic acid dissociation constants. A more detailed description is given by Samperio-Ramos et al. (2017).

**3.1 Carbonate chemistry parameters**

- Preliminary tests indicated good stability of carbonate chemistry parameters in media, obtained by the CO$_2$ regulation system (Table 1). After CO$_2$-equilibration, initial TC values ranged from $1905 \pm 26$ to $2215 \pm 16$ at a TA mean value of $2386 \pm 16$ µmol kg$^{-1}$ (one-way ANOVA; F=1.729, p=0.2382 among treatments), corresponding to a CO$_2$ range of $225 \pm 1$ and $914 \pm 13$ µatm. Although carbonate chemistry in CO$_2$-manipulated experiments can be strongly affected by biological activity (Howes et al. 2017; Miller and Kelley 2021), during our experiments TA and TC remained fairly stable (t-tests; p>0.05) within treatments, over the 8-day experimental period.

**Table 1.** Carbonate chemistry parameters in experimental media for each pH treatment at day 0 and day 8: total alkalinity (TA), total dissolved inorganic carbon concentration (TC)) and estimated pCO$_2$ (µatm). Means and standard deviations were calculated from sampling (n = 3).

| pH-Treatments | TA (µmol kg$^{-1}$) | | TC (µmol kg$^{-1}$) | | pCO$_2$ (µatm) | |
|---|---|---|---|---|---|---|
| | Day 0 | Day 8 | Day 0 | Day 8 | Day 0 | Day 8 |
| 8.25 | $2376 \pm 12$ | $2335 \pm 25$ | $1905 \pm 26$ | $1869 \pm 61$ | $225 \pm 1$ | $221 \pm 4$ |
| 8.10 | $2380 \pm 15$ | $2329 \pm 28$ | $2012 \pm 28$ | $1971 \pm 44$ | $353 \pm 2$ | $349 \pm 5$ |
| 7.90 | $2390 \pm 17$ | $2347 \pm 40$ | $2129 \pm 47$ | $2085 \pm 36$ | $616 \pm 12$ | $599 \pm 8$ |
| 7.75 | $2401 \pm 14$ | $2365 \pm 26$ | $2215 \pm 16$ | $2178 \pm 39$ | $914 \pm 18$ | $925 \pm 27$ |

In addition, the following paragraphs have been included in the sections 3.4 and 3.5

- Samperio et al. (2017) reported an increase in dissolved organic carbon exudation by 19% and 15% during exponential and stationary phases, respectively, as $CO_2$ levels increased from 225 µatm (177.06 ± 10.95 fmol C cell$^{-1}$ day$^{-1}$) to 900 µatm (209.74 ± 50.00 fmol C day$^{-1}$ cell$^{-1}$) in the culture medium. The authors suggested that ocean acidification could significantly enhance the release of phenolic compounds when *E. huxleyi is* grown under low-iron conditions. They detected phenolic compounds only in the stationary phase and their release rate was affected by $CO_2$ conditions, observing a strong correlation between the concentrations of produced phenolic compounds with exuded dissolved organic carbon, indicating that these compounds constituted a relatively constant fraction of the organic matter excreted by *E. huxleyi.* Extracellular release of phenolic compounds was statistically higher at $pCO_2$ 900 µatm (0.41 ± 0.02 fmol C cell$^{-1}$ day$^{-1}$) than at $pCO_2$ 350µatm (0.36 ± 0.02 fmol C cell$^{-1}$day$^{-1}$; Tukey contrast: t value = 2.495; $p < 0.1$). While Samperio et al. (2017) studied the total polyphenol contents through the Arnow spectrophotometric assay, in our study we have identified some polyphenols (Table 1), concluding that the highest concentrations of these polyphenols identified were exuded in the cultures at pH 8.25 (43±3 nM) and 7.75 (18.0±0.9 nM). The high level of these exuded polyphenols at pH 8.25 could be due to the redox chemistry of inorganic Fe, intimately linked to the pH (Pérez-Almeida et al., 2019). When the pH increases the solubility of inorganic Fe(III) decreases and the oxidation rate constants of Fe(II) increases. Wu et al. (2016) studied the interaction between Fe and ten phenols at pH = 8.0, finding that only caffeic acid, gallic acid, and protocatechuic acid protected 69%, 64% and 33% of the initial iron (II), respectively, due to the chelating capacity of the catechol and galloyl groups with Fe(II).

- Samperio et al. (2017) reported higher contributions of dissolved carbohydrates to excreted DOC during the exponential phase (18–37%) than during the stationary phase (14–23%), significantly increased as time elapsed from the exponential to the stationary phase. However, acidification of the culture medium with $CO_2$ did not affect the levels of carbohydrates exuded per cell in any of the three growth phases, as these levels did not change significantly in any of them as the pH dropped to pH 7.75. The amount of total intracellular carbohydrates also remained constant between pH 8.25 and pH 7.75. The coccolithophore *E. huxleyi* have shown diverse metabolic responses to ocean acidification and to combinations of ocean acidification with other environmental factors with significant differences between strains (Langer et al., 2009; Tong et al. 2017; Gafar et al. 2019).

4: Line 170-175: Should it be possible to calculate the specific growth rates for the exponential phase of each growth curve and then compare them under different pH conditions?

No significant differences were found between the specific growth rates of five replicates:

| pH | Specific Growth Rate (day^-1) |
|---|---|
| 7.75 | 0.564±0.009 |
| 7.9 | 0.582±0.014 |
| 8.1 | 0.587±0.014 |
| 8.25 | 0.575±0.011 |
| ANOVA ($p$=0.336) | |

5: Line 210: diatoms?

This has been corrected

10: Line 255: Typo of the citation "Santana_Casiano et al., 2014"

This has been corrected